# Shortwave radiative forcing, rapid adjustment, and feedback to the surface by sulphate geoengineering: Analysis of the Geoengineering Model Intercomparison Project G4 scenario

Hiroki Kashimura[1,a], Manabu Abe[1], Shingo Watanabe[1], Takashi Sekiya[1], Duoying Ji[2], John C. Moore[2], Jason N. S. Cole[3], and Ben Kravitz[4]

[1]Japan Agency for Marine-Earth Science and Technology, Yokohama, Japan.
[2]College of Global Change and Earth System Science, Beijing Normal University, Beijing, China.
[3]Canadian Centre for Climate Modelling and Analysis, Environment and Climate Change Canada, Victoria, British Columbia, Canada.
[4]Atmospheric Sciences and Global Change Division, Pacific Northwest National Laboratory, Washington, USA.
[a]current affiliation: Department of Planetology/Center for Planetary Science, Kobe University, Kobe, Japan.

*Correspondence to:* Hiroki Kashimura (hiroki@cps-jp.org)

**Abstract.** This study evaluates the forcing, rapid adjustment, and feedback of net shortwave radiation at the surface in the G4 experiment of the Geoengineering Model Intercomparison Project by analysing outputs from six participating models. G4 involves injection of $5\,\mathrm{Tg\,yr^{-1}}$ of $SO_2$, a sulphate aerosol precursor, into the lower stratosphere from year 2020 to 2069 against a background scenario of RCP4.5. A single layer atmospheric model for shortwave radiative transfer is used to estimate the direct forcing of solar radiation management (SRM), and rapid adjustment and feedbacks from changes in the water vapour amount, cloud amount, and surface albedo (compared with RCP4.5). The analysis shows that the globally and temporally averaged SRM forcing ranges from $-3.6$ to $-1.6\,\mathrm{W\,m^{-2}}$, depending on the model. The sum of the rapid adjustments and feedback effects due to changes in the water vapour and cloud amounts increase the downwelling shortwave radiation at the surface by approximately 0.4 to $1.5\,\mathrm{W\,m^{-2}}$ and hence weaken the effect of SRM by around 50 %. The surface albedo changes decrease the net shortwave radiation at the surface; it is locally strong ($\sim -4\,\mathrm{W\,m^{-2}}$) in snow and sea ice melting regions, but minor for the global average. The analyses show that the results of the G4 experiment, which simulates sulphate geoengineering, include large inter-model variability both in the direct SRM forcing and the shortwave rapid adjustment from change in the cloud amount, and imply a high uncertainty in modelled processes of sulphate aerosols and clouds.

## 1 Introduction

Geoengineering, or climate engineering, is the deliberate large-scale manipulation of the planetary environment to counteract anthropogenic climate change (e.g., Shepherd, 2009). One major category of geoengineering for lessening the effects of global warming is solar radiation management (SRM), which aims to reduce the amount of solar radiation at the Earth's surface. One of several SRM approaches (e.g., Lane et al., 2007) is to mimic a volcanic eruption by injecting sulphate aerosol precursors, such as $SO_2$, into the stratosphere (e.g., Budyko, 1974; Crutzen, 2006); this approach is called sulphate geoengineering. Large

volcanic eruptions carry $SO_2$ gases into the stratosphere; these gases are photo-chemically oxidized to form sulphate aerosols, which have high reflectivity in the visible and ultraviolet regions of the electromagnetic spectrum. Sulphate aerosols increase the solar reflectivity of the atmosphere, decreasing the shortwave radiation (SW) reaching the surface, and therefore cooling the air temperature. For example, the 1991 eruption of Mount Pinatubo reduced the globally averaged surface air temperature

by up to 0.5 K (Parker et al., 1996).

To explore the cooling effect of and the climate responses from sulphate geoengineering, or more generally SRM, several climate-modelling groups performed various experiments using global climate models or Earth System Models (ESMs). Some experiments involved simplifying the net effects of SRM by reducing the solar constant (Govindasamy and Caldeira, 2000; Bala et al., 2008; Govindasamy et al., 2002, 2003; Matthews and Caldeira, 2007); whereas, the studies listed in Table 1 have

simulated sulphate geoengineering with models that can partly or fully calculate the production of sulphate aerosols from the injected $SO_2$ and the dynamical transportation. The listed studies used different forcing for geoengineering, different scenarios for the baseline, and different models. Therefore, it is difficult to compare these studies or evaluate the uncertainty in the geoengineering simulations. However, Jones et al. (2010) compared the results of two different models in an experiment similar to that of Robock et al. (2008). They showed the different responses by the two models and emphasized the importance

of intercomparing many different climate models with a common experimental design in order to assess the impact of the geoengineering.

The Geoengineering Model Intercomparison Project (GeoMIP) (Kravitz et al., 2011) was established to coordinate simulations with a common framework and to determine the robust effects and responses to geoengineering processes. For the first series of GeoMIP experiments, four experiments named G1, G2, G3, and G4 were proposed. The first two are designed

to counteract quadrupled $CO_2$ radiative forcing (G1) and a 1 % increase in the $CO_2$ concentration per year (G2) by simply reducing the solar constant. The last two are designed to inject $SO_2$ into the lower stratosphere and decrease SW flux reaching the surface by increasing the SW reflection by sulphate aerosols. Both G3 and G4 use the RCP4.5 scenario for the baseline experiment and inject $SO_2$ every year from 2020 to 2069. The amount of $SO_2$ injected in G3 gradually increases to maintain the net radiative flux at the top-of-atmosphere (TOA) at the 2020 levels, while the radiative forcing of the green-

house gases increases according to the RCP4.5 scenario. Conversely, in G4 the $SO_2$ injection rate is fixed at $5\ \mathrm{Tg\,yr^{-1}}$. A summary of the G1–G4 studies is presented by Kravitz et al. (2013d) and the latest list of GeoMIP studies is available at http://climate.envsci.rutgers.edu/GeoMIP/index.html.

As summarized by Kravitz et al. (2013d), studies analysing GeoMIP experiments have explored and clarified climate model responses to radiative forcing and its dependence on various factors. In addition, the dependence (or uncertainty) of the direct

forcing to the net surface SW induced by sulphate aerosol injection (hereafter SRM forcing) on models should be also studied well, since estimation of the SRM forcing is important when considering the costs and benefits of geoengineering. The G1 and G2 experimental designs have limited utility in understanding sulphate aerosol geoengineering because the SRM is introduced simply and directly by the reduction of the solar constant. In G3, the amount of injected $SO_2$ mimicked in each model varies by year, which is useful for controlling the absolute amount of forcing but not the injection rate. In contrast, in G4 the rate of $SO_2$

injection is fixed at $5\ \mathrm{Tg\,yr^{-1}}$ throughout the SRM period, and the annually averaged strength of the SRM forcing should be

almost constant during the SRM period in each model, but may differ among models. Therefore, the G4 experiment is suitable for directly exploring the strength and the model dependence or uncertainty of the SRM forcing.

There are numerous sources of inter-model differences in response to the same (or similar) forcing. On processes related to the SRM forcing, modelled aerosol microphysics including formation, growth, transportation, and removal may differ, and such differences result in the difference in meridional distribution of the aerosol optical depth (AOD). Even though the prescribed AOD is given, a difference in an assumed particle size for the stratospheric sulphate aerosols causes difference in the SRM forcing (Pierce et al., 2010). On a broad scale, different models have distinct climate sensitivities and thus different global mean temperature responses to the same forcing. In addition, different models have various representations of processes, which affects the direct response to the forcing as well as different feedback from the responses. For example, cloud adjustments (Schmidt et al., 2012), sea ice changes (Moore et al., 2014), and stratospheric ozone changes (Pitari et al., 2014) are all known to affect the climate response to geoengineering through feedback. The ocean response operates on longer timescales and has also been shown to be important in understanding the response to geoengineering (Kravitz et al., 2013b). Yu et al. (2015) calculated the difference in globally and temporally averaged near-surface air temperature of G4 (over 2030–2069) from "baseline climate" (RCP4.5 over 2010–2029) and showed a standard deviation of up to $\pm 0.31$ K among models, while the model mean of the temperature difference was 0.28 K. This spread is larger than that of $\pm 0.21$ K of temperature increase in RCP4.5 scenario for the same models. Whilst the models in G4 assume the same rate of $SO_2$ injection, model responses to the SRM differ widely. Investigation into what causes such a large inter-model variability is very important for SRM simulation studies.

A simple procedure is used for quantifying the contributions of different types of SW rapid adjustments and feedbacks in the climate model behaviour to geoengineering with stratospheric sulphate aerosols. Here, a rapid adjustment is defined as a reaction to the SRM forcing without changes in globally averaged surface air temperature, whereas a feedback is defined as a reaction due to surface air temperature changes in the global mean induced by the SRM forcing (e.g., Sherwood et al., 2015). (Hereafter, the term "total reaction" refers to the sum of a rapid adjustment and a feedback.) In recent studies of the climate change, rapid adjustments are included in forcing agents and the concept of effective radiative forcing is widely used (e.g., Andrews, 2014; Zhang et al., 2016). However, for the study of the sulphate geoengineering simulation, which is not well verified by observations and thus is expected to have many uncertainties, the separation of the direct forcing and total reactions is important to improve the simulation and to enhance the degree of understanding of the sulphate geoengineering by refining individual related processes. Many studies on climate energy balance have analysed changes in the net radiation flux at TOA, where the energy budget is closed by SW and longwave radiation (LW) (e.g., Trenberth et al., 2014; Wild et al., 2014). However, in the geoengineering study, the radiative changes at the surface are also important, because vegetation, agriculture, and solar power generation for example will be strongly affected by radiative changes at the surface as well as surface temperature changes (e.g., Campillo et al., 2012). Surface SW is also important for ocean carbon cycle and fisheries through changes in amounts of phytoplankton (e.g., Miller et al., 2006). Though the surface energy budget is balanced among SW, LW, sensible heat flux, and latent heat flux, Kleidon et al. (2015) showed that the latter three are mainly determined by the air and/or surface temperature. Hence, this study focuses on changes in surface air temperature and SW. The direct SW forcing to the surface are

evaluated by considering the total reactions due to changes in water vapour amounts, cloud amounts, and surface albedo. Also, these total reactions are decomposed into adjustments and feedbacks, which indicate the rapid change just after injection of $SO_2$ and the change with globally averaged surface air temperature change by SRM, respectively. We provide results for both global and local effects, focusing on cross-model commonalities and differences. The following section describes the data and

methods used in this study. Section 3 presents the results of the analyses. Section 4 provides a short discussion. Summary and concluding remarks are provided in Section 5.

## 2   Data and methods

The models analysed in this study are listed in Table 2. Note that the method of simulating sulphate aerosols differs among the participating models. HadGEM2-ES and MIROC-ESM-CHEM-AMP calculate the formation of sulphate aerosols from

$SO_2$ injected from the lower stratosphere on the equator, and their horizontal distribution of sulphate AODs differ. BNU-ESM, MIROC-ESM, and MIROC-ESM-CHEM use a prescribed AOD, which is formulated as one fourth of the strength of the 1991 eruption of Mount Pinatubo following Sato et al. (1993) and provided in http://climate.envsci.rutgers.edu/GeoMIP/geomipaod.html. The annual cycle and latitudinal distribution of the prescribed AOD, which is zonally uniform, is shown in Fig. 1; this annual cycle is repeated every year during the SRM period. In CanESM2, a constant field of AOD ($\sim 0.047$) has

been given to express the effect of the $SO_2$ injection. The MIROC-ESM, MIROC-ESM-CHEM, and MIROC-ESM-CHEM-AMP are based on the same framework but differ in their treatment of atmospheric chemistry. An online atmospheric chemistry module is coupled in the MIROC-ESM-CHEM and MIROC-ESM-CHEM-AMP, whereas MIROC-ESM is not coupled with the chemistry module. In the MIROC-ESM-CHEM, the prescribed AOD is used for the stratospheric sulphate aerosols and for the calculation of the surface area density of the sulphur. Conversely, the MIROC-ESM-CHEM-AMP fully calculates the

chemistry and micro-physics of the stratospheric sulphate aerosol formation from $SO_2$ (a detailed description is presented in Sekiya et al., 2016).

    The mean stratospheric sulphate aerosol particle sizes and standard deviation of their log-normal distribution ($\sigma$) in each model are shown in Table 1. In HadGEM2-ES, the tropospheric aerosol scheme and the associated microphysical properties (Bellouin et al., 2011) is simply extended into the stratosphere. Modifications to the stratospheric aerosol size distribution have been applied in subsequent HadGEM2-ES studies (Jones et al., 2016a, b), but have not been applied here. In MIROC-

ESM-CHEM-AMP, the microphysics module for stratospheric sulphate aerosols treats them in three modes as shown in Table 2 in Sekiya et al. (2016); however, to calculate radiative processes on the aerosols, a particle size of 0.243 μm is assumed for simplification. Because the newly developed microphysics module for sulphate aerosols in MIROC-ESM-CHEM-AMP was not well-tested or tuned for the troposphere by a long-term climate simulation yet, it may cause unexpected drift in the

simulated climate due to changes in concentration and/or distribution of the tropospheric sulphate aerosols. To avoid such situation, the sulphate aerosol microphysics was calculated only in the stratosphere in G4 and RCP4.5.

    Note that the following five models also participated in the GeoMIP-G4 experiment but are not used in this study. GEOSCCM (Rienecker et al., 2008) and ULAQ (Pitari et al., 2002) do not include an ocean model and the sea surface temperature is

prescribed, so that the surface temperature decrease by the SRM is not simulated in a way that is conducive to the analyses undertaken. IPSL-CM5A-LR (Dufresne et al., 2013) and NorESM1-M (Bentsen et al., 2013; Iversen et al., 2013) have some issues in calculation of the LW effects of the sulphate aerosols (Ferraro and Griffiths, 2016). GISS-E2-R (Schmidt et al., 2006) has issues in its output of clear-sky SW flux at the surface that preclude the incorporation of this data in the analyses.

The model output variables used in this study are monthly means of surface air temperature ($T$), upwelling and downwelling SW fluxes at the surface and TOA for all-sky and clear-sky. The data for both experiments (RCP4.5 and G4) from the models listed in Table 2 with all ensemble members are used.

Since the SRM forcing is mainly induced by the reflection of the SW by stratospheric sulphate aerosols, the atmospheric reflection rate is very important. In order to consider rapid adjustments and feedbacks on the SW due to the SRM forcing, the
atmospheric absorption rate and the surface albedo are also important. To estimate these rates and the albedo from SW fluxes described in the previous paragraph, a single-layer atmospheric model of SW transfer used in Donohoe and Battisti (2011) (hereafter DB11) is applied. DB11's single-layer model assumes that a fraction $R$ of the downwelling solar radiation flux at the TOA $S$ is reflected back to space, and a fraction $A$ is absorbed by the atmosphere at the same single layer. A fraction $\alpha$ of the transmitted radiation flux $S(1-R-A)$ is then reflected by the surface. This reflected upwelling radiative flux is reflected
back to the surface at the rate of $R$ and absorbed at the rate of $A$ at the atmospheric layer, and the remainder $S\alpha(1-R-A)^2$ is transmitted to space. This process continues, forming an infinite geometric series, as shown in Fig. 1 of DB11; therefore, the TOA upwelling SW flux ($F_{\text{TOA}}^{\uparrow}$), surface downwelling SW flux ($F_{\text{SURF}}^{\downarrow}$), and surface upwelling SW flux ($F_{\text{SURF}}^{\uparrow}$) can be written as follows:

$$F_{\text{TOA}}^{\uparrow} = S\left[R + \alpha(1-R-A)^2 + \alpha^2 R(1-R-A)^2 + \alpha^3 R^2(1-R-A)^2 + \cdots\right]$$
$$= SR + \alpha S(1-R-A)^2\left[1 + (\alpha R) + (\alpha R)^2 + \cdots\right] = SR + \alpha S\frac{(1-R-A)^2}{1-\alpha R}, \tag{1}$$

$$F_{\text{SURF}}^{\downarrow} = S\left[(1-R-A) + \alpha R(1-R-A) + \alpha^2 R^2(1-R-A) + \alpha^3 R^3(1-R-A) + \cdots\right]$$
$$= S(1-R-A)\left[1 + (\alpha R) + (\alpha R)^2 + (\alpha R)^3 + \cdots\right] = S\frac{(1-R-A)}{1-\alpha R}, \tag{2}$$

$$F_{\text{SURF}}^{\uparrow} = \alpha F_{\text{SURF}}^{\downarrow} = \alpha S\frac{(1-R-A)}{1-\alpha R}. \tag{3}$$

Here, the infinite series in the second lines of Eqs. (1) and (2) converge to the final expression on the right-hand side because
$\alpha R < 1$. The fractions $R$, $A$, and $\alpha$ are positive and less than unity. Note that, to the best of our knowledge, the idea of forming the infinite geometric series from SW transfer between a single layer and the surface can be traced back to Rasool and Schneider (1971), who calculated the effect of aerosol on the global temperature by considering a single aerosol layer.

From Eqs. (1)–(3), $R$, $A$, and $\alpha$ can be calculated when $S$, $F_{\text{TOA}}^{\uparrow}$, $F_{\text{SURF}}^{\downarrow}$, and $F_{\text{SURF}}^{\uparrow}$ are given. Surface albedo $\alpha$ can be obtain immediately by Eq. (3) as

$$\alpha = \frac{F_{\text{SURF}}^{\uparrow}}{F_{\text{SURF}}^{\downarrow}}. \tag{4}$$

Substitution of the product of Eqs. (2) and (3) into Eq. (1) yields

$$R = \frac{SF_{\text{TOA}}^{\uparrow} - F_{\text{SURF}}^{\downarrow}F_{\text{SURF}}^{\uparrow}}{S^2 - F_{\text{SURF}}^{\uparrow 2}}, \tag{5}$$

for calculating the value of $R$. Then, $A$ is calculated using values of $R$ and $\alpha$ by the following form of Eq. (2):

$$A = (1 - R) - \frac{F_{\text{SURF}}^{\downarrow}}{S}(1 - \alpha R). \tag{6}$$

Note that, $R$, $A$, and $\alpha$ cannot be obtained when $S = 0$ such as during the polar night.

Based on the DB11's single-layer model described above, the strength of the SRM forcing and the total reactions due
to changes in the water vapour amount, cloud amount, and surface albedo are estimated using the method described in the
remainder of this section. Since GeoMIP participating models provide all-sky and clear-sky values for $F_{\text{TOA}}^{\uparrow}$, $F_{\text{SURF}}^{\downarrow}$, and
$F_{\text{SURF}}^{\uparrow}$, values of $R$, $A$, and $\alpha$ can be calculated for both all-sky and clear-sky; superscript "as" is used for all-sky and "cs" for
clear-sky. Defining the cloud effects on radiative transfer for a variable $X$ by $X^{\text{cl}} \equiv X^{\text{as}} - X^{\text{cs}}$, the all-sky value is the sum of
the clear-sky value and the cloud effect: $X^{\text{as}} = X^{\text{cs}} + X^{\text{cl}}$, where superscript "cl" is for the cloud effect. For further simplicity,
the cloud effect on the surface albedo is assumed to be negligible (i.e., $\alpha^{\text{as}} \approx \alpha^{\text{cs}}$), and $\alpha^{\text{as}}$ is used in the following analyses and
the superscript omitted. Now, the monthly mean of $R^{\text{cs}}$, $R^{\text{cl}}$, $A^{\text{cs}}$, $A^{\text{cl}}$, and $\alpha$ is calculated on each grid-point for RCP4.5 and
G4 experiments.

Net SW at the surface is a key variable in this study and can be written as follows:

$$F_{\text{SURF}}^{\text{net}} \equiv F_{\text{SURF}}^{\downarrow\text{as}} - F_{\text{SURF}}^{\uparrow\text{as}} = (1 - \alpha)S\left[\frac{1 - (R^{\text{cs}} + R^{\text{cl}}) - (A^{\text{cs}} + A^{\text{cl}})}{1 - \alpha(R^{\text{cs}} + R^{\text{cl}})}\right]. \tag{7}$$

Here, $F_{\text{SURF}}^{\text{net}}$ is regarded as a function of $S$, $R^{\text{cs}}$, $R^{\text{cl}}$, $A^{\text{cs}}$, $A^{\text{cl}}$, and $\alpha$. The difference of $F_{\text{SURF}}^{\text{net}}$ between RCP4.5 and G4
experiments is defined as

$$\Delta F_{\text{SURF}}^{\text{net}} \equiv F_{\text{SURF}}^{\text{net}}(S, R_{\text{G4}}^{\text{cs}}, R_{\text{G4}}^{\text{cl}}, A_{\text{G4}}^{\text{cs}}, A_{\text{G4}}^{\text{cl}}, \alpha_{\text{G4}}) - F_{\text{SURF}}^{\text{net}}(S, R_{\text{RCP}}^{\text{cs}}, R_{\text{RCP}}^{\text{cl}}, A_{\text{RCP}}^{\text{cs}}, A_{\text{RCP}}^{\text{cl}}, \alpha_{\text{RCP}}), \tag{8}$$

where the experiment names are indicated by subscripts "RCP" and "G4". ($S$, the TOA downwelling solar radiation, is same
for RCP4.5 and G4.) Hereafter, $F_{\text{SURF}}^{\text{net}}(\text{RCP}) \equiv F_{\text{SURF}}^{\text{net}}(S, R_{\text{RCP}}^{\text{cs}}, R_{\text{RCP}}^{\text{cl}}, A_{\text{RCP}}^{\text{cs}}, A_{\text{RCP}}^{\text{cl}}, \alpha_{\text{RCP}})$ is written for convenience.

To estimate the strength of the SRM forcing and the total reactions due to changes in the water vapour amount, cloud amount,
and surface albedo on the net SW at the surface, the following is assumed:

1. The sulphate aerosols increased by the $SO_2$ injection amplify the reflection rate of the clear-sky atmosphere ($R^{\text{cs}}$), whilst
their effect on the absorption rate ($A^{\text{cs}}$) is negligible.

2. The change in water vapour amount affects the absorption rate of the clear-sky atmosphere ($A^{\text{cs}}$), whilst its effect on the
reflection rate ($R^{\text{cs}}$) is negligible.

3. The amounts of other substances that affects the reflection or absorption rate of the clear-sky atmosphere do not change
considerably, and their effects are negligible.

Though the sulphate aerosols can absorb near infrared radiation, which is a part of SW, its effect on the SRM forcing is ignored since its amount is insignificant compared to the SW reflected by the sulphate aerosols (Haywood and Ramaswamy, 1998). (An error due to ignoring the SW absorption by the sulphate aerosols is estimated at the end of this paper.)

Under the above assumptions, the strength of the SRM forcing $F_{\mathrm{SRM}}$ is defined by

$$F_{\mathrm{SRM}} \equiv F_{\mathrm{SURF}}^{\mathrm{net}}(S, R_{\mathrm{G4}}^{\mathrm{cs}}, R_{\mathrm{RCP}}^{\mathrm{cl}}, A_{\mathrm{RCP}}^{\mathrm{cs}}, A_{\mathrm{RCP}}^{\mathrm{cl}}, \alpha_{\mathrm{RCP}}) - F_{\mathrm{SURF}}^{\mathrm{net}}(\mathrm{RCP}). \tag{9}$$

This is a change of net surface SW when only $R^{\mathrm{cs}}$ is changed to the value of G4. Similarly, the effects of total reactions from changes in the water vapour amount ($E_{\mathrm{WV}}$), cloud amount ($E_{\mathrm{C}}$), and surface albedo ($E_{\mathrm{SA}}$) are defined as follows:

$$E_{\mathrm{WV}} \equiv F_{\mathrm{SURF}}^{\mathrm{net}}(S, R_{\mathrm{RCP}}^{\mathrm{cs}}, R_{\mathrm{RCP}}^{\mathrm{cl}}, A_{\mathrm{G4}}^{\mathrm{cs}}, A_{\mathrm{RCP}}^{\mathrm{cl}}, \alpha_{\mathrm{RCP}}) - F_{\mathrm{SURF}}^{\mathrm{net}}(\mathrm{RCP}), \tag{10}$$

$$E_{\mathrm{C}} \equiv F_{\mathrm{SURF}}^{\mathrm{net}}(S, R_{\mathrm{RCP}}^{\mathrm{cs}}, R_{\mathrm{G4}}^{\mathrm{cl}}, A_{\mathrm{RCP}}^{\mathrm{cs}}, A_{\mathrm{G4}}^{\mathrm{cl}}, \alpha_{\mathrm{RCP}}) - F_{\mathrm{SURF}}^{\mathrm{net}}(\mathrm{RCP}), \tag{11}$$

$$E_{\mathrm{SA}} \equiv F_{\mathrm{SURF}}^{\mathrm{net}}(S, R_{\mathrm{RCP}}^{\mathrm{cs}}, R_{\mathrm{RCP}}^{\mathrm{cl}}, A_{\mathrm{RCP}}^{\mathrm{cs}}, A_{\mathrm{RCP}}^{\mathrm{cl}}, \alpha_{\mathrm{G4}}) - F_{\mathrm{SURF}}^{\mathrm{net}}(\mathrm{RCP}). \tag{12}$$

Here, the following three points should be noted. First, $E_{\mathrm{WV}}$, $E_{\mathrm{C}}$, and $E_{\mathrm{SA}}$ are measures for the sum of SW radiative rapid adjustment and feedback, and do not include any LW effects; changes in the water vapour and cloud amounts can, however, affect LW transfer. Second, the sum of $F_{\mathrm{SRM}}$, $E_{\mathrm{WV}}$, $E_{\mathrm{C}}$, and $E_{\mathrm{SA}}$ is not exactly equal to $\Delta F_{\mathrm{SURF}}^{\mathrm{net}}$, since Eq. (7) is not linear.
However, if $\Delta F_{\mathrm{SURF}}^{\mathrm{net}} \approx F_{\mathrm{SRM}} + E_{\mathrm{WV}} + E_{\mathrm{C}} + E_{\mathrm{SA}}$ is satisfied, it can be stated that the decomposition of $\Delta F_{\mathrm{SURF}}^{\mathrm{net}}$ into $F_{\mathrm{SRM}}$, $E_{\mathrm{WV}}$, $E_{\mathrm{C}}$, and $E_{\mathrm{SA}}$ is reasonable. Finally, $E_{\mathrm{C}}$ includes both the effect of changes in cloud cover and cloud albedo. This is because $R^{\mathrm{cl}}$ and $A^{\mathrm{cl}}$ can be written as follows, by expressing $R^{\mathrm{as}}$ and $A^{\mathrm{as}}$ with the total cloud-area fraction $\gamma$, the reflection rate of a fully cloud-covered atmosphere $r^{\mathrm{fca}}$, and the absorption rate of a fully cloud-covered atmosphere $a^{\mathrm{fca}}$,

$$R^{\mathrm{cl}} = R^{\mathrm{as}} - R^{\mathrm{cs}} = (1 - \gamma)R^{\mathrm{cs}} + \gamma r^{\mathrm{fca}} - R^{\mathrm{cs}} = \gamma(r^{\mathrm{fca}} - R^{\mathrm{cs}}), \tag{13}$$

$$A^{\mathrm{cl}} = A^{\mathrm{as}} - A^{\mathrm{cs}} = (1 - \gamma)A^{\mathrm{cs}} + \gamma a^{\mathrm{fca}} - A^{\mathrm{cs}} = \gamma(a^{\mathrm{fca}} - A^{\mathrm{cs}}). \tag{14}$$

These expressions mean that cloud effects ($R^{\mathrm{cl}}$ and $A^{\mathrm{cl}}$) include both the total cloud-area fraction and reflection or absorption
rate of a fully cloud-covered atmosphere, which depends on cloud albedo or absorption rate. Therefore, $E_{\mathrm{C}}$ includes both the effect of changes in coverage, albedo and SW absorption rate of clouds. In addition, $E_{\mathrm{C}}$ should not include the "masking effect" (Zhang et al., 1994; Colman, 2003; Soden et al., 2004) of the clouds because the clear-sky values $R^{\mathrm{cs}}$ and $A^{\mathrm{cs}}$ are unchanged from those in RCP4.5.

In this study, the SRM forcing and the three total reactions on net SW at the surface from the changes in the water vapour
amount, cloud amount, and surface albedo, defined by Eqs. (9)–(12), are calculated on each grid-point where $S > 0$ from the monthly mean data. At grid points where $S = 0$, $F_{\mathrm{SRM}} = E_{\mathrm{WV}} = E_{\mathrm{C}} = E_{\mathrm{SA}} = 0$.

To decompose the total reactions ($E_{\mathrm{WV}}$, $E_{\mathrm{C}}$, and $E_{\mathrm{SA}}$) into rapid adjustments and feedbacks, a method similar to the Gregory plot (Gregory et al., 2004) is used. That is, the globally and annually averaged data of total reactions are plotted against that of

$\Delta T$ ($\equiv T_{\text{G4}} - T_{\text{RCP}}$), and linear regression lines in the following forms are obtained by the least squares method.

$$\overline{E_{\text{WV}}} = Q_{\text{WV}} - P_{\text{WV}}\overline{\Delta T}, \tag{15}$$

$$\overline{E_{\text{C}}} = Q_{\text{C}} - P_{\text{C}}\overline{\Delta T}, \tag{16}$$

$$\overline{E_{\text{SA}}} = Q_{\text{SA}} - P_{\text{SA}}\overline{\Delta T}. \tag{17}$$

Here, $Q_X$ ($X =$WV, C, SA) denotes the rapid adjustment, $-P_X$ is the feedback parameter, and the overline denotes the global and annual average. This method is similar to the Gregory plot, but note that $\Delta T$ is the surface temperature difference between the G4 experiment and the RCP4.5 scenario experiment in which the anthropogenic radiative forcing depends on time and the simulated climate does not reach an statistically equilibrium state.

## 3 Results

### 3.1 Surface air temperature and shortwave radiation

Figure 2 shows the time series of globally averaged surface air temperature ($T$) with a 12-month running mean for G4 and RCP4.5. For all models, $T$ in G4 decreases or remains within $+0.3$ K from the baseline for a few decades and begins increasing from around 2040 or earlier, whereas $T$ in RCP4.5 steadily increases. Accordingly, the difference in $T$ between RCP4.5 and G4 increases for 10–25 years from 2020 and then stops rising. That is, the cooling effect of SRM gradually affects the global mean of $T$ because of slow feedback and/or thermal inertia of the modelled climate system, and takes a few decades to reach steady state. After that, the SRM becomes unable to prevent the temperature from increasing any more, delaying global warming for a few decades as compared with RCP4.5. This is simply because the anthropogenic forcing in RCP4.5 keeps increasing but the amount of $SO_2$ injection per year is fixed in G4. In addition, after halting SRM at the end of 2069, $T$ increases rapidly and then returns to or approaches the RCP4.5 level in each model. This rapid increase has been called the termination effect of SRM (e.g., Wigley, 2006; Jones et al., 2013; Kravitz et al., 2013d).

To properly compare the SRM effects among the models, we eliminate some of the transient behaviour and focus on the years 2040 to 2069, in which the amount of cooling in G4 compared with RCP4.5 is roughly kept constant. (Although the reason for the transient behaviour of the SRM's cooling effect is an important topic, it is beyond the scope of this study.) Figure 3 shows the relationship between $\Delta T$ and $\Delta F_{\text{SURF}}^{\text{net}}$, the difference in net SW at the surface, averaged over the globe, for 2040–2069. This figure shows a strong correlation between the mean $\Delta T$ and $\Delta F_{\text{SURF}}^{\text{net}}$; the correlation coefficient for the six filled symbols is 0.88. This strong correlation allows $\Delta F_{\text{SURF}}^{\text{net}}$ to be used as a measure of the SRM effects at least for $-1.1 \lesssim \Delta T \lesssim -0.2$ K, although the surface air temperature depends on the energy balance among SW, LW, and sensible and latent heat fluxes at the surface. Moreover, as described at the end of Section 1, it is important to explore the SW flux at the surface to estimate the effect of SRM on vegetation and human activities such as agriculture and solar power generation. Therefore, this study mainly focuses on SW at the surface and estimates the SRM forcing and total reactions of SW due to changes in the water vapour amount, cloud amount, and surface albedo. One concern is that half the models used in this study

have only one ensemble member, and half are MIROC-based models. The effects of this are analysed in Section 4.2 and shown to be relatively unimportant.

## 3.2 Time-evolution of global mean forcing and SW total reactions

The strength of the SRM forcing ($F_{\text{SRM}}$) defined by Eq. (9) and the SW total reactions due to changes in the water vapour amount ($E_{\text{WV}}$), cloud amount ($E_{\text{C}}$), and surface albedo ($E_{\text{SA}}$) defined by Eqs. (10)–(12) are calculated for each model. Figure 4 shows the time-evolution of the globally averaged values of these measures with a 12-month running mean. $\Delta F_{\text{SURF}}^{\text{net}}$ and $\Delta T$ are also shown in this figure. In this subsection, the focus is on the qualitative features common to all or some of the models, whilst the quantitative differences are described in the following subsection.

In the models that used the prescribed or constant AOD field for the SRM (BNU-ESM, CanESM2, MIROC-ESM, and MIROC-ESM-CHEM), $F_{\text{SRM}}$ immediately reaches a model-dependent negative value after 2020 and remains almost constant; it then vanishes instantly after the termination. These features are consistent with the fact that the given AOD for the SRM was instantly added and removed in these models. Conversely, in the models that calculate the formation and transport of the sulphate aerosols from the injected $SO_2$ (HadGEM2-ES and MIROC-ESM-CHEM-AMP), $F_{\text{SRM}}$ takes approximately four years to become saturated. During the period in which SRM is imposed, $F_{\text{SRM}}$ in MIROC-ESM-CHEM-AMP is almost constant, but $F_{\text{SRM}}$ in HadGEM2-ES varies by approximately $1.0 \text{ W m}^{-2}$.

The values of $E_{\text{SA}}$ are both negative and small in all of the models. This shows that, at least for the global average, the surface albedo under G4 is higher than that under RCP4.5. However, changes in the surface albedo do not significantly affect $\Delta F_{\text{SURF}}^{\text{net}}$.

Both $E_{\text{WV}}$ and $E_{\text{C}}$ are positive, implying that the decreases in water vapour and cloud amounts under SRM lead to more downwelling SW at the surface, counteracting the enhanced aerosol reflection by SRM. One reason for the decrease of water vapour is the temperature reduction, which results in less evaporation (Kravitz et al., 2013c). Less water vapour may cause reduced cloud amounts; less water vapour and reduced cloud amounts increase the atmospheric SW transmissivity and reduce the SRM's cooling effect. The strengths of $E_{\text{WV}}$ and $E_{\text{C}}$ are comparable in each model except MIROC-ESM-CHEM-AMP (a reason for this exception is discussed in the next subsection). After SRM termination, $E_{\text{WV}}$ remains positive for one or two decades. This is consistent with changes in $\Delta T$; i.e., the water vapour amount in G4 remains less than that in RCP4.5 for a while after the termination. The inter-annual variability of $E_{\text{C}}$ is much larger than that of $E_{\text{WV}}$, and the gradual transition to the state of RCP4.5 after the termination (like $E_{\text{WV}}$) is not apparent. Through the whole simulation period, the inter-annual variability of $E_{\text{C}}$ dominates that of $\Delta F_{\text{SURF}}^{\text{net}}$. It should be noted that the phases in wave-like, year-to-year variability of $\Delta F_{\text{SURF}}^{\text{net}}$ and $\Delta T$ do not agree, although time-averaged $\Delta F_{\text{SURF}}^{\text{net}}$ is well correlated with $\Delta T$ as shown in Fig. 3. This is because of thermal inertia and nonlinearities in the Earth system.

## 3.3 Inter-model dispersion of global mean forcing and SW total reactions

For the inter-model comparison of the results, the global means of $F_{\text{SRM}}$, $E_{\text{WV}}$, $E_{\text{C}}$, and $E_{\text{SA}}$ are averaged over the period 2040–2069. Figure 5 shows the relationship between these values and $\Delta T$ in the same manner as Fig. 3; $\Delta F_{\text{SURF}}^{\text{net}}$ is shown

again. The mean values of $F_{\mathrm{SRM}}$ vary widely from approximately $-3.6$ to $-1.6 \,\mathrm{W\,m^{-2}}$, depending on the model. The cooling effect of $F_{\mathrm{SRM}}$ in each member or the ensemble mean is reduced by $E_{\mathrm{WV}}$ and $E_{\mathrm{C}}$ and is slightly increased by $E_{\mathrm{SA}}$. The net effect is approximately equal to $\Delta F_{\mathrm{SURF}}^{\mathrm{net}}$, which is strongly correlated with $\Delta T$; the residual is less than $0.06 \,\mathrm{W\,m^{-2}}$. This supports the validity of the decomposition of $\Delta F_{\mathrm{SURF}}^{\mathrm{net}}$ into SRM forcing and the total reactions due to changes in the water vapour amount, cloud amount, and surface albedo.

The two models with sulphate aerosol calculation (HadGEM2-ES and MIROC-ESM-CHEM-AMP) show stronger $F_{\mathrm{SRM}}$ than the others. This outcome indicates that the prescribed AOD, which is based on one-fourth of the Mount Pinatubo eruption, likely underestimates the AOD that results from actual $SO_2$ injection at a rate of $5 \,\mathrm{Tg\,yr^{-1}}$. It is the difference in the mean AOD rather than its meridional distribution as shown in Fig. S1 that leads to the underestimation of the AOD in G4. The globally and temporally averaged stratospheric sulphate AOD in MIROC-ESM-CHEM-AMP is 0.083 and that in HadGEM2-ES is approximately 0.054, though that of the prescribed AOD is 0.037. Note that the above value for HadGEM2-ES is the difference (G4 − RCP4.5) in the sulphate AOD for both troposphere and stratosphere. This is because HadGEM2-ES used the same microphysics calculation of the sulphate aerosols with the same aerosol size distribution in both the troposphere and the stratosphere; sulphate AOD solely for the stratosphere is not available for HadGEM2-ES.

In CanESM2 and MIROC-ESM-CHEM, the $F_{\mathrm{SRM}}$ values are very similar among the ensemble members. This is consistent with the fact that the given AOD fields for mimicking the $SO_2$ injection effects in G4 are identical among ensemble members of each model. On the other hand, the values of $F_{\mathrm{SRM}}$ in the ensemble members of HadGEM2-ES have considerable differences, because the distribution of the sulphate AOD is affected by the chaotic nature of transport and various other processes in the ESM. Even after averaging over 30 years, the mean seasonal cycles of the sulphate AOD can differ among the ensemble members as shown in Fig. S1.

Pitari et al. (2014) have shown that, in the G4 simulation, SW radiative forcing at the tropopause calculated off-line by a radiative transfer code (Chou and Suarez, 1999; Chou et al., 2001) varies from around $-2.1$ to $-1.0 \,\mathrm{W\,m^{-2}}$ between the models. Since both the analysis methods and the participating models presented here differ from those of Pitari et al., it is difficult to compare the two results. However, the results ($F_{\mathrm{SRM}} \sim -3.6$ to $-1.6 \,\mathrm{W\,m^{-2}}$) show that model dependence of the SRM forcing might be larger than that shown by Pitari et al.

Figure 5 shows that $E_{\mathrm{WV}}$ is strongly anti-correlated with $\Delta T$; the correlation coefficient for the filled symbols is $-0.94$. In contrast, $E_{\mathrm{C}}$ seems to have no correlation with $\Delta T$, with a correlation coefficient of 0.01. This result shows that the SW total reaction from the change in water vapour amount is much simpler (i.e., almost linear with $\Delta T$ across all models) than that from changing the cloud amount, which depends strongly on the cloud parameterization scheme. Furthermore, the results of the ensemble members of CanESM2 and MIROC-ESM-CHEM show that the variation in $E_{\mathrm{C}}$ mainly causes the variation in $\Delta F_{\mathrm{SURF}}^{\mathrm{net}}$, which is well correlated with $\Delta T$, though $F_{\mathrm{SRM}}$ is same among the members. Thus, among the ensemble members, higher $E_{\mathrm{C}}$ seems to bring less cooling. MIROC-ESM-CHEM-AMP marks the strongest forcing of the SRM among the models but also marks the largest increase of SW from changing the cloud amount. Accordingly, this model shows the moderate values in $\Delta F_{\mathrm{SURF}}^{\mathrm{net}}$ and $\Delta T$; a possible explanation is given in the following analysis.

To compare ratios of the total reaction and the surface cooling to the magnitude of the SRM forcing, $E_{WV}$, $E_C$, $E_{SA}$, and $\Delta T$ by $|F_{SRM}|$ are normalized, as shown in Fig. 6. This figure shows the approximate sensitivity of each total reaction per unit forcing of SRM and the normalized surface cooling. The value range of $E_C/|F_{SRM}|$ (0.19–0.55) is significantly wider than that of $E_{WV}/|F_{SRM}|$ (0.27–0.42) and that of $E_{SA}/|F_{SRM}|$ (−0.12 to −0.06). In addition, the three MIROC-based models show higher $E_C/|F_{SRM}|$ (0.34–0.55) than other three models (0.19–0.34). This means that the sensitivity of the total reaction due to change in cloud amount in the MIROC-based models is higher than other models. This may be why MIROC-ESM-CHEM-AMP, whose $E_C/|F_{SRM}|$ is as high as those of MIROC-ESM and MIROC-ESM-CHEM, exhibits high $E_C$ and yields moderate cooling, although $F_{SRM}$ is very strong, as shown in Fig. 5. That is, high sensitivity of $E_C$ to the SRM forcing will weaken the cooling of surface air temperature as well as $\Delta F_{SURF}^{net}$.

The wide variability of $E_C/|F_{SRM}|$ among the models implies a large uncertainty in the models' cloud processes. Moreover, the spread of $E_C/|F_{SRM}|$ among nine ensemble members of MIROC-ESM-CHEM is also large. The variability among the ensemble members implies that the cloud amount is considerably affected by the chaotic properties and high sensitivity to the initial state of the Earth system or ESM, because any model settings other than the initial state are the same among the ensemble members. This result therefore suggests that the cooling of the surface air temperature by the SRM depends significantly on the initial state through total reaction due to changes in the cloud amount.

### 3.4 Decomposition of total reaction into rapid adjustment and feedback

The total reactions due to changes in water vapour amounts, cloud amounts, and surface albedo discussed in the previous two subsections are the sums of the rapid adjustment, which are independent of $\Delta T$, and the feedback, which depends linearly on $\Delta T$. In this subsection, we attempt to decompose the rapid adjustment and the feedback using a so-called Gregory plot (Gregory et al., 2004). Figure 7 shows globally and annually averaged $E_{WV}$, $E_C$, and $E_{SA}$ as a function of averaged $\Delta T$ for each model. Now, we consider that a slope and a y-intercept show a feedback parameter and an amount of rapid adjustment, respectively, as shown by Eqs. (15)–(17); these values and correlation coefficients are shown in Table 3. The multi-model mean values are also shown.

$E_{WV}$ shows high negative correlation with $\Delta T$ in all models, and the rapid adjustment (+0.30 $\mathrm{Wm^{-2}}$ in multi-model mean) and the feedback (−0.91 $\mathrm{Wm^{-2}K^{-1}}$) are clearly separated. That is, the surface SW increase due to less water vapour is caused by both the rapid direct response to SRM and the surface cooling; note that the negative sign corresponds to an increase in surface SW with cooling. The rapid decrease of the water vapour would result from reduced convection due to change in vertical temperature profile caused by the injected stratospheric sulphate aerosols.

Unlike $E_{WV}$, $E_C$ is not well-correlated with $\Delta T$. In addition, the spread of $E_C$ is large. This means that the rapid adjustment due to cloud changes varies largely, depending on the simulated state of ESM. The feedback of SW cloud radiative effect is not dominant in G4 experiment. Such positive and large rapid adjustment due to the cloud changes and the small cloud feedback are consistent with Kravitz et al. (2013c), who analysed the GeoMIP-G1 experiment.

The y-intercept of $E_{\text{SA}}$ is almost zero, so that the rapid adjustment from the surface albedo change is negligible. The feedback parameter is $0.38\ \text{Wm}^{-2}\text{K}^{-1}$ in the multi-model mean, and the strength (absolute value) of the feedback is less than a half of that of $E_{\text{WV}}$.

### 3.5 Robust features in geographical distribution

To explore robust features in the effects of the SRM in G4, the multi-model mean of the surface air temperature and net SW at the surface is calculated. Figures 8a and 8b show $\Delta T$ and $\Delta F_{\text{SURF}}^{\text{net}}$ averaged over the period 2040–2069. The zonal means are shown in the right-hand side panel for each variable. Here, model grid intervals are equal to or narrower than 2.8125 deg, so that the geographical regions mentioned below are represented by enough grid points. However, properties of the Sea of Okhotsk and Hudson Bay may depend on related channels, which may be not well resolved. The geographical distribution of the multi-model mean shows that cooling of the surface air temperature is very strong in and around the Arctic Region, except for Greenland and Europe, and stronger on land than over the ocean in other regions. Such features agree with previous studies such as Robock et al. (2008). Reduction of $F_{\text{SURF}}^{\text{net}}$ is strong in the eastern part of Southern Africa, Tibet, East Asia, Sea of Okhotsk, Hudson Bay, and South America. In contrast, $F_{\text{SURF}}^{\text{net}}$ increased compared with RCP4.5 in the equatorial region of the Western Pacific, Southern Ocean, except near the Antarctic coast and northern part of the Atlantic. The above reduction and increase are mainly due to $E_{\text{C}}$ and $E_{\text{SA}}$; details will be discussed later in this section. The spatial distribution of the sign of $\Delta F_{\text{SURF}}^{\text{net}}$ varies, whereas $\Delta T$ is negative over the whole globe. Although $\Delta T$ and $\Delta F_{\text{SURF}}^{\text{net}}$ are correlated in the global mean (Fig. 3), the spatial distribution of $\Delta T$ does not necessarily need to agree with that of $\Delta F_{\text{SURF}}^{\text{net}}$ because circulation and hydrological processes transport and redistribute energy.

Qualitatively opposite geographical features in $\Delta T$ and $\Delta F_{\text{SURF}}^{\text{net}}$ appear in the simulated climate change in RCP4.5 shown in Figs. 8c and 8d, calculated as the difference between the 2010–2039 average and the 2040–2069 average of the RCP4.5 data. Note that the very high positive value in East Asia in Fig. 8d is due to a large reduction of anthropogenic aerosol emission assumed in the late 21st century in the RCP4.5 scenario (Thomson et al., 2011; Westervelt et al., 2015). With the exception of the effects of such assumed emission reduction, sulphate geoengineering can delay global warming almost without regional biases; that is, regions where surface air temperature increases are relatively high in RCP4.5 undergo a large amount of cooling by the sulphate geoengineering and regions with a relatively low increases in temperature receive a small amount of cooling. Model dependence in $\Delta T$ shown by coloured lines in Fig. 8a is relatively large in high latitudes in the Northern Hemisphere but small (i.e., comparable with the spread of the global mean $\Delta T$) in other regions. For $\Delta F_{\text{SURF}}^{\text{net}}$ shown in Fig. 8b, all models show qualitatively similar average features at least in the zonal mean, and the range is about $\pm 0.75\ \text{W m}^{-2}$.

Next, the multi-model mean of global distributions (averaged over 2040–2069) of (a) $F_{\text{SRM}}$, (b) $E_{\text{WV}}$, (c) $E_{\text{C}}$, and (d) $E_{\text{SA}}$ are calculated, as shown in Fig. 9. The SRM forcing is relatively weak in the regions where the annual mean surface albedo is high, such as Greenland, the Sahara, the Middle East, Australia, and Antarctica. This is mainly because the net SW at the surface is low due to the high surface albedo, and accordingly the absolute value of the SRM forcing becomes low. This can be shown via low order approximation: the net SW at the surface can be written as $F_{\text{SURF}}^{\text{net}} \approx (1-\alpha)S(1-R-A)$, and the SRM forcing can be approximated as $F_{\text{SRM}} \approx -(1-\alpha_{\text{RCP}})S(R_{\text{G4}}^{\text{cs}}-R_{\text{RCP}}^{\text{cs}})$, whose absolute value becomes small when $\alpha_{\text{RCP}}$ is high.

Except for these high surface-albedo regions, the spatial variation in SRM forcing is not very large, even though the incoming solar radiation is strong at low latitudes and weak at high latitudes. This is because the atmospheric reflection rate depends on the solar zenith angle, and the reflection rate becomes higher as the zenith angle increases (e.g., Joseph et al., 1976). That is, strong solar radiation at low latitudes is reflected with low efficiency and weak solar radiation at the high latitudes is reflected

with high efficiency. Accordingly, the latitudinal distribution of the SRM forcing is close to uniform in many models. The above feature is a notable aspect in sulphate geoengineering compared with idealized SRM experiments such as G1 and G2, in which the solar constant is simply reduced (Kravitz et al., 2013a) and the forcing is proportional to the cosine of latitude. Latitudinal distribution of $F_{\mathrm{SRM}}$ in HadGEM2-ES and MIROC-ESM-CHEM-AMP shows a stronger latitudinal dependence. These results are consistent with the (approximate) distribution of the stratospheric sulphate AOD as shown in Fig. S1.

The SW total reaction due to the change in the water vapour amount (Fig. 9b) is close to uniform compared with that of the cloud amount (Fig. 9c). The slight increase of $E_{\mathrm{WV}}$, which implies less water vapour, in the equatorial region is consistent of decrease of precipitation reported by Rasch et al. (2008a) and Robock et al. (2008) under SRM. $E_{\mathrm{C}}$ has a large spatial variability, which yields many of the spatial variation of $\Delta F_{\mathrm{SURF}}^{\mathrm{net}}$, such as positive values in the equatorial region of the Western Pacific, the Southern Ocean, and the northern part of the Atlantic, and negative values in the eastern part of the Southern Africa, East Asia,

and South America. Because $\Delta F_{\mathrm{SURF}}^{\mathrm{net}}$ (Fig. 8b) and the simulated climate change of $F_{\mathrm{SURF}}^{\mathrm{net}}$ in RCP4.5 (Fig. 8d) are opposite in sign, the above result suggests that the SRM offsets increases in the cloud amount simulated in the RCP4.5 scenario, in the positive regions in Fig. 9c and vice versa in the negative regions. The remaining features in $\Delta F_{\mathrm{SURF}}^{\mathrm{net}}$ are caused by the effect of surface albedo change (Fig. 9d), which has large negative values in Tibet, the Sea of Okhotsk, Hudson Bay, and the Southern Ocean near the Antarctic coast. That is, snow and sea ice remain in these regions in the G4 experiment because of the SRM.

At high latitudes, the decrease of the net SW at the surface by the change in surface albedo is as large as the SW increase by the change in cloud amount (see the line graph in panels c and d), although $E_{\mathrm{SA}}$ is minor in the global mean.

## 4   Discussion

### 4.1   Difference between the surface and TOA

This study has focused on the surface net SW because of its importance to human activities. However, the situation at TOA is

also of interest, because the energy budget of the Earth system is closed at TOA. Now, we discuss how the measures used in this study differ when TOA is used for the analysis. The net SW at TOA can be written as

$$F_{\mathrm{TOA}}^{\mathrm{net}} \equiv S - F_{\mathrm{TOA}}^{\uparrow \mathrm{as}} = S \left\{ 1 - (R^{\mathrm{cs}} + R^{\mathrm{cl}}) - \alpha \frac{\left[ 1 - (R^{\mathrm{cs}} + R^{\mathrm{cl}}) - (A^{\mathrm{cs}} + A^{\mathrm{cl}}) \right]^2}{1 - \alpha (R^{\mathrm{cs}} + R^{\mathrm{cl}})} \right\}, \qquad (18)$$

so that the direct forcing of SRM and the total reactions measured at TOA ($F_{\mathrm{SRM}}^{\mathrm{TOA}}$, $E_{\mathrm{WV}}^{\mathrm{TOA}}$, $E_{\mathrm{C}}^{\mathrm{TOA}}$, and $E_{\mathrm{SA}}^{\mathrm{TOA}}$) can be calculated in the same manner described in Section 2. Figure 10 shows their globally and temporally averaged values' dependencies on $\Delta T$. The difference of $F_{\mathrm{TOA}}^{\mathrm{net}}$ is also plotted.

The qualitative features of the measures other than $E_{\text{WV}}^{\text{TOA}}$ are same as the analysis at the surface shown in Fig. 6. The quantitative difference in the SRM forcing ($F_{\text{SRM}}^{\text{TOA}} - F_{\text{SRM}}$) is as small as $-0.047$ Wm$^{-2}$ (1.8 %) for the multi-model mean. In contrast, $|E_{\text{SA}}^{\text{TOA}}|$ is less than that of $|E_{\text{SA}}|$ by about 35 %. This is mainly because the upward shortwave radiation that was reflected at the surface must pass the atmosphere being decreased by absorption and reflection before reaching the TOA. The ratio $E_{\text{SA}}^{\text{TOA}}/E_{\text{SA}}$, of course, agrees with $(1 - R_{\text{RCP}}^{\text{as}} - A_{\text{RCP}}^{\text{as}})/(1 - R_{\text{RCP}}^{\text{as}})$, which can be obtained through algebraic manipulation. The difference of $E_{\text{C}}^{\text{TOA}} - E_{\text{C}}$ is 0.12 Wm$^{-2}$ (16.5 %) for the multi-model mean. Remember that the effect of the cloud amount change includes both changes in reflection rate ($R^{\text{cl}}$) and absorption rate ($A^{\text{cl}}$). The effect of a change in $R^{\text{cl}}$ should appear almost equally at the surface and TOA, as the case for the SRM forcing, because both $R^{\text{cl}}$ and $R^{\text{cs}}$ appear in the Eqs. (7) and (18) in the same way. Therefore, most of $E_{\text{C}}^{\text{TOA}} - E_{\text{C}}$ should be caused by the difference in how the change of the absorption rate affects the net SW at the surface and that at TOA. This is discussed below.

The total reaction at TOA due to the change in water vapour amount shows a negative sign, which is opposite to that at the surface. This disagreement is attributed as follows: Surface cooling reduces the amount of water vapour in the atmosphere and the SW absorption rate decreases. Then, more incoming solar radiation reaches the surface, so that the decrease in water vapour amount increases SW flux at the surface. On the other hand, when the SW absorption rate decreases, the more upwelling SW that was reflected at the surface pass through the atmosphere and reaches TOA. This leads to a cooling effect. Because the effect of decrease in the SW absorption rate is carried to TOA by the upwelling SW that was reflected at the surface by the rate of $\alpha$, $|E_{\text{WV}}^{\text{TOA}}|$ is much less than $|E_{\text{WV}}|$. Note that, in our single layer model, SW absorption above the clouds is not included, so that upwelling SW at TOA reflected by the clouds without reaching the surface is independent of the absorption rate. Therefore, $E_{\text{WV}}^{\text{TOA}}$ could be underestimated, and the change in water vapour may not be negligible for the energy budget at TOA. Furthermore, we have not explored LW in this study. An analysis on LW rapid adjustment and feedbacks due to changes in water vapour and clouds is left as our future work.

From the above discussion, we have found that the effect of changes in atmospheric SW absorption rate appears differently between at the surface and at TOA (in its sign and amount), but that in reflection rate appears almost equally. The effect of change in the surface albedo is weaker at TOA than at the surface. We will bear these properties in our mind, when we discuss the influence of SRM on the energy budget of the climate system, which is usually considered at TOA, and human activities, which are mainly performed at the surface.

To fairly compare feedback parameters in G4 with those under greenhouse gas forcing, we decompose the total reactions at TOA into rapid adjustment and feedback in the same manner that we performed in Section 3.4. The rapid adjustment and feedback parameters calculated at TOA are listed in Table S1. The multi-model-averaged feedback parameter of surface albedo in G4 is 0.27 Wm$^{-2}$K$^{-1}$. This value is close to the surface albedo feedback parameter of 0.26 Wm$^{-2}$K$^{-1}$ in A1B scenario (Soden and Held, 2006) and that of 0.30 Wm$^{-2}$K$^{-1}$ in the quadrupled $CO_2$ experiment (Donohoe et al., 2014). On the other hand, the multi-model-averaged feedback parameter of water vapour in G4 is 0.15 Wm$^{-2}$K$^{-1}$ and that (for SW at TOA) in quadrupled $CO_2$ experiment is 0.30 Wm$^{-2}$K$^{-1}$. These comparisons suggest that the SW feedback of surface albedo under sulphate geoengineering is consistent with that under greenhouse gas forcing, whereas that of water vapour is about a half of that under greenhouse gas forcing. The difference in the water vapour feedback would be due to differences in the vertical

temperature profile and/or the atmospheric circulation under sulphate geoengineering and those under greenhouse gas forcing (e.g., McCusker et al., 2015). In contrast, the surface albedo feedback would not depend on such atmospheric features and would be dominated by changes of sea ice and snow, which mainly depend on the surface air temperature change.

## 4.2 Inequality in the number of ensemble and participating models

One concern in this study is that half of the models used have only one ensemble member, and half are MIROC-based models. Because the numbers of ensemble members differ among models as listed in Table 2, each member in each model is not equally weighted in calculation of the multi-model means described in Section 3.5. Responses to the SRM forcing in the three MIROC-based models should be similar to each other as shown in Fig. 6, so that the results of multi-model mean can be biased to that of the MIROC-based models. Therefore, we re-calculated multi-model means by using only one run for each model (Fig. S2); and also tested multi-model means with a weight of 1/3 for the MIROC-based models (Fig. S3). There are no significant difference among Figs. 9, S2, and S3. Therefore, inequality in the number of ensemble and participating models has no significant effects on our results.

## 5 Summary and concluding remarks

The results from six models (listed in Table 2) that simulated GeoMIP experiment G4, which is designed to simulate sulphate geoengineering by injecting 5 Tg of $SO_2$ into the stratosphere every year from 2020 to 2069 in the RCP4.5 scenario as the baseline, have been analysed. A single-layer model proposed by Donohoe and Battisti (2011) has been applied to estimate the strength and its inter-model variability of the SRM forcing ($F_{SRM}$) to the surface net shortwave radiation, whose difference between G4 and RCP4.5 ($\Delta F_{SURF}^{net}$) has a strong correlation with the cooling of the surface air temperature ($\Delta T$), as shown in Fig. 3. The SW total reactions due to changes in the water vapour amount ($E_{WV}$), cloud amount ($E_C$), and surface albedo ($E_{SA}$) have been also estimated. Here, a total reaction is defined as a sum of a rapid adjustment, which does not depend on $\Delta T$, and a feedback, which is proportional to $\Delta T$. Decomposition of the estimated total reactions into the rapid adjustment and the feedback is also done by using a method based on the Gregory plot (Gregory et al., 2004). Note that, unlike the usual Gregory plot, $\Delta T$ is defined by the difference between $T_{G4}$ and $T_{RCP}$, and both experiments are not approaching a statistically equilibrium state, so that the rapid response could vary depending on the state of the modelled climate system.

It has been shown that the globally and temporally averaged $F_{SRM}$ of each model varies widely from about $-3.6$ to $-1.6$ $W\,m^{-2}$. Inter-model variations comprise a substantial range, and narrowing this uncertainty is essential for understanding the effects of sulphate geoengineering and its interactions with chemical, micro-physical, dynamical, and radiative processes related to the formation, distribution, and shortwave-reflectance of the sulphate aerosols introduced from the $SO_2$ injection (Rasch et al., 2008b; Kremser et al., 2016). From a point of view of an environmental assessment of sulphate geoengineering, we note that there is such large uncertainty in the simulated SRM forcing.

Our analysis has also shown that, in the global average, changes in the water vapour and cloud amounts (from RCP4.5) increase the SW at the surface and reduce the effect of $F_{SRM}$ by approximately 0.4–1.2 $W\,m^{-2}$ and 0.5–1.5 $W\,m^{-2}$, respec-

tively. This is due to the smaller amounts of water vapour and clouds, which mainly block the downwelling solar radiation from reaching the surface by absorption and reflection, respectively. $E_{WV}$ is well correlated with $\Delta T$ in multi-model comparison, whereas $E_C$ is not. The reduction rate of $E_C$ varies from 19 % to 55 % as compared to $F_{SRM}$ depending on both models and ensemble runs (i.e., initial states), whereas that of $E_{WV}$ is 27–42 %. The effect of surface albedo changes is small in the global average, but is significant in the regions where snow or ice melts in the RCP4.5 scenario.

The decomposition analysis has revealed that about 37 % (multi-model mean) of $E_{WV}$ is explained by the rapid adjustment and the rest is the feedback. On the other hand, almost all of $E_C$ consists of the rapid adjustment, and a linear relationship between $E_C$ and $\Delta T$ for the global and annual mean was not obtained for any models. The cloud rapid adjustment in G4 deduced in this study is similar as found for G1 by Kravitz et al. (2013c) but disagree with that in the quadrupled $CO_2$ experiment shown by Andrews et al. (2012). One should expect that the rapid adjustment in response to SRM is different from that due to $CO_2$, because the vertical distribution of the direct forcing is different and the cloud rapid adjustment can be caused by various processes (e.g., changes in atmospheric stability). More detailed studies on cloud processes in SRM are required for the reduction of the uncertainty and for a better assessment of impact of the sulphate geoengineering on climate and human activities.

The multi-model mean horizontal distribution of $\Delta T$ suggests that stratospheric sulphate aerosol geoengineering can delay global warming without significant regional biases, unlike the results of the GeoMIP-G1 experiment (Kravitz et al., 2013a). In G1, the incoming solar radiation was just reduced by a constant fraction, so that the SRM forcing has large latitudinal variation (strong in low-latitudes and weak in high-latitudes). Conversely, in G4, the distribution of sulphate aerosol optical depth (AOD) is internally calculated or externally given, and the reflection of the solar radiation is locally calculated. Here, at least for the prescribed AOD calculated from observed AOD after the 1991 Mount Pinatubo eruption, sulphate aerosols are assumed to spread out globally and form a somewhat uniform distribution as shown in Fig. 1. Because the reflection rate, as well as the incoming solar radiation, depends on the solar zenith angle, as described previously, the resultant SRM forcing does not have large latitudinal variation, as shown in Fig. 9a.

This study has the following three limitations. First, the single-layer model used treats the reflection of downward radiation and that of upward radiation by the same rate. As noted above, however, the reflection rate depends on the incident angle, so errors could be significant in regions that have high solar zenith angle and high surface albedo, such as Greenland and Antarctica.

Second, the SW absorption by the sulphate aerosols has been ignored, because its amount is considered minor compared to the SW reflection. If the absorption by the sulphate aerosols is non-negligible, $E_{WV}$ should be regarded as the sum of a part of SRM forcing by absorption and total reaction due to the change in the water vapour amount, and the forcing and total reaction are not well separated from each other. At least for MIROC-ESM-CHEM, this study confirms that the influence of SW absorption by the sulphate aerosols on $E_{WV}$ is less than 4.5 % by performing the G4 experiment with vanishing SW absorption coefficients of the sulphate aerosols. In other words, the SRM forcing due to SW absorption by the sulphate aerosols is less than 1.5 % of that due to reflection ($F_{SRM}$). The magnitude of errors in the other models should be similar to that in MIROC-ESM-CHEM.

Finally, SW at the surface has been the focus of this analysis and the energy balance has not been considered. $\Delta T$ can be affected by other types of rapid adjustment and feedback. For example, the reduced water vapour in G4 causes less SW absorption by the atmosphere and cooling of the troposphere. The greenhouse effect due to the water vapour would be also decreased. Then, in total, the effect of change in water vapour amount may be a cooling effect (i.e., a positive feedback). On the other hand, in the stratosphere, the LW absorption by the injected sulphate aerosols will heat the air and increase water vapour, which contributes to ozone losses (Tilmes et al., 2008; National Research Council, 2015). Further analysis is required to separate the effect of water vapour from the LW flux. Analyses of the full energy balance and other types of feedback will form part of future work.

## 6   Data availability

All data used in this study, except for the data of MIROC-ESM-CHEM-AMP, are available through the Earth System Grid Federation (ESGF) Network (http://esgf.llnl.gov). The data of MIROC-ESM-CHEM-AMP are available by contacting the corresponding author.

*Author contributions.*   HK, MA, SW, and TS analysed the data. SW, TS, DJ, JM, and JC developed the models and performed the experiment. BK designed and organized the experiment. All authors contributed to the discussion.

The authors declare that they have no conflict of interest.

*Acknowledgements.*   We thank Dr. Aaron Donohoe and anonymous reviewers for useful comments, which greatly help us to improve the manuscript. We thank all participants of the Geoengineering Model Intercomparison Project and their model development teams, CLIVAR/WCRP Working Group on Coupled Modelling for endorsing GeoMIP, and the scientists managing the Earth System Grid data nodes who have assisted with making GeoMIP output available. We thank Drs. Charles Curry, James M. Haywood, and Andy Jones for model development and comments on the manuscript. We also thank Drs. Masahiro Sugiyama, Hideo Shiogama, and Seita Emori for useful comments. HK, MA, and SW were supported by the SOUSEI Program, MEXT, Japan. Simulations of MIROC-based models were conducted using the Earth Simulator. The Pacific Northwest National Laboratory is operated for the U.S. Department of Energy by Battelle Memorial Institute under contract DE-AC05-76RL01830.

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

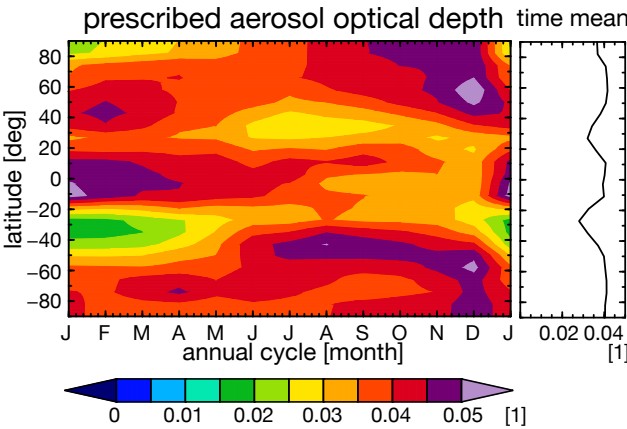

**Figure 1.** Annual cycle and latitudinal distribution of the prescribed aerosol optical depth provided from the GeoMIP for G4 experiment and used in BNU-ESM, MIROC-ESM, and MIROC-ESM-CHEM. Line graph shows the annual mean.

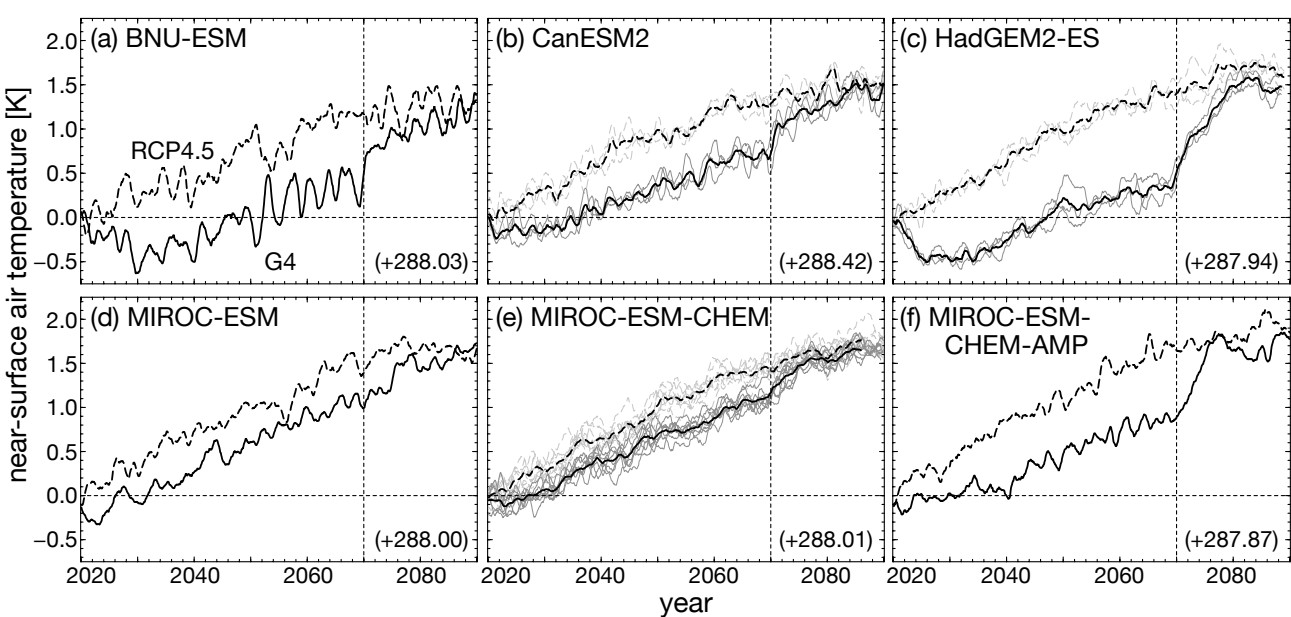

**Figure 2.** Globally averaged surface air temperature in G4 (solid) and RCP4.5 (dashed) experiments. 12-month running mean is applied. Values are offset by the RCP4.5 average from 2018 to 2022, the beginning of SRM, shown at the right bottom on each panel. In panels (b), (c), and (e), black curves show the ensemble mean and grey curves show ensemble members. The vertical dashed lines indicate the SRM termination (2070).

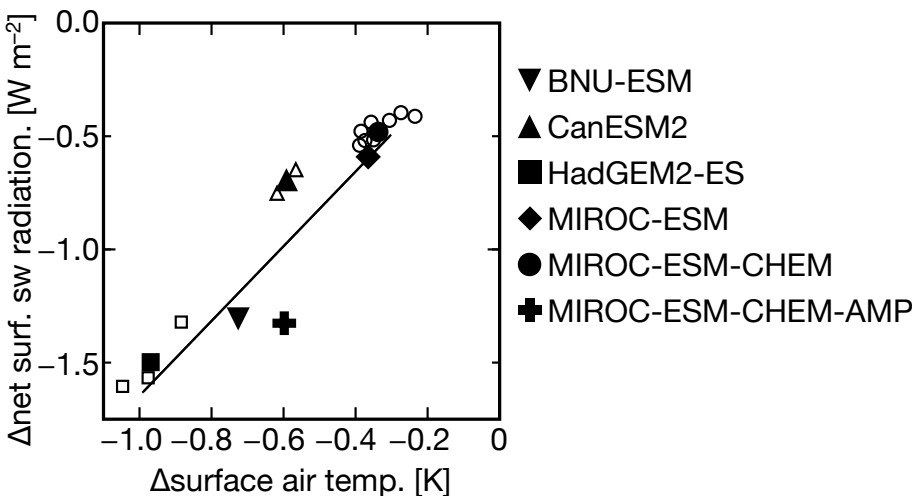

**Figure 3.** Relationship between the difference in the globally and temporally averaged surface air temperature (x-axis) and that of the net shortwave radiation at the surface (y-axis). The term of average is from 2040 to 2069. For CanESM2, HadGEM2-ES, and MIROC-ESM-CHEM, the ensemble mean is shown by filled symbols and the each member by unfilled ones. The regression line is for the filled symbols of the six models.

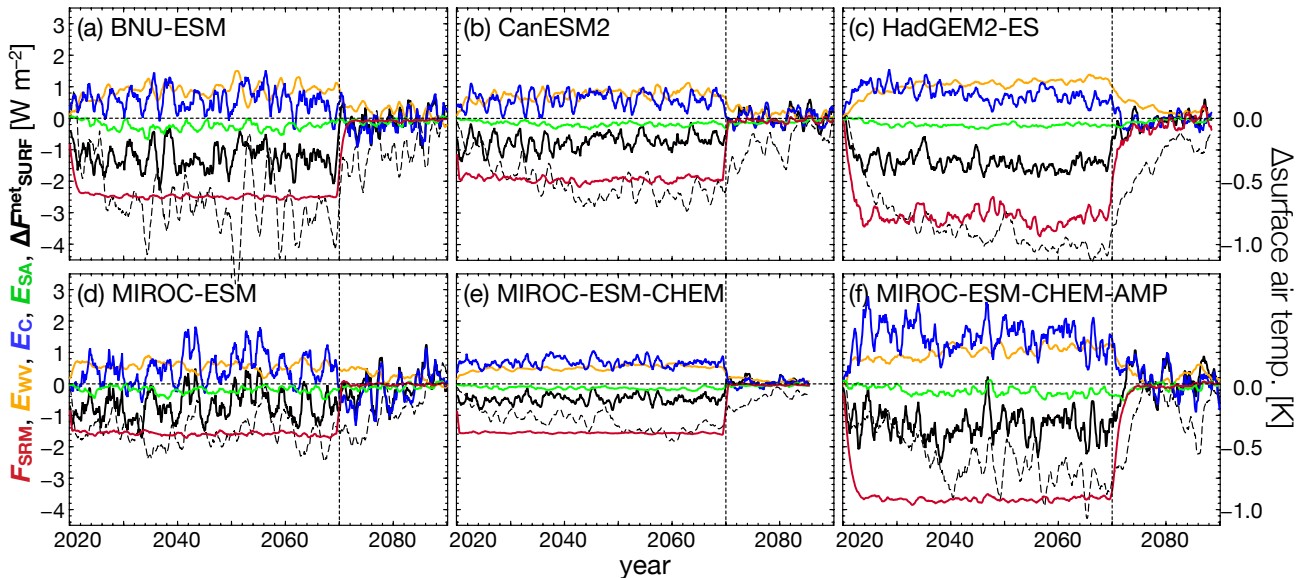

**Figure 4.** Same as Fig. 2 but for the SRM forcing (red), SW feedback due to changes in the water vapour (orange), cloud amounts (blue), and surface albedo (green) defined in Eqs. (9–12), and the difference in the net shortwave radiation at the surface (black, solid). The difference in the surface air temperature is also plotted by dashed-black curves whose values are shown by the right axis.

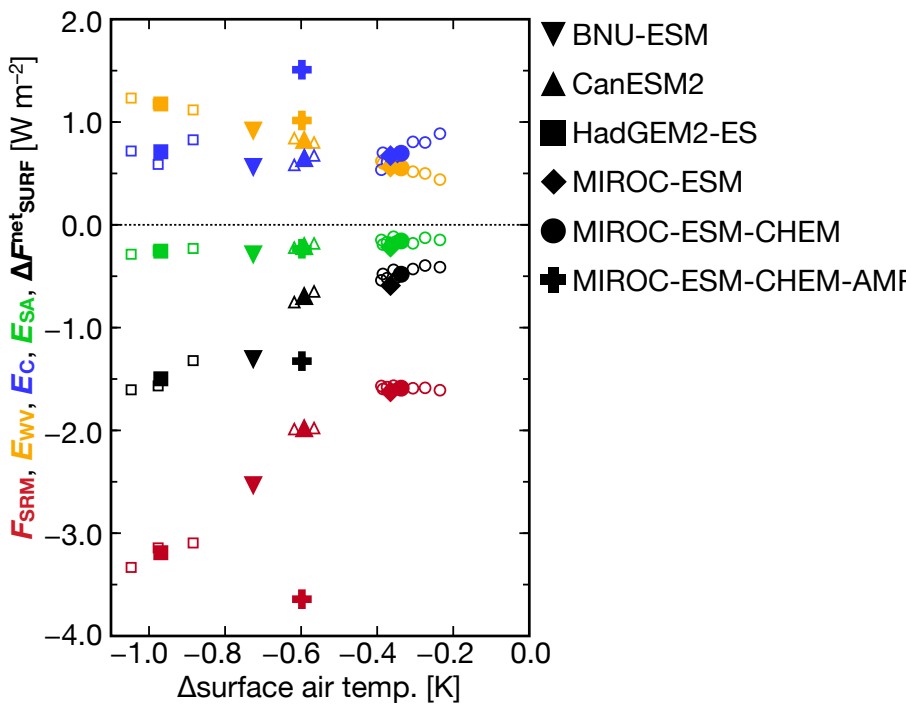

**Figure 5.** Same as Fig. 3 but also for SRM forcing (red), SW feedback due to changes in the water vapour amount (orange), cloud amount (blue), and surface albedo (green).

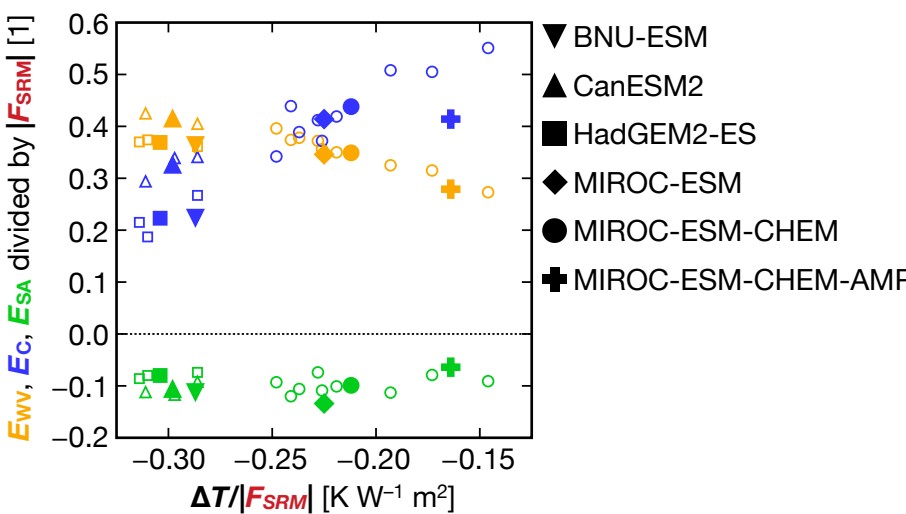

**Figure 6.** Same as Fig. 5 but for the SW feedback effects normalized by the absolute value of the SRM forcing. Note that the difference in the surface air temperature (x-axis) is also divided by $|F_{SRM}|$.

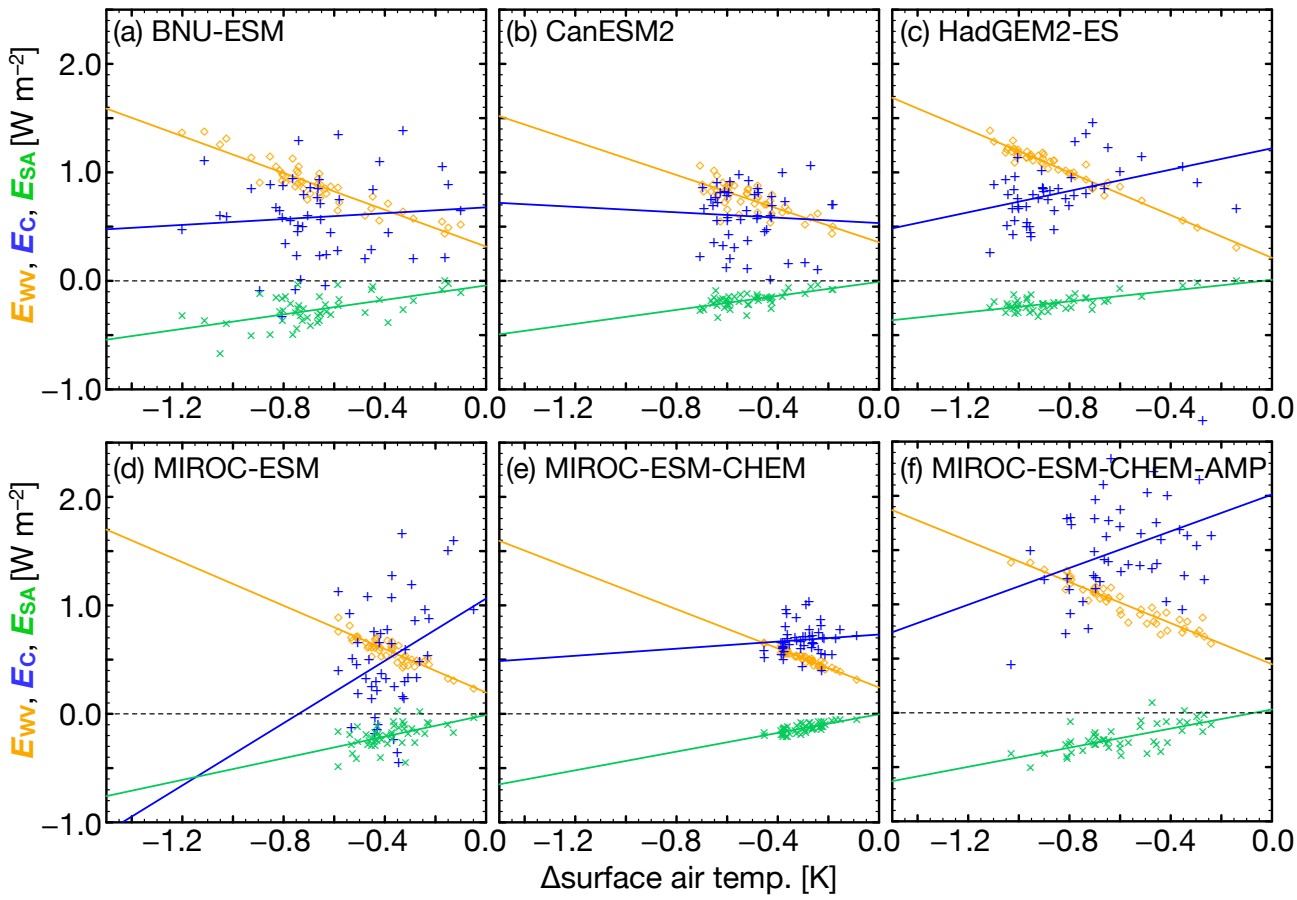

**Figure 7.** Globally and annually averaged relationship between $\Delta T$ (x-axis) and $E_{WV}$ (orange $\diamond$), $E_C$ (blue $+$), and $E_{SA}$ (green $\times$) for each year from 2021 to 2069. Regression line for each plot is shown by the same color, and a slope (feedback parameter), a y-intercept (rapid adjustment), and a correlation coefficient for each plot are shown in Table 3. Ensemble mean data are used for the plots on (b) CanESM2, (c) HadGEM2-ES, and (e) MIROC-ESM-CHEM.

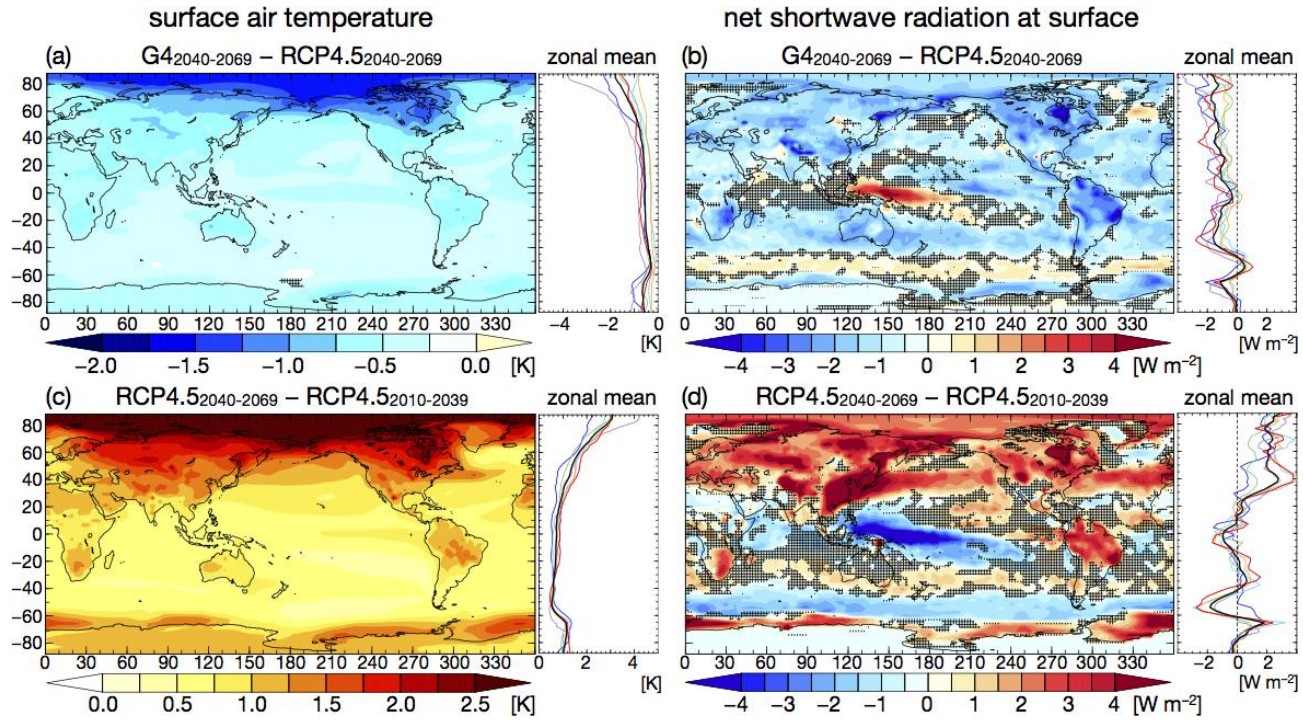

**Figure 8.** Multi-model mean of difference in the surface air temperature and net shortwave radiation at the surface. Panels (a) and (b) show the difference between G4 and RCP4.5 averaged over 2040–2069. Panels (c) and (d) show the difference between RCP4.5 averaged over 2040–2069 and that over 2010–2039. The colour shading shows the horizontal distribution of the multi-model mean and the black thick line on the right-hand side shows the zonal mean of the multi-model mean. Other coloured thin lines display the ensemble mean (or the result of the single run) of each model (blue: BNU-ESM, green: CanESM2, purple: HadGEM2-ES, cyan: MIROC-ESM, orange: MIROC-ESM-CHEM, red: MIROC-ESM-CHEM-AMP). Hatching indicates the region where 2 or more models (out of 6) disagreed on the sign of the difference. Note that the multi-model mean is calculated by averaging ensemble means (or single run for models that has no ensembles) of the six models. That is, in the multi-model mean, each run of a model is weighted by the reciprocal of the ensemble number of the model.

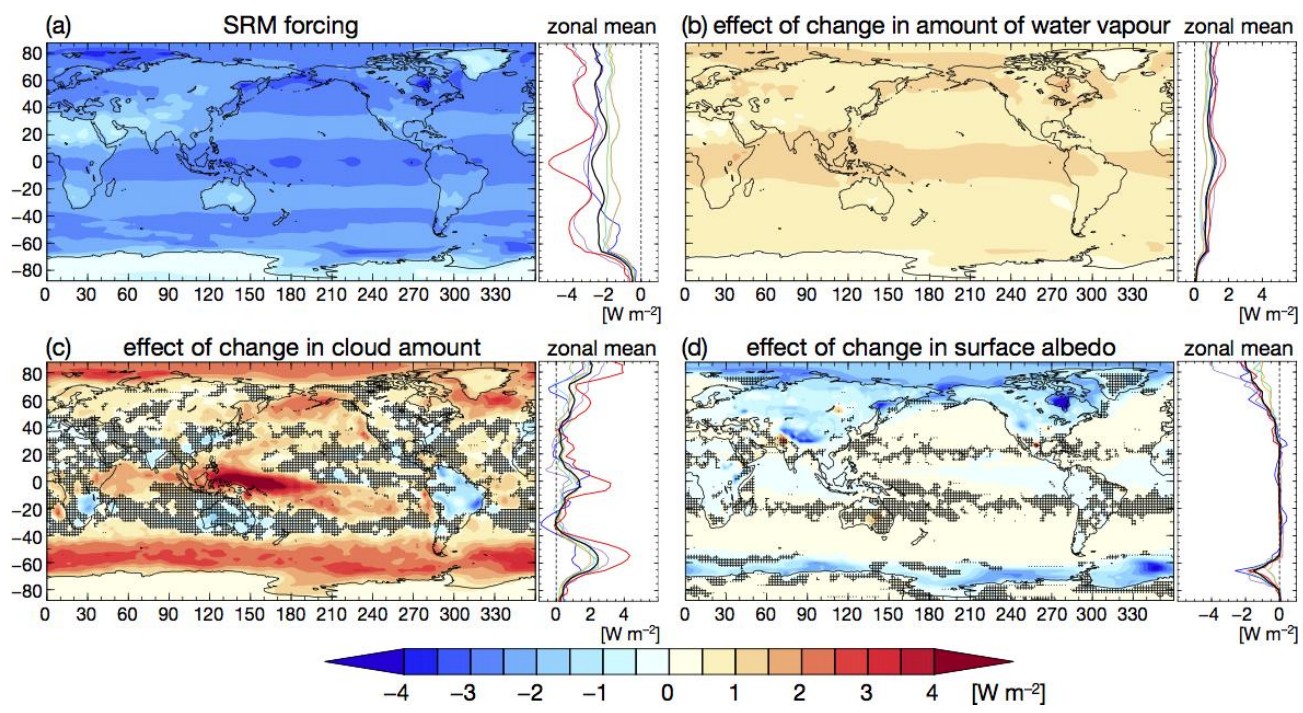

**Figure 9.** Same as Fig. 8 but for multi-model mean of (a) SRM forcing, SW feedback due to changes in the (b) water vapour amount, (c) cloud amount, and (d) surface albedo, averaged over 2040–2069.

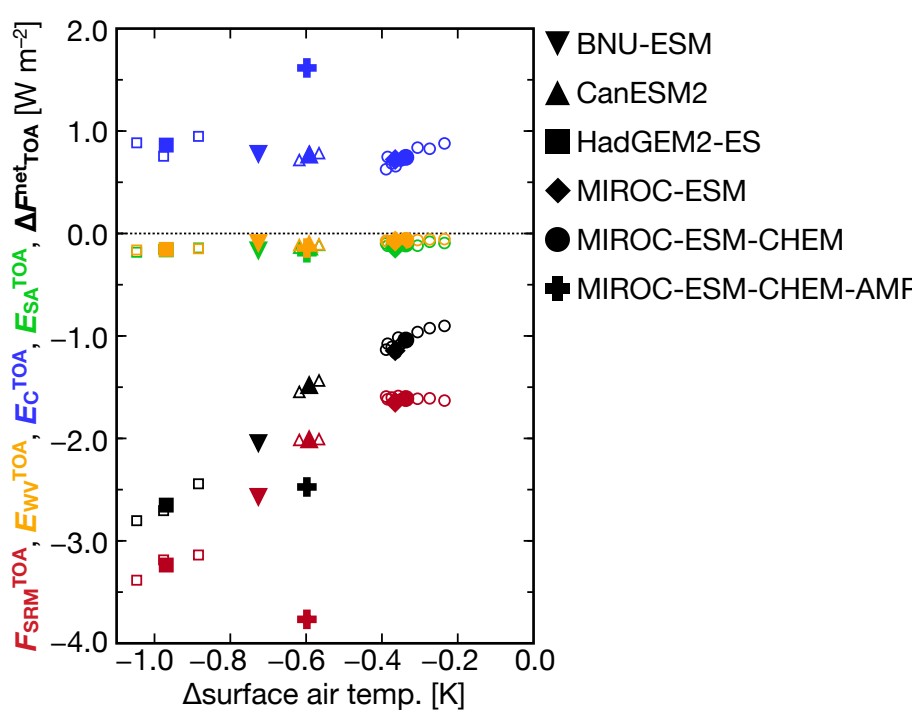

**Figure 10.** Same as Fig. 5 but the variables except for the surface temperature are calculated at the top of the atmosphere.

**Table 1.** Previous studies on simulation of sulphate geoengineering with calculation of stratospheric sulphate aerosols. Injected $SO_2$ amounts, baseline experiments, and model names are shown.

| Studies | $SO_2$ [Tg yr$^{-1}$] | Baseline experimets | Models |
|---|---|---|---|
| Rasch et al. (2008a) | 2–4 | doubled $CO_2$ | CAM3 |
| Robock et al. (2008) | 3–10 | A1B scenario | GISS GCM ModelE |
| Heckendorn et al. (2009), Pierce et al. (2010) | 2–20 | present day (year 2000) | MA-ECHAM4 |
| Niemeier and Timmreck (2015) | 2–200 | RCP8.5 scenario | ECHAM5 |

**Table 2.** Models participating in GeoMIP G4 experiments and used in this study. Manners of simulating sulphate aerosol optical depth (AOD), particle sizes and standard deviation of their log-normal distribution ($\sigma$), and ensemble members are shown for each model.

| Models | Sulphate AOD | Particle size [µm] ($\sigma$) | Ensemble members |
|---|---|---|---|
| BNU-ESM Ji et al. (2014) | Prescribed | 0.426 (1.25) | 1 |
| CanESM2 Arora and Boer (2010); Arora et al. (2011) | Uniform | 0.350 (2.0) | 3 |
| HadGEM2-ES Collins et al. (2011) | Internally Calculated | 0.0065 (1.3), 0.095 (1.4) | 3 |
| MIROC-ESM Watanabe et al. (2011) | Prescribed | 0.243 (2.0) | 1 |
| MIROC-ESM-CHEM Watanabe et al. (2011) | Prescribed | 0.243 (2.0) | 9 |
| MIROC-ESM-CHEM-AMP Watanabe et al. (2011); Sekiya et al. (2016) | Internally Calculated | 0.243 (2.0) | 1 |

**Table 3.** Values of rapid adjustment ($Q_X$) [Wm$^{-2}$], feedback parameter ($-P_X$) [Wm$^{-2}$K$^{-1}$], and correlation coefficient (R$_X$) due to changes in where $X$ = WV, C, SA. Multi-model means are also shown.

| Models | $Q_{WV}$ | $-P_{WV}$ | R$_{WV}$ | $Q_C$ | $-P_C$ | R$_C$ | $Q_{SA}$ | $-P_{SA}$ | R$_{SA}$ |
|---|---|---|---|---|---|---|---|---|---|
| BNU-ESM | 0.32 | −0.85 | −0.93 | 0.68 | 0.14 | 0.08 | $-4.2 \times 10^{-2}$ | 0.33 | 0.56 |
| CanESM2 | 0.36 | −0.78 | −0.75 | 0.53 | −0.12 | −0.06 | $-9.0 \times 10^{-3}$ | 0.32 | 0.65 |
| HadGEM2-ES | 0.21 | −0.98 | −0.97 | 1.22 | 0.49 | 0.40 | $8.7 \times 10^{-3}$ | 0.25 | 0.71 |
| MIROC-ESM | 0.20 | −1.00 | −0.90 | 1.06 | 1.44 | 0.33 | $-9.9 \times 10^{-3}$ | 0.50 | 0.52 |
| MIROC-ESM-CHEM | 0.24 | −0.90 | −0.96 | 0.73 | 0.16 | 0.09 | $-3.5 \times 10^{-3}$ | 0.43 | 0.75 |
| MIROC-ESM-CHEM-AMP | 0.45 | −0.95 | −0.94 | 2.00 | 0.85 | 0.37 | $3.1 \times 10^{-2}$ | 0.44 | 0.67 |
| Multi-model mean | 0.30 | −0.91 | −0.91 | 1.04 | 0.49 | 0.20 | $-4.1 \times 10^{-3}$ | 0.38 | 0.64 |