# Peer review of "Shortwave radiative forcing, rapid adjustment, and feedback to the surface by sulphate geoengineering: Analysis of the Geoengineering Model Intercomparison Project G4 scenario"

_Atmospheric Chemistry and Physics, 2016_

## Referee Comment (RC1) · Anonymous Referee #1 · 15 Aug 2016

Kashimura et al. determine the shortwave radiative forcing at the surface of stratospheric sulfur injection (SAI) and it's changes due to clouds. They use results of experiment G4 of the geoengineering model intercomparison project, constant injection of 2.5 Tg(S)/y, of six different models for the study. They apply a single-layer model of short-wave (SW) radiation to estimate the feedbacks caused by the reduced incoming SW radiation due to the scattering sulfate aerosol layer. This is a strong simplification but it allows to differentiate between different cloud feedbacks.

It is important to know the rate of SW reduction at the surface to estimate the impact

of geoengineering. The single-layer model provides information on the impact of SAI on clouds and the study highlights the differences between the models. A comparable study has not previously been performed. I recommend the publication of this work after considering the following remarks.

**General:**

Kashimura et al. concentrate on SW radiation. However, stratospheric sulfate aerosols absorb long-wave (LW) radiation, which heats the stratosphere. This reduces the efficiency of SAI. The injection rate, necessary to counterbalance a certain anthropogenic forcing, is determined by the top of atmosphere forcing imbalance, not by the SW radiation at the surface. Therefore, the LW absorption is important and the role of LW radiation needs to be discussed. The relevance for the presented results should be described in more detail.

A second aspect which is not or only shortly discussed is the meridional distribution of the aerosols. The two models coupled to an aerosol microphysics show most probably different distributions. This has a clear impact on the forcing (English et al.(2013), Niemeier and Timmreck (2015)). The importance of the particle size is not mentioned et all. Scattering of SW radiation decreases with increasing particle size (Pierce et al. (2010)). Is the particle radius similar in the models prescribing the AOD? Do the two aerosol models simulate similar AOD?

These aspects will not change the presented results but may provide some additional explanation of differences.

**Introduction:**

Line 16: Rasch (2008) and Robock (2008) do not use full aerosol microphysics. E.g. Rasch (2008) prescribe the aerosols. There are several more recent studies available: e.g. Heckendorn et al. (2009), Pierce et al (2010), English et al (2013), Niemeier and Timmreck (2015) all with full aerosol micro-physics. They may provide information of the LW impact.

Impact of LW radiations, particle size and meridional distribution might be discussed in the introduction.

**Methods:** Page 5 end of the page: *'effect on the absorption rate is negligible'.* The absorption in the near infrared should be discussed prior to this point.

**Results:** Line 6: *'for a few decades'* Please be a bit more specific.

Line 17: You discuss at the end the problem of comparing ensemble mean data to single model results. This came to my mind already here.

Page 8:
Line 10: *'except MIROC-ESM-CHEM-AMP'* Why?

Line 22: cooling and heating effect: You may better name it positive and negative forcing.

Line 26: How are the modes of the aerosol module set up? Do you use the same mode width as described in Sekiya (2016)? The injection strength under geoengineering conditions is smaller compared to a volcanic eruption. This may cause

Line 28/29: Why do they differ? Horizontal distribution, particle size?

Line 33: I would expect that the average over time of the AOD is similar between the ensemble members. You may explain this better if you show a zonal mean of the AOD for the two models and, in case they differ, the ensemble members.

Page 9:
1st sentence: *'varies from....'* between the models.

Page 10:
Line 3 and 4: You list many regional details. Can we trust the model in this detail?

Line 31: The difference in meridional distribution of the aerosols are an notable aspect. However, this is important in modeling because the model results differ. So the different

results show possible behavior of nature. Which of them represents nature best is another question.

Page 10/11:
Do the results agree with previous studies?

**Discussion:**

Page 11:
Line 18-20: You may add references.

Page 12:
Line 10 to 15: This is a serious concern. Would your results differ when you use one simulation of each model, e.g. always r1? You can test this to give a less broaden statement here.

Figure 7:
Line thickness differs in the zonal mean plot. Does this show ensemble mean and single results? Please note it somewhere.

You hatch regions were the models agree. Do you mean disagree? The hatching is so strong that it would make no sense to hatch the regions were the models agree.

What do you mean with *'The color tone shows the horizontal distribution'*?

**References**

English, J. M., Toon, O., and M.J., M.: Microphysical simulations of large volcanic eruptions: Pinatubo and Toba, J. Geophys. Res. Atmos., 118, 1880–1895, doi:10.1002/jgrd.50196, 2013.

Heckendorn, P., Weisenstein, D., Fueglistaler, S., Luo, B. P., Rozanov, E., Schraner, M., Thomason, L. W., and Peter, T.: The impact of geoengineering aerosols on stratospheric temperature and ozone, Environ. Res. Lett., 4, 045 108, doi:10.1088/1748-9326/4/4/045108, 2009.

Niemeier, U. and Timmreck, C.: What is the limit of climate engineering by strato-spheric injection of SO2?, Atmospheric Chemistry and Physics, 15, 9129–9141, doi:10.5194/acp-15-9129-2015, http://www.atmos-chem-phys.net/15/9129/2015/, 2015.

Pierce, J. R., Weisenstein, D. K., Heckendorn, P., Peter, T., and Keith, D. W.: Efficient formation of stratospheric aerosol for climate engineering by emission of condensible vapor from aircraft, GRL, 37, L18 805, doi:10.1029/2010GL043975, 2010.

---

## Referee Comment (RC2) · A. Donohoe (Referee) · 23 Aug 2016

This manuscript employs a single column isotropic shortwave radiation model to decompose the changes in the net surface shortwave flux in response to solar radiation management in the geomip model ensemble. The use of the single column model in conjunction with the assumption that changes in clear sky reflection and absorption are due to sulfate aerosol forcing and water vapor feedbacks respectively is very clever (especially putting these changes back into the full sky equations). However, I do question whether the cloud feedback can be isolated from the effective radiative forcing of

aerosols associated with the direct and rapid response of clouds. I suggest an improved methodology below. I highly suspect that much of what the Authors interpret as a cloud feedback (i.e. associated with temperature changes) is actually the cloud changes due to the aerosol forcing itself and is better characterized as a forcing. I also question the use of the surface radiative budget as opposed to the top of atmosphere of tropopause. As such, I think the main conclusions of the manuscript are not supported and the work could be misleading for the field. I do recognize that the analysis pursued could allow the authors to determine the magnitude of forcing and feedbacks associated with each cloud, water vapor and surface albedo changes and, potentially informs which physical processes determine both the robust changes in the ensemble average and the cause of inter-model differences. There is great potential for the work to offer new insights into the response to geoengineering but, as is, the methodology is flawed and conclusions are misleading. I do not recommend publication of the manuscript in its current form; the Author's need to fundamentally modify the methodology and focus of the manuscript.

I'm not sure I understand the rationale/agree with the premise that the net shortwave flux at the surface is a useful metric for understanding inter-model differences in the response to solar radiation management (SRM). Why favor this metric over the forcing, or the net (longwave plus shortwave) radiative change either at the surface or (preferably) the tropopause? Is the an a priori physical reason to expect the correlation between net surface shortwave and temperature response? I could not find one in the manuscript. In particular, the shortwave water vapor feedback differs in both sign and magnitude when considering the surface fluxes versus the tropopause or TOA and it's hard to justify the interpretation of this feedback defined at the surface (as pursued in the current manuscript); in a warmer planet, the moister atmosphere directly absorbs more solar radiation which has a heating impact on the climate system but this reduces the downwelling shortwave flux to the surface which the Authors would interpret as a cooling feedback in the framework used within the manuscript. This feedback is found in the current manuscript to have a magnitude of order one half the net surface shortwave

change and likely confuses the results and interpretation of the manuscript. I'm not sure that the correlation found between the temperature response and net shortwave flux at the surface is anything more than a statistical coincidence (given the number of independent data points available when accounting for expected correlations between ensemble members of the same model). I believe that looking at the same diagnostics (including LW changes) from the perspective of the TOA radiation alongside the surface would help to illuminate the underlying physical mechanisms responsible for the inter-model differences in the response to SRM.

**Main points: Separation of cloud feedbacks from direct aerosol forcing of clouds**
Clouds respond directly to forcing agents (e.g. aerosol, carbon dioxide, etc) and to changes in surface temperature. The IPCC (and field as a whole) includes the rapid cloud response to forcing agents in the "effective" radiative forcing whereas the cloud radiative changes due to surface temperature changes are generally classified as a radiative feedback. The present manuscript associates all the cloud changes with the feedback (equation 11) and I suspect much of what is called a cloud feedback is actually inter-model differences in the effective cloud forcing. This suspicion is based on two lines of evidence:

1. The cloud radiative changes in figure 4 seem to coincide with the nearly step function changes in aerosol as opposed to the surface temperature changes. Panels E and C are the best examples. The cloud radiative changes ramp up almost immediately at 2020, before the surface temperature has decreased and return to near their unperturbed value almost immediately when the SRM stops at year 2070 even though the surface temperature takes longer to recover.

2. The published cloud feedbacks differ in sign and magnitude from those found elsewhere in the literature for the same models. More fundamentally, the Authors conclude that cloud changes damp the response to geo-engineering whereas the models included in the study have been found to have positive net cloud feedbacks in response to CO2 (see Table 1 of Andrews et al. 2012 – Forcing. Feedbacks and climate sensitivity in the CMIP5 coupled atmosphere-ocean climate models) The comparison I'm making is unfair to Authors since I am comparing net cloud radiative impacts at the TOA to the surface SW impact. However, figure 3 of the above manuscript suggests a sign difference for at least the hadGEM3-ES model. Either way, the ensemble average negative cloud feedback suggested by the Authors seems at odds with the literature, is likely confused with the effective forcing and should be further analyzed (remove forcing, look at net radiative impact, compare TOA and surface) since this result contradicts and confuses the existing literature. A fairly straightforward solution to the above objections would be to compute the same fields outlined in equations 10-12 for each year of the simulation where the SRM is approximately constant (2025-2070 ish) and plot the radiative changes of each term versus the surface temperature change for all. As suggested by Gregory, the feedback is the slope of the linear best fit line and the effective forcing of each term is the y-intercept. This would also allow the Authors to calculate the impact of the aerosols on the shortwave absorption within the atmosphere which is alluded to in the discussion. I think this would appropriately isolate the effective forcing of clouds and the Authors might find the very interesting result that the inter-model differences in climate response to SRM is well correlated with effective forcing where the latter includes both the direct forcing of the aerosols and the rapid impact of the aerosols on the cloud radiative effect

**Use of the surface radiation budget** The surface energy budget is not closed with respect to the radiation and it is widely recognized that changes in surface radiation are balanced by turbulent energy fluxes with only small temperature adjustments. Generally, the radiative changes are viewed at a level where the system is closed with respect to radiation – either the tropopause or TOA. It is fair to challenge this paradigm and the surface radiative budget may be useful for geo-engineering but that point should be discussed and analyzed, not taken for granted as it is in the current manuscript. In

particular, one place the surface radiative changes are less than useful is the interpretation of atmospheric solar absorption on the surface energy budget. As the atmosphere warms and moistens it absorbs more shortwave radiation that would have otherwise mostly (since the majority of the Earth's surface is dark) been absorbed at the surface. As a result, less shortwave is fluxed to the surface, which would be seen as a cooling influence on the surface. Yet, in the column average, slightly more shortwave is absorbed. Since most of this additional shortwave absorption occurs in the lower troposphere, where water vapor is abundant, it is tightly coupled to the surface energy budget and will warm the surface even if the surface shortwave flux is reduced as a result. Radiative kernels estimate this feedback to result in $+1.0$ W m$^{-2}$ K$^{-1}$ more absorption in the atmospheric column and $+0.3$ W m$^{-2}$ K$^{-1}$ as measured at the TOA (Donohoe et al. 2014, Shortwavbe and longwave contributions to global warming under increasing CO2, PNAS). Therefore, the surface feedback would be deduced to be $-0.7$ W m$^{-2}$ K$^{-1}$ with the wrong sign and more than twice the magnitude of the changes at the TOA. In the very least, the manuscript should include similar diagnostics at the TOA to resolve this sign paradox and a discussion of these points to support the assertion that surface shortwave changes are a useful metric.

To play devil's advocate, it seems like most of correlation between the temperature response and net surface shortwave comes from the forcing. Is the use of net shortwave at the surface a better predictor of the temperature (statistically distinguishable) from that of forcing alone (surface or TOA)? The latter certainly would result in a stronger regression – and one more consistent with climate sensitivity—than using surface shortwave even if the correlation is slightly worse. More generally, what would the correlation be if one used forcing alongside published estimates of the model's climate sensitivity in response to CO2? It looks like the outlier from the strong relationship between forcing and response is the MIROC-CHEM-AMP which has a pronounced cloud feedback. As suggested above, I believe that cloud feedback is mis-identified and is really an effective forcing associated with with rapid cloud changes due to the direct impact of the aerosols. I think that calculating the effective forcing may offer

a better correlation with the climate response than the net surface shortwave metric used in the manuscript.

Please also note the supplement to this comment:
http://www.atmos-chem-phys-discuss.net/acp-2016-711/acp-2016-711-RC2-supplement.pdf

---

## Author Comment (AC1) · 28 Nov 2016

**"Shortwave radiative forcing and feedback to the surface by sulphate geoengineering: Analysis of the Geoengineering Model Intercomparison Project G4 scenario" by Hiroki Kashimura et al.**

Response to the Referee #1

Dear Referee#1

We thank the referee for a carful review and constructive comments. Please find below the authors' response. In this reply we denote referee's comments and questions using blue; our responses are in black and relevant text in the manuscript in Times font with changes shown in red.

We revised the title of the manuscript as
 "Shortwave radiative forcing, rapid adjustment, and feedback to the surface by sulphate
 geoengineering: Analysis of the Geoengineering Model Intercomparison Project G4 scenario"
following another reviewer's comment.
* * *
Kashimura et al. determine the shortwave radiative forcing at the surface of stratospheric sulfur injection (SAI) and it's changes due to clouds. They use results of experiment G4 of the geoengineering model Intercomparison project, constant injection of 2.5 Tg(S)/y, of six different models for the study. They apply a single-layer model of short-wave (SW) radiation to estimate the feedbacks caused by the reduced incoming SW radiation due to the scattering sulfate aerosol layer. This is a strong simplification but it allows to differentiate between different cloud feedbacks.

It is important to know the rate of SW reduction at the surface to estimate the impact of geoengineering. The single-layer model provides information on the impact of SAI on clouds and the study highlights the differences between the models. A comparable study has not previously been performed. I recommend the publication of this work after considering the following remarks.

General:

Kashimura et al. concentrate on SW radiation. However, stratospheric sulfate aerosols absorb long-wave (LW) radiation, which heats the stratosphere. This reduces the efficiency of SAI. The injection rate, necessary to counterbalance a certain anthropogenic forcing, is determined by the top of atmosphere forcing imbalance, not by the SW radiation at the surface. Therefore, the LW absorption is important and the role of LW radiation needs to be discussed. The relevance for the presented results should be described in more detail.

=>We agree that the LW absorption by the stratospheric aerosols is important for studying SAI. However, there are many interactions among LW, temperature, and various other components of the climate system through the emission and absorption of LW. Because of such complexity, unlike the SW effects that we have explored in this study, it is difficult to distinguish and estimate the LW effect of each process.
We carefully consider how to include an analysis and discussion about LW radiation, and decided to estimate the rapid adjustment (or response) of LW radiation in the clear-sky,

by using a method similar to the Gregory plot. (Note that another referee requested to distinguish *rapid adjustments*, which is independent on ΔT, and *feedbacks*, which is proportional to ΔT, from what we call "feedback effect" in the previous manuscript; and a method similar to the Gregory plot was added to the revised manuscript.) The LW rapid response should include, at least, the effect of LW absorption by the stratospheric sulphate aerosols and that of the rapid adjustment of the water vapour. We added such analysis and discussion on the new Section 4.2 as follows.

Page 14–15
Section 4.2 Rapid adjustment of longwave radiation

This study has concentrated on SW for the reasons described in Section 1; however, it may be valuable for some readers to mention the role of LW. A well-known effect of LW in the sulphate aerosol geoengineering is heating of the stratosphere. The sulphate aerosols induced by the $SO_2$ injection absorb LW and heat the stratosphere (e.g., Heckendorn et al., 2009; Pitari et al., 2014). For the energy budget at TOA, increase of the LW absorption results in decrease of the outgoing LW, which manifests as a heating of the climate system. Needless to say, there are many interactions among LW, temperature, and various other components of the climate system, through the emission and absorption of LW. Because of such complexity, unlike the SW changes that we have explored in this study, it is difficult to distinguish and estimate the effect of each factor on LW changes.

One possible and useful analysis for LW is to estimate the rapid adjustment (or response), which is independent of ΔT, by the same method used in Section 3.4. Gregory-like plots are made for the difference of net LW for clear-sky at the surface ($\Delta LW_{SURF}^{CS}$) and at TOA ($\Delta LW_{TOA}^{CS}$) as shown by black "+" signs and red "✗" signs, respectively, in Fig. 11. The rapid adjustment in the clear-sky at the TOA shown by the y-intercept of the $\Delta LW_{TOA}^{CS}$ regression line shows a heating effect of about 0.57 Wm$^{-2}$ in the multi-model mean. This rapid adjustment should mainly consist of the effect of LW absorption due to the stratospheric sulphate aerosols, since the decrease of the water vapour suggested by the rapid adjustment of EWV yields less LW absorption and an increase in outgoing LW at TOA (i.e., sense of cooling). It is important to take this heating effect in mind when we consider the energy budget at TOA for the sulphate geoengineering. Though the sulphate aerosols' LW effect is significant at TOA, such effect might become less significant at the surface, because the rapid adjustment estimated from $\Delta LW_{SURF}^{CS}$ is small compared to the SRM forcing and total reactions at the surface.

A second aspect which is not or only shortly discussed is the meridional distribution of the aerosols. The two models coupled to an aerosol microphysics show most probably different distributions. This has a clear impact on the forcing (English et al. (2013), Niemeier and Timmreck (2015)).

=>Because HadGEM2-ES calculates sulphate aerosols both in the stratosphere and troposphere in the same way and does not output the *stratospheric* sulphate AOD separately, we cannot obtain AOD due to the $SO_2$ injection accurately. The difference (G4 – RCP4.5) of the sulphate AOD, which is the sum of the AOD in the troposphere and that in the stratosphere, may give an *approximate* distribution of the stratospheric sulphate AOD in G4, but a fair comparison with the prescribed AOD and MIROC-ESM-CHEM-AMP is impossible. For readers who want to refer to the *approximate* AOD distribution in HadGEM2-ES, the stratospheric sulphate AOD distribution in MIROC-ESM-CHEM-AMP, and the prescribed AOD, we provide a figure (Fig. S1) as a supplemental file. This figure

shows that the difference in the globally averaged amount of AOD should be significant rather than the meridional distribution of the AOD. This is newly mentioned in the manuscript as follows.

Page 10, line 11–16

It is the difference in the mean AOD rather than its meridional distribution as shown in Fig. S1 that leads to the underestimation of the AOD in G4. The globally and temporally averaged stratospheric sulphate AOD in MIROC-ESM-CHEM-AMP is 0.083 and that in HadGEM2- ES is approximately 0.054, though that of the prescribed AOD is 0.037. Note that the above value for HadGEM2-ES is the difference (G4 – RCP4.5) in the sulphate AOD for both troposphere and stratosphere because HadGEM2-ES does not calculate the sulphate aerosols in the tropospheric and stratosphere separately.

The importance of the particle size is not mentioned at all. Scattering of SW radiation decreases with increasing particle size (Pierce et al. (2010)). Is the particle radius similar in the models prescribing the AOD? Do the two aerosol models simulate similar AOD?

=>We added a sentence mentioning the importance of the particle size in Introduction. In addition, we added a paragraph describing the particle size of participating models in Section 2, and we added the particle sizes to Table 1.

Page 3, line 17–19, Section 1

Even though the prescribed AOD is given, a difference in an assumed particle size for the stratospheric sulphate aerosols causes difference in the SRM forcing (Pierce et al. 2010).

Page 4, line 31–page 5 line 5, Section 2

The mean stratospheric sulphate aerosol particle sizes and standard deviation of their log-normal distribution (σ) in each model are also shown in Table 1. In HadGEM2-ES, the tropospheric aerosol scheme and the associated microphysical properties (Bellouin et al. 2011) is simply extended into the stratosphere. Modifications to the stratospheric aerosol size distribution have been applied in subsequent HadGEM2-ES studies (Jones et al. 2016a,b), but have not been applied here. In MIROC-ESM-CHEM-AMP, the microphysics module for stratospheric sulphate aerosols treats them in three modes as shown in Table 2 in Sekiya et al. (2016); however, to calculate radiative processes on the aerosols, a particle size of 0.243 μm is assumed for simplification. In addition, the microphysics of the tropospheric sulphate aerosols is not calculated in MIROC-ESM-CHEM-AMP to avoid drift in the simulated climate.

These aspects will not change the presented results but may provide some additional explanation of differences.

Introduction:

Line 16: Rasch (2008) and Robock (2008) do not use full aerosol microphysics. E.g. Rasch (2008) prescribe the aerosols.

=>For Rasch (2008) and Rohbock (2008), we understood that the particle size distribution was not internally calculated but prescribed in their model. We added the following sentence to mention this.

Page 2, line 18–19, Section 1:

The models used in these two studies include formation, transportation, and removal of the stratospheric sulphate aerosols, but the particle size distribution was prescribed.

There are several more recent studies available: e.g. Heckendorn et al. (2009), Pierce et al (2010), English et al (2013), Niemeier and Timmreck (2015) all with full aerosol micro-physics.

=>Thank you for giving us useful info. We cited Heckendorn et al. (2009), Pierce et al. (2010), and Niemeier and Timmreck (2015). English et al (2013) was not cited because it is a study about large volcanic eruptions.

Page 2, line 19–23, Section 1
Heckendorn et al. (2009) and Pierce et al. (2010) calculated full microphysics of sulphate aerosols with an assumption of zonally homogeneous conditions. They simulated 2–20 Tg yr$^{-1}$ SO$_2$ injection with a present day (year 2000) condition run as their reference simulation. Niemeier and Timmreck (2015) used models with full microphysics of sulphate aerosols, and performed a sulphate geoengineering experiment with SO$_2$ injection rates of 2–200 Tg yr$^{-1}$ to counteract the anthropogenic forcing of RCP8.5.

They may provide information of the LW impact. Impact of LW radiations, particle size and meridional distribution might be discussed in the introduction.

=>We added sentences mentioning the importance of the particle size and meridional distribution in the introduction as follows.

Page 3, line 15–19, Section 1
On processes related to the SRM forcing, modelled aerosol microphysics including formation, growth, transportation, and removal may differ, and such differences result in the difference in meridional distribution of the aerosol optical depth (AOD). Even though the prescribed AOD is given, a difference in an assumed particle size for the stratospheric sulphate aerosols causes difference in the SRM forcing (Pierce et al., 2010).

Importance of LW radiation was introduced and discussed in the new Section 4.2. We consider that discussing the LW radiation in the introduction will impair the flow of sentences.

Methods:
Page 5 end of the page: 'effect on the absorption rate is negligible'. The absorption in the near infrared should be discussed prior to this point.

=>We carefully consideredthis suggestion, but to discuss influence of the near infrared radiation quantitatively, we need some measures introduced in Section 2. Therefore, this cannot be discussed at this point, and we kept the discussion about the near infrared radiation at the end of the manuscript.

Results:
Line 6: 'for a few decades' Please be a bit more specific.

=> Expression was changed to "10–25 years".

Line 17: You discuss at the end the problem of comparing ensemble mean data to single model results. This came to my mind already here.

=>Because inserting the discussion here will break the flow of the sentences, we added a sentence announcing that the discussion is given in Section 4. Here, Section 4 is newly added for Discussion.

Page 9, line 3–4, Section 3.1
One concern is that half the models used in this study have only one ensemble member, and half are MIROC-based models. The effects of this are analysed in Section 4.3 and shown to be relatively unimportant.

=> This reason was described in the next subsection. We added a short note to announce this to readers.

Page 9, line 25–26, Section 3.2
The strengths of $E_{WV}$ and $E_C$ are comparable in each model except MIROC-ESM-CHEM-AMP (a reason for this exception is discussed in the next subsection).

Line 22: cooling and heating effect: You may better name it positive and negative forcing.

=>We carefully consider this and also from the other comments, we recognized "cooling/heating" is misleading, since the decrease/increase of SW at the surface does not necessary causes cooling/heating of the surface air temperature in total (including the effects of LW radiation etc.). However, the expression "positive and negative" may also confuse readers because, one may read "positive" as "plus in sign in amount" or "direct proportion to ΔT" when the word is modifying feedback effect. Hence, we revised the expression of "cooling/heating" that was modifying feedbacks to, for example, "decrease/increase of net SW at the surface" through the manuscript. We consider "cooling" used for the SRM forcing and temperature is not misunderstandable, so that we remained such expression in the manuscript.

Line 26: How are the modes of the aerosol module set up? Do you use the same mode width as described in Sekiya (2016)? The injection strength under geoengineering conditions is smaller compared to a volcanic eruption. This may cause

=>We used the same mode as Sekiya et al. (2016) for stratosphere, but unlike Sekiya et al. the calculation of the sulphate microphysics was not performed in the troposphere to avoid an unexpected drift of the simulated climate and keep the climate in MIROC-ESM-CHEM-AMP in RCP4.5 similar to that in MIROC-ESM-CHEM. This info is now described in Section 2.

Page 5, line 1–5, Section 2
In MIROC-ESM-CHEM-AMP, the microphysics module for stratospheric sulphate aerosols treats them in three modes as shown in Table 2 in Sekiya et al. (2016); however, to calculate radiative processes on the aerosols, a particle size of 0.243 μm is assumed for simplification. In addition, the microphysics of the tropospheric sulphate aerosols is not calculated in MIROC-ESM-CHEM-AMP to avoid drift in the simulated climate.

Line 28/29: Why do they differ? Horizontal distribution, particle size?

=>We checked the sulphate AOD in MIROC-ESM-CHEM and HadGEM2-ES, and compared them with the prescribed AOD. We found that the reason for the underestimate is the estimated mean amount of the AOD rather than the qualitative difference in the meridional distribution as shown in Fig. S1. We added the following sentences to the manuscript and added a new figure as a supplement. Unfortunately, we cannot separate the *stratospheric* sulphate AOD from the output data of HadGEM2-ES, since it does not distinguish sulphate aerosols in the troposphere and stratosphere.

Page 10 line 11–16, Section 3.3
It is the difference in the mean AOD rather than its meridional distribution as shown in Fig. S1 that leads to the underestimation of the AOD in G4. The globally and temporally averaged stratospheric sulphate AOD in MIROC-ESM-CHEM-AMP is 0.083 and that in HadGEM2- ES is approximately 0.054, though that of the prescribed AOD is 0.037. Note that the above value for HadGEM2-ES is the difference (G4 – RCP4.5) in the sulphate AOD for both troposphere and stratosphere because HadGEM2-ES does not calculate the sulphate aerosols in the tropospheric and stratosphere separately.

Line 33: I would expect that the average over time of the AOD is similar between the ensemble members. You may explain this better if you show a zonal mean of the AOD for the two models and, in case they differ, the ensemble members.

=>Here, we said that CanESM2 and MIROC-ESM-CHEM have no differences in SRM forcing among ensemble members, but HadGEM2-ES has. The expression might be confusing, so that we slightly changed the word. For HadGEM2-ES, we drew the mean seasonal cycles of the stratospheric AOD and attached them as a supplement file. It is clear that even averaging over 30 years, the meridional distribution of the stratospheric AOD differs among the ensemble members for HadGEM2-ES. We added the following sentence.

Page 10, line 21–22, Section 3.3
Even after averaging over 30 years, the mean seasonal cycles of the sulphate AOD can differ among the ensemble members as shown in Fig. S1.

Page 9:
1st sentence: 'varies from....' between the models.

=>The expression was added as suggested.

Page 10, line 23–24, Section 3.3
Pitari et al. (2014) have shown that SW radiative forcing at the tropopause calculated off-line by a radiative transfer code (Chou and Suarez, 1999; Chou et al., 2001) varies from around –2.1 to –1.0 W m$^{-2}$ between the models.

Page 10:
Line 3 and 4: You list many regional details. Can we trust the model in this detail?

=>Grid intervals of the models are equal to or narrower than 2.8125 deg, so that the mentioned regions are well resolved in the model. However, properties of the Sea of Okhotsk and Hudson Bay may depend on related channels, which may be not well resolved. We added the following sentences to note about this.

Page 13, line 15–17, Section 3.5
Here, model grid intervals are equal to or narrower than 2.8125 deg, so that the geographical regions mentioned above are represented by enough grid points. However, properties of the Sea of Okhotsk and Hudson Bay may depend on related channels, which may be not well resolved.

Line 31: The difference in meridional distribution of the aerosols are an notable aspect. However, this is important in modeling because the model results differ. So the different results show possible behavior of nature. Which of them represents nature best is another question.

=>For the present anyone cannot answer, "Which of them represents nature best?" because there are no field experiments on SAI in the global scale and a long period. Comparison with the observational data of volcanic eruptions is useful but there are significant difference between SAI and natural volcanic eruption (e.g., continuity of injection, amounts and particle sizes of aerosols).

Page 10/11:
Do the results agree with previous studies?

=>Geographical distribution of ΔT agrees with previous studies (e.g., Robock et al., 2008), and that of $E_{WV}$ is consistent with decrease of precipitation reported by Rasch et al. (2008) and Robock et al. (2008). For other measures, we could not find the previous studies that can be fairly compared with this study (i.e., simulation of sulphate geoengineering; not by reducing the solar constant.). We added the following sentences to mention that our result of ΔT and $E_{WV}$ are consistent with previous studies.

Page 12, line 10, Section 3.5
Such features agree with previous studies such as Robock et al. (2008).

Page 13, line 11–12, Section 3.5
The slight increase of $E_{WV}$, which implies less water vapour, in the equatorial region is consistent of decrease of precipitation reported by Rasch et al. (2008a) and Robock et al. (2008) under SRM.

Discussion:
Page 11:
Line 18-20: You may add references.

=> We added Rasch et al., (2008b) and Kremser et al., (2016) for the references.

Page 15, line 30–Page 16 line 2, Section 5
Inter-model variations comprise a substantial range, and narrowing this uncertainty is essential 30 for understanding the effects of sulphate geoengineering and its interactions with chemical, micro-physical, dynamical, and radiative processes related to the formation, distribution, and shortwave-reflectance of the sulphate aerosols introduced from the $SO_2$ injection (Rasch et al., 2008b; Kremser et al., 2016).

Page 12:
Line 10 to 15: This is a serious concern. Would your results differ when you use one

simulation of each model, e.g. always r1? You can test this to give a less broaden statement here.

=>As suggested, we tested how the multi-model mean results differ when using r1 data only. This result is shown in Fig. S2 in the supplement file. We also checked how the multi-model mean results differ when adding a weight of 1/3 to MIROC-based models to remove the bias that 3 out of 6 models is the MIROC-based model. This result is shown in Fig. S3. In both cases, we did not find significant difference compared with Fig. 9 in the manuscript, so that we can state that inequality in the number of ensemble and participating models have no significant effects to our results. These are described in the new Section 4.3.

Page 15, line 4–12, Section 4.3
4.3 Inequality in the number of ensemble and participating models
One concern in this study is the half of the models used have only one ensemble member, and half are MIROC-based models. Because the numbers of ensemble members differ among models as listed in Table 1, each member in each model is not equally weighted in calculation of the multi-model means described in Section 3.5. Responses to the SRM forcing in the three MIROC-based models should be similar to each other as shown in Fig. 6, so that the results of multi-model mean can be biased to that of the MIROC-based models. Therefore, we re-calculated multi-model means are calculated by using only one run for each model (Fig. S2), and also tested multi-model means with a weight of 1/3 multiplied for the MIROC-based models (Fig. S3). There are no significant difference among Figs. 9, S2, and S3. Therefore, inequality in the number of ensemble and participating models has no significant effects on our results.

Figure 7:
Line thickness differs in the zonal mean plot. Does this show ensemble mean and single results? Please note it somewhere.

=>Black line is thicker than others, because black line shows the multi-model mean. Other coloured lines have the same thickness. We add "thick" and "thin" in the expression.
The ensemble mean and single results are not distinguished by line thickness. Readers need to remember which model has an ensemble, but we think this is not difficult for the readers.

Caption of Fig. 8: the black thick line on the right-hand side shows the zonal mean of the multi-model mean. Other coloured thin lines display the ensemble mean

You hatch regions were the models agree. Do you mean disagree? The hatching is so strong that it would make no sense to hatch the regions were the models agree.

=>Hatching indicates the region where 2 or more models disagreed on the sign. Namely, the region where 6 all models show the same sign and where 5 models show the same sign are not hatched, but the regions where only 4 or 3 models show the same sign are hatched. The previous expression might be unreadable, so that we changed the expression as follows:

Caption of Fig. 8: Hatching indicates the region where 2 or more models (out of 6) disagreed on the sign of the difference.

What do you mean with 'The color tone shows the horizontal distribution'?

=>This is just an expression problem. We mean colour shading on the maps.

Caption of Fig. 8: The colour shading shows the horizontal distribution of the multi-model mean

References

English, J. M., Toon, O., and M.J., M.: Microphysical simulations of large volcanic eruptions: Pinatubo and Toba, J. Geophys. Res. Atmos., 118, 1880–1895, doi:10.1002/jgrd.50196, 2013.

Heckendorn, P., Weisenstein, D., Fueglistaler, S., Luo, B. P., Rozanov, E., Schraner, M., Thomason, L. W., and Peter, T.: The impact of geoengineering aerosols on stratospheric temperature and ozone, Environ. Res. Lett., 4, 045108, doi:10.1088/1748-9326/4/4/045108, 2009.

Niemeier, U. and Timmreck, C.: What is the limit of climate engineering by stratospheric injection of SO2?, Atmospheric Chemistry and Physics, 15, 9129–9141, doi:10.5194/acp-15-9129-2015, http://www.atmos-chem-phys.net/15/9129/2015/, 2015.

Pierce, J. R., Weisenstein, D. K., Heckendorn, P., Peter, T., and Keith, D. W.: Efficient formation of stratospheric aerosol for climate engineering by emission of condensible vapor from aircraft, GRL, 37, L18 805, doi:10.1029/2010GL043975, 2010.

---

## Author Comment (AC2) · 28 Nov 2016

**"Shortwave radiative forcing and feedback to the surface by sulphate geoengineering: Analysis of the Geoengineering Model Intercomparison Project G4 scenario" by Hiroki Kashimura et al.**

Response to the Referee #2, Dr. Aaron Donohoe

Dear Dr. Donohoe

We thank you for a carful review and constructive comments. Please find below the authors' response. In this reply we denote referee's comments and questions using blue; our responses are in black and relevant text in the manuscript in Times font with changes shown in red.

The referee's comments are kindly repeated in detail after "main points", so that quotations of changed sentences from the manuscript is written after the comments in "main points".
* * *
This manuscript employs a single column isotropic shortwave radiation model to decompose the changes in the net surface shortwave flux in response to solar radiation management in the geomip model ensemble. The use of the single column model in conjunction with the assumption that changes in clear sky reflection and absorption are due to sulfate aerosol forcing and water vapor feedbacks respectively is very clever (especially putting these changes back into the full sky equations).

However, I do question whether the cloud feedback can be isolated from the effective radiative forcing of aerosols associated with the direct and rapid response of clouds.

=> We recognized that we did not distinguish the rapid response (or adjustment), which does not depend on ΔT, and feedback, which is proportional to ΔT in the previous manuscript. This may be the main reason for many of your comments. What we called "feedback" in the previous manuscript was the sum of the rapid response and feedback. In the revised manuscript, we defined "rapid adjustment" and "feedback" as described above, and we defined the word "total reaction" as the sum of rapid adjustment and feedback, for convenience. We revised the expression related to "feedback" through the text.

We also revised the title of the manuscript as
"Shortwave radiative forcing, rapid adjustment, and feedback to the surface by sulphate geoengineering: Analysis of the Geoengineering Model Intercomparison Project G4 scenario".

I suggest an improved methodology below. I highly suspect that much of what the Authors interpret as a cloud feedback (i.e. associated with temperature changes) is actually the cloud changes due to the aerosol forcing itself and is better characterized as a forcing.

=>Thank you for the suggestion. We used the methods similar to the Gregory plots and found that the previously called "cloud feedback" (now we call this "total reaction of clouds") is a rapid adjustment due to the cloud amount change. The referee's suspicion was correct.

Though the rapid adjustment is characterized as a forcing (i.e., effective radiative forcing; ERF) in the recent studies of climate change, we consider that for the study of geoengineering simulation, it is better to separate the direct forcing and rapid adjustment to explore which processes have a large uncertainty in the sulphate geoengineering simulation, which is not well verified by observations or field experiments in global scale. We added sentences mentioning these points in Introduction.

I also question the use of the surface radiative budget as opposed to the top of atmosphere of tropopause. As such, I think the main conclusions of the manuscript are not supported and the work could be misleading for the field.

=> This study used net shortwave radiation (SW) at the surface, but did not consider the radiative (energy) budget. We consider that the SW at the surface is very important for vegetation and human activities such as agriculture and solar power generation, and they will be strongly affected by the solar radiation management (SRM). Moreover, the recent study of Kleidon et al. (2015) showed that longwave radiation (LW), sensible heat flux, and latent heat flux can be derived from SW changes at the surface.
On the other hand, we agree that many studies on the climate system used the energy budget at the top of the atmosphere (TOA) and many readers in the field of climate science are accustomed to considering at TOA. Thus, we introduced the measures calculated at TOA and compared them with those at the surface in the revised manuscript. We consider that this discussion clarifies the meaning of the conclusions for readers in the climate science.

I do recognize that the analysis pursued could allow the authors to determine the magnitude of forcing and feedbacks associated with each cloud, water vapor and surface albedo changes and, potentially informs which physical processes determine both the robust changes in the ensemble average and the cause of inter-model differences. There is great potential for the work to offer new insights into the response to geoengineering but, as is, the methodology is flawed and conclusions are misleading. I do not recommend publication of the manuscript in its current form; the Author's need to fundamentally modify the methodology and focus of the manuscript.

=>We consider that the first reason why the referee thought, "the methodology is flawed and conclusions are misleading" is our misuse of the word "feedback". The second reason is a lack of explanation of why we use the net SW at the surface. And the third reason is a lack of comparison with the estimation at TOA. We have corrected the word misuse and added the explanation (in Introduction) and discussion (Section 4.1), so that we believe our study becomes valuable for readers.

I'm not sure I understand the rationale/agree with the premise that the net shortwave flux at the surface is a useful metric for understanding inter-model differences in the response to solar radiation management (SRM). Why favor this metric over the forcing, or the net (longwave plus shortwave) radiative change either at the surface or (preferably) the tropopause?

=>As described above, SW flux at the surface is important to consider the influence of SRM to vegetation and human activities; in addition, LW radiation, sensible heat flux, and latent heat flux can be derived from SW changes at the surface as reported by Kleidon et al. (2015). These are the reason for using surface SW radiation in this study.

Is there an a priori physical reason to expect the correlation between net surface shortwave and temperature response? I could not find one in the manuscript.

=> We simply consider that it is natural to expect the correlation between changes in net SW radiation at the surface and that in surface air temperature, because these two are in the relation of "forcing and response". Though detailed analyses of full energy balance are required for accurate prediction of ΔT, it is useful and important to show a rough and easy relation for ΔT. The strong correlation between ΔT and $\Delta F^{net}_{SURF}$ at least for the range of -1.1 < ΔT < 0.2 is a part of findings in this study as shown in Fig. 3.

In particular, the shortwave water vapor feedback differs in both sign and magnitude when considering the surface fluxes versus the tropopause or TOA and it's hard to justify the interpretation of this feedback defined at the surface (as pursued in the current manuscript); in a warmer planet, the moister atmosphere directly absorbs more solar radiation which has a heating impact on the climate system but this reduces the downwelling shortwave flux to the surface which the Authors would interpret as a cooling feedback in the framework used within the manuscript. This feedback is found in the current manuscript to have a magnitude of order one half the net surface shortwave change and likely confuses the results and interpretation of the manuscript.

=>Your comment is correct. The effect of water vapour differs in both sign and magnitude when considering at the surface and at TOA. Amounts of water vapour in G4 is less than that in RCP4.5, and SW absorption rate of the atmosphere in G4 is less than that in RCP4.5. This means more incoming solar radiation reaches the surface (i.e., sense of heating). On the other hand, at TOA, less absorption rate results in increase of outgoing SW radiation (i.e., sense of cooling). At TOA, the upwelling SW radiation that is affected by the absorption rate experiences a reflection at the surface. Therefore the magnitude at TOA is much less than that at the surface. This interpretation is consistent with our results at the surface and newly added results at TOA. These results and discussion are added in the new Section 4.1, and we consider this section clarifies the meaning of the water vapour reaction.
We also revised the expression "cooling/heating" for rapid responses and feedbacks at the surface to simply "decrease/increase of net SW at the surface", because decrease/increase of the SW at the surface does not necessary result in cooling/heating in total (including effects of LW).

I'm not sure that the correlation found between the temperature response and net shortwave flux at the surface is anything more than a statistical coincidence (given the number of independent data points available when accounting for expected correlations between ensemble members of the same model).

=> As described above, we consider that it is natural to expect the correlation between changes in net SW radiation at the surface and that in surface air temperature, because these two are in the relation of "forcing and response".
Six data points (one from each model) are used to obtain the correlation coefficient of 0.88. Ensemble mean is used for the models that have ensemble runs to avoid overweighting the models that have many ensemble runs. We consider that the number of data points is enough to state the correlation.
At least for the range of ΔT from –1.1 to 0.2 K as shown by Fig. 3, it is a statistical fact that ΔT and $\Delta F_{SURF}^{net}$ has a good correlation.
We added some words to clarify as follows.

Page 8, line 22–28, Section 3.1

For CanESM2, HadGEM2-ES, and MIROC-ESM-CHEM, the filled symbols indicate the ensemble mean whilst the unfilled symbols indicate individual ensemble members; for the other models, the filled symbols indicate the results of a single run. This figure shows a strong correlation between the mean $\Delta T$ and $\Delta F_{SURF}^{net}$ ; the correlation coefficient for the six filled symbols is 0.88. This strong correlation allows $\Delta F_{SURF}^{net}$ to be used as a measure of the SRM effects at least for $-1.1 < \Delta T < -0.2$ K, although the surface air temperature depends on the energy balance among SW, LW, and sensible and latent heat fluxes at the surface.

I believe that looking at the same diagnostics (including LW changes) from the perspective of the TOA radiation alongside the surface would help to illuminate the underlying physical mechanisms responsible for the inter-model differences in the response to SRM.

=>As suggested, we added a discussion comparing the results at the surface and at TOA. We cannot treat LW radiation in the same manner as SW radiation, because we need to consider LW emission from atmosphere, surface, and clouds. Hence, we simply analysed the LW rapid adjustment in the clear-sky condition, which should represent effect of LW absorption by stratospheric sulphate aerosols. These discussions are added as Sections 4.1 and 4.2.

Main points:

Separation of cloud feedbacks from direct aerosol forcing of clouds

Clouds respond directly to forcing agents (e.g. aerosol, carbon dioxide, etc) and to changes in surface temperature. The IPCC (and field as a whole) includes the rapid cloud response to forcing agents in the "effective" radiative forcing whereas the cloud radiative changes due to surface temperature changes are generally classified as a radiative feedback. The present manuscript associates all the cloud changes with the feedback (equation 11) and I suspect much of what is called a cloud feedback is actually inter-model differences in the effective cloud forcing. This suspicion is based on two lines of evidence:

1.      The cloud radiative changes in figure 4 seem to coincide with the nearly step function changes in aerosol as opposed to the surface temperature changes. Panels E and C are the best examples. The cloud radiative changes ramp up almost immediately at 2020, before the surface temperature has decreased and return to near their unperturbed value almost immediately when the SRM stops at year 2070 even though the surface temperature takes longer to recover.

2.      The published cloud feedbacks differ in sign and magnitude from those found elsewhere in the literature for the same models. More fundamentally, the Authors conclude that cloud changes damp the response to geo-engineering whereas the models included in the study have been found to have positive net cloud feedbacks in response to CO2 (see Table 1 of Andrews et al. 2012 – Forcing. Feedbacks and climate sensitivity in the CMIP5 coupled atmosphere-ocean climate models) The comparison I'm making is unfair to Authors since I am comparing net cloud radiative impacts at the TOA to the surface SW impact. However, figure 3 of the above manuscript suggests a sign difference for at least the hadGEM3-ES model. Either way, the ensemble average negative cloud feedback suggested by the Authors seems at odds with the literature, is likely confused with the effective forcing and should be further analyzed (remove forcing, look at net

radiative impact, compare TOA and surface) since this result contradicts and confuses the existing literature.

A fairly straightforward solution to the above objections would be to compute the same fields outlined in equations 10-12 for each year of the simulation where the SRM is approximately constant (2025-2070 ish) and plot the radiative changes of each term versus the surface temperature change for all. As suggested by Gregory, the feedback is the slope of the linear best fit line and the effective forcing of each term is the y-intercept. This would also allow the Authors to calculate the impact of the aerosols on the shortwave absorption within the atmosphere which is alluded to in the discussion. I think this would appropriately isolate the effective forcing of clouds and the Authors might find the very interesting result that the inter-model differences in climate response to SRM is well correlated with effective forcing where the latter includes both the direct forcing of the aerosols and the rapid impact of the aerosols on the cloud radiative effect.

=>Thank you for the detailed explanation and suggestion. First of all, we misused the word "feedback" in the previous manuscript. We had used "feedback" for the sum of rapid response and feedback (in the meaning in the field of climate science). We have recognized we need to try to separate the rapid response and feedback. In the revised manuscript, we made plots similar to the Gregory plot as suggested by the reviewer. As the reviewer suspected, most part of $E_C$ is "rapid response (or adjustment)", which do not depend on ΔT, and the feedback part is not dominant.
Because the rapid adjustment of the cloud is caused by various processes (e.g., changes in atmospheric stability and water vapour distribution), its sign and amount can be different (or inconsistent) between $CO_2$ increased simulations, such as Andrews et al., and SRM simulations. In fact, Kravitz et al., (2013, JGR-Atmos, Vol. 118, pp.13087–13102) analysed GeoMIP-G1 experiment and showed a positive (sense of heating) SW cloud rapid adjustment of about 5.5 W m$^{-2}$, which is consistent with our results. We consider more detailed studies on cloud processes in SRM is needed. However, it is out of scope of this study.

We added description on the method at the end of Section 2, its result in the new Section 3.4, and some remarks on the difference between our results and Andrews et al. in Section 5 as follows:

Page 8, line 1–7, Section 2
To decompose the total reactions ($E_{WV}$, $E_C$, and $E_{SA}$) into rapid adjustments and feedbacks, a method similar to the Gregory plot (Gregory et al., 2004) is used. That is, the globally and annually averaged data of total reactions are plotted against that of ΔT ($\equiv T_{G4} - T_{RCP}$), and linear regression lines in the following forms are obtained by the least squares method.

$$E_{WV} = Q_{WV} - P_{WV}\Delta T, (15)$$

$$E_C = Q_C - P_C\Delta T, (16)$$

$$E_{SA} = Q_{SA} - P_{SA}\Delta T. (17)$$

Here, $Q_X$ ($X =$ WV, C, SA) denotes the rapid adjustment, $-P_X$ is the feedback parameter, and the overline denotes the global and annual average. This method is similar to the Gregory plot, but note that ΔT is the surface temperature difference between the G4 experiment and the RCP4.5 scenario

experiment, in which the anthropogenic radiative forcing depends on time and the simulated climate does not reach an statistically equilibrium state.

Page 11, line 17–34, Section 3.4

3.4 Decomposition of total reaction into rapid adjustment and feedback

The total reactions due to changes in water vapour amounts, cloud amounts, and surface albedo discussed in the previous two subsections are the sum of the rapid adjustment, which are independent of $\Delta T$, and the feedback, which depends linearly $\Delta T$. In this subsection, we attempt to decompose the rapid adjustment and the feedback using a so-called Gregory plot (Gregory et al., 2004). Figure 7 shows globally and annually averaged $E_{WV}$, $E_C$, and $E_{SA}$ as a function of averaged $\Delta T$ for each model. Now, we consider that a slope and a y-intercept show a feedback parameter and an amount of rapid adjustment, respectively, as shown by Eqs. (15)–(17); these values and correlation coefficients are shown in Table 2. The multi-model mean values are also shown.

There are no qualitative inter-model differences and each model has the following properties. $E_{WV}$ (orange ◇) shows high negative correlation with $\Delta T$, and the rapid adjustment and the feedback are clearly separated. In the multi-model mean, the rapid adjustment is –0.30 Wm–2 and the feedback parameter is –0.91 $Wm^{-2}K^{-1}$.

Unlike $E_{WV}$, $E_C$ (blue +) is not well-correlated with $\Delta T$. In addition, the spread of the blue plots is large. This means that the amount of rapid adjustment due to cloud changes varies largely, depending on the simulated state of ESM. The feedback of SW cloud radiative effect is not dominant in G4 experiment.

The y-intercept of $E_{SA}$ (green ✗) is almost zero, so that the rapid adjustment from the surface albedo change is negligible. The feedback parameter is 0.38 $Wm^{-2}K^{-1}$ in the multi-model mean, and the strength (absolute value) of the feedback is less than a half of that of $E_{WV}$.

Page 16, line 10–17, Section 5

The decomposition analysis has revealed that about 37 % (multi-model mean) of Ewv is explained by the rapid adjustment and the rest is the feedback. On the other hand, almost all amount of $E_C$ consists of the rapid adjustment, and a linear relationship between $E_C$ and $\Delta T$ for the global and annual mean was not obtained for any models. The cloud rapid adjustment in G4 deduced in this study is similar as found for G1 by Kravitz et al. (2013c) but disagree with that in the 4xCO2 experiment shown by Andrews et al. (2012). Because the rapid adjustment due to changes in clouds can be caused by various processes (e.g., changes in atmospheric stability), it is possible that the cloud rapid adjustment differs between SRM and global warming. More detailed studies on effect of clouds in SRM are required for the reduction of the uncertainty and for a better assessment of impact of the sulphate geoengineering on climate and human activities.

Use of the surface radiation budget

The surface energy budget is not closed with respect to the radiation and it is widely recognized that changes in surface radiation are balanced by turbulent energy fluxes with only small temperature adjustments. Generally, the radiative changes are viewed at a level where the system is closed with respect to radiation – either the tropopause or TOA. It is fair to challenge this paradigm and the surface radiative budget may be useful for geo-engineering but that point should be discussed and analyzed, not taken for granted as it is in the current manuscript.

=>We agree with the reviewer that the system is closed with respect to radiation at TOA and the energy budget or balance is generally viewed at TOA. However, this study

intends to estimate forcing and reactions to the surface SW radiation, which is important to consider the influence of SRM, especially for vegetation and human activities. Exploring full energy budget or balance is out of scope of this study. (We do not consider that it is much meaningful to struggle with the energy balance in G4 experiment; because, the baseline experiment RCP4.5 is a scenario experiment and does not reach statistically equilibrium state.) As the reviewer pointed out, the description for "why this study analyses surface SW radiation" in the previous manuscript was too short. In the revised manuscript we explained our motivation and purpose of this study at the end of Introduction and repeated at the end of Section 3.1 as follows:

Page 3, line 30–Page 4, line 13, Section 1
A simple procedure is used for quantifying the contributions of different types of SW rapid adjustments and feedbacks to the climate model behaviour to geoengineering with stratospheric sulphate aerosols. Here, a rapid adjustment is defined as a reaction to the SRM forcing without changes in globally averaged surface air temperature, whereas a feedback is defined as a reaction due to surface air temperature changes in the global mean induced by the SRM forcing (e.g., Sherwood et al., 2015). (Hereafter, the term "total reaction" refers to the sum of a rapid adjustment and a feedback.) In the recent studies of the climate change, rapid adjustments are included in forcing agents and the concept of effective radiative forcing is widely used. However, for the study of the sulphate geoengineering simulation, which is not well verified by observations and thus is expected to have many uncertainties, the separation of the direct forcing and total reactions is important to improve the simulation and to enhance the degree of understanding of the sulphate geoengineering by refining individual related processes. Many studies on climate energy balance have analysed changes in the net radiation flux at TOA, where the energy budget is closed by SW and longwave radiation (LW). However, in the geoengineering study, the radiative changes at the surface are also important, because vegetation, agriculture, and solar power generation for example will be strongly affected by radiative changes at the surface as well as surface temperature changes. Though the surface energy budget is balanced among SW, LW, sensible heat flux, and latent heat flux, Kleidon et al. (2015) showed that the latter three are mainly determined by the air and/or surface temperature. Hence, this study focuses on changes in surface air temperature and SW. The direct SW forcing to the surface are evaluated by considering the total reactions due to changes in water vapour amounts, cloud amounts, and surface albedo. Also, these total reactions are decomposed into adjustments and feedbacks, which indicate the rapid change just after injection of $SO_2$ and the change with globally averaged surface air temperature change by SRM, respectively. We provide results for both global and local effects, focusing on cross-model commonalities and differences.

Page 8, line 25–Page 9, line 3, Section 3.1
This figure shows a strong correlation between the mean $\Delta T$ and $\Delta F_{SURF}^{net}$; the correlation coefficient for the six filled symbols is 0.88. This strong correlation allows $\Delta F_{SURF}^{net}$ to be used as a measure of the SRM effects at least for $-1.1 < \Delta T < -0.2$ K, although the surface air temperature depends on the energy balance among SW, LW, and sensible and latent heat fluxes at the surface. Moreover, as described at the end of Section 1, it is important to explore the SW flux at the surface to estimate the effect of SRM on vegetation and human activities such as agriculture and solar power generation. Therefore, this study mainly focuses on SW at the surface and estimates the SRM forcing and the total reaction of SW due to changes in the water vapour amount, cloud amount, and surface albedo.

In particular, one place the surface radiative changes are less than useful is the interpretation of atmospheric solar absorption on the surface energy budget. As the atmosphere warms and moistens it absorbs more shortwave radiation that would have

otherwise mostly (since the majority of the Earth's surface is dark) been absorbed at the surface. As a result, less shortwave is fluxed to the surface, which would be seen as a cooling influence on the surface. Yet, in the column average, slightly more shortwave is absorbed. Since most of this additional shortwave absorption occurs in the lower troposphere, where water vapor is abundant, it is tightly coupled to the surface energy budget and will warm the surface even if the surface shortwave flux is reduced as a result. Radiative kernels estimate this feedback to result in +1.0 W m^-2 K^-1 more absorption in the atmospheric column and +0.3 W m^-2 K^-1 as measured at the TOA (Donohoe et al. 2014, Shortwave and longwave contributions to global warming under increasing CO2, PNAS). Therefore, the surface feedback would be deduced to be -0.7 W m^-2 K^-1 with the wrong sign and more than twice the magnitude of the changes at the TOA. In the very least, the manuscript should include similar diagnostics at the TOA to resolve this sign paradox and a discussion of these points to support the assertion that surface shortwave changes are a useful metric.

=>As we described above, we consider that it is important to explore surface SW radiation under SRM. We agree with reviewer's comment that the increase of the water vapour gives a positive feedback in total (i.e., sum of SW and LW effects), and in the case of geoengineering, the less water vapour may give cooling effect in total. We recognized that the use of word "heating" for the water vapour and cloud effects was misleading, because we only consider changes in SW at the surface. We changed the expression in the manuscript to describe that changes in water vapour and cloud amounts increase the SW radiation at the surface.
We also include the similar analysis at TOA and discuss the difference between the surface and TOA in the new Section 4.1. Especially, difference in the water vapour effect is notable and well explained. The explanation is consistent with the reviewer's above comment.

Page 13, line 23–Page 14, line 23 Section 4.1
**4.1 Difference between the surface and TOA**

This study has focused on the surface net SW because of its importance to human activities. However, the situation at TOA is also of interest. Now, we discuss how the measures used in this study differ when TOA is used for the analysis. The net SW at TOA can be written as
*[Equation 18]*
so that the direct forcing of SRM and the total reactions measured at TOA ($F_{SRM}^{TOA}$, $E_{WV}^{TOA}$, $E_C^{TOA}$, and $E_{SA}^{TOA}$) can be calculated in the same manner described in Section 2. Figure 10 shows their globally and temporally averaged values' dependencies on $\Delta T$. The difference of $F_{TOA}^{net}$ is also plotted.

The qualitative features of the measures other than $E_{WV}^{TOA}$ are same as the analysis at the surface shown in Fig. 6. The quantitative difference in the SRM forcing ($F_{SRM}^{TOA} - F_{SRM}$) is as small as $-0.047$ Wm$^{-2}$ (1.8 %) for the multi-model mean. In contrast, $|E_{SA}^{TOA}|$ is less than that of $|E_{SA}|$ by about 35 %. This is because the upward shortwave radiation that was reflected at the surface must pass the atmosphere being decreased by the absorption and reflection before reaching TOA. The difference of $E_C^{TOA} - E_C$ is 0.12 Wm$^{-2}$ (16.5 %) for the multi-model mean. Remember that the effect of the cloud amount change includes both changes in reflection rate ($R^{cl}$) and absorption rate ($A^{cl}$). The effect of a change in $R^{cl}$ should appear almost equally at the surface and TOA, as the case for the SRM forcing, because both $R^{cl}$ and $R^{cs}$ appear in the Eqs. (7) and (18) in the same way. Therefore, most of $E_C^{TOA} - E_C$ should be caused by the difference in how the change of the absorption rate affects the net SW at surface and that at TOA. This is discussed below.

The total reaction at TOA due to the change in water vapour amount shows a negative sign at TOA, which is opposite to that at the surface. This disagreement is attributed as follows: Surface cooling reduces the amount of water vapour in the atmosphere and the SW absorption rate decreases. Then, more incoming solar radiation reaches the surface, so that the decrease in water vapour amount brings increase of SW flux at the surface. On the other hand, when the SW absorption rate decreases, the more upwelling SW that was reflected at the surface pass through the atmosphere and reaches TOA. This leads to a cooling effect. Because the effect of decrease in the SW absorption rate is carried to TOA by the upwelling SW that was reflected at the surface by the rate of $\alpha$, $|E_{SA}^{TOA}|$ it is much less than $|E_{SA}|$. This does not mean that the change in water vapour is negligible for the energy budget at TOA, because we have not explored LW in this study. An analysis of LW rapid adjustment of clear-sky is discussed in the next subsection, but that of clouds and LW feedback is left as our future work.

From the above discussion, we have found that the effect of changes in atmospheric SW absorption rate appears differently between at the surface and at TOA (in its sign and amount), but that in reflection rate appears almost equally. The effect of change in the surface albedo is weaker at TOA than at the surface. We will bear these properties in our mind, when we discuss the influence of SRM on the energy budget of the climate system, which is usually considered at TOA, and human activities, which are mainly performed at the surface.

To play devil's advocate, it seems like most of correlation between the temperature response and net surface shortwave comes from the forcing. Is the use of net shortwave at the surface a better predictor of the temperature (statistically distinguishable) from that of forcing alone (surface or TOA)? The latter certainly would result in a stronger regression – and one more consistent with climate sensitivity—than using surface shortwave even if the correlation is slightly worse. More generally, what would the correlation be if one used forcing alongside published estimates of the model's climate sensitivity in response to CO2? It looks like the outlier from the strong relationship between forcing and response is the MIROC-CHEM-AMP which has a pronounced cloud feedback. As suggested above, I believe that cloud feedback is misidentified and is really an effective forcing associated with rapid cloud changes due to the direct impact of the aerosols. I think that calculating the effective forcing may offer a better correlation with the climate response than the net surface shortwave metric used in the manuscript.

=>We calculated ERF and found that ERF has a slightly better correlation than $\Delta F^{net}_{SURF}$, as the reviewer expected. However, finding the best predictor of $\Delta T$ is not the aim of this study. Although the ERF would be the better predictor of $\Delta T$, ERF is a sum of forcing due to the SW reflection by injected sulphate aerosols and the rapid responses of many other modelled physical processes in the ESMs. Therefore, it is difficult to explore, estimate, and compare contributions of each process to change in SW at the surface, by using ERF. Similarly, using the climate sensitivity to $CO_2$ increase estimated in the published papers will not give information about the contribution of each modelled process. We considered the description about the aim of this study was not enough, so that we added more description in Introduction as we showed above.
(The reviewer is correct in the point that the "cloud feedback" which we previously called was not a feedback but a rapid adjustment.)

---

## Referee Report (RR1)

I want to thank the Author's for the careful and thoughtful revisions and response to reviews. The revised manuscript is technically sound, well written and has addressed my previous concerns. I believe the conclusions are sound, novel and will make a strong impact on the existing literature on geoengineering. I continue to disagree with the emphasis on the surface energy budget (as opposed to TOA) and think the distinction between shortwave feedbacks and rapid response to SRM could be brought to the forefront of the manuscript in order to reach a broader climate dynamics audience. However, that emphasis is simply my opinion on the importance of the work and, as written, the focus on the changes in net shortwave at the surface is justified within the text. I elaborate on my suggestion below and otherwise have only minor suggestions of clarification. I think this manuscript is publishable nearly as is. Thanks again to the Author's for their efforts.

**Primary emphasis of the text.**

The main conclusion I draw from the results presented in this manuscript are: if one accounts for inter-model spread in the adjusted radiative forcing under SRM management, inter-model spread in surface temperature response is primarily due to differences in the shortwave forcing (as opposed to feedbacks). The spread in the adjusted forcing is a consequence of both the inter-model differences in the direct aerosol forcing and the rapid cloud response and the Authors have developed a novel technique for separating the direct forcing and rapid cloud response. The implication of this work is: the climate response to greenhouse gases and SRM can be understood from simply calculating the adjusted forcing and using the same climate sensitivity. While there is work to be done to understand the rapid cloud response – and where the observed system may lie within the substantial inter-model spread – I think this conclusion is a powerful result that I have not seen stated elsewhere. If robust, it would suggest that running long SRM model simulations is less important than understanding the short term cloud response to aerosol. I think this result is significant to the larger climate dynamics community and should be the central focus of the writing in the manuscript.

Again, this is just my opinion. As written, I think the manuscript addresses the more specific issue of how surface shortwave will change under SRM and this may be of more interest to the geo-engineering community that is reading this special issue. I personally think the results and implications are impactful to a larger community.  But I leave it to the Authors to make this determination since they know their audience better than I do.

**Specific points:**

Section 3.4:Comparison of feedbacks with those estimated from greenhouse forcing; The Authors find an ensemble average surface albedo feedback of +0.38 W m$^{-2}$K$^{-1}$ and a shortwave water vapor absorption feedback of -0.91 W m$^{-2}$K$^{-1}$ at the surface. It would be helpful to compare these feedbacks to those (ensemble average over the larger CMIP5 ensemble) reported in the literature under the response to greenhouse forcing. Numbers for the TOA are more prevalent in the literature and I suggest reporting TOA numbers here too (since the technique used in the manuscript gets at both). Specifically, these numbers seem consistent with the ensemble mean surface albedo feedback at TOA of +0.3W m$^{-2}$ K$^{-1}$ given by Bony et. al feedback review paper (since the surface number should be slightly bigger in

magnitude as discussed in the current manuscript) and the shortwave water vapor absorption of +1.0 W $m^{-2}K^{-1}$ in the atmospheric column and 0.3 W $m^{-2}K^{-1}$ at the TOA (implying -0.7 at the surface) given by Donohoe et al. 2014 (Shortwave and longwave contributions to global warming under increasing carbon dioxide, PNAS). I think the point that the SW feedbacks in response to SRM are consistent with those under greenhouse forcing is a powerful statement.

Page 14. Line 18. Worth pointing out that the 35% difference between surface albedo feedback as measured at the TOA compared to that at the surface is consistent with the (single pass) basic state atmospheric opacity (1-A-R).

Page 14. Line 20. The impact of shortwave atmospheric absorption on reflected SW at the TOA has 2 contributions in response to decreased water vapor under SRM: 1. Less absorption of shortwave above cloud top results in less shortwave reflected off the top of clouds reaching the TOA and 2. Less absorption above bright surfaces reduces the surface contribution to reflected SW at the TOA. The Author's are correct in pointing out that 2 dominates in the isotropic SW model used in this study by construction since the atmospheric absorption and reflection occur at the same vertical level (hence prohibiting the absorption above cloud top. This is a limitation of the model (I'm criticizing myself not the Authors here – Lol). But, in the real world it's possible that the SW water vapor absorption above cloud top is the dominant affect over the surface contribution. I don't know of and can't think of a way to back out the relative contributions, but it's worth discussing this point in the text.

Page 10. Line 32. Suggest rewording to "… implying that decreases in water vapor and cloud amount under SRM lead to more downwelling SW at the surface, counteracting the enhanced aersol reflection by SRM".

Page 10. Lines 33-34. The text suggest that decreases in water vapor are due to surface cooling only, where as I would have thought that the robust positive rapid response in Ewv (positive) implies that the atmosphere dries out before the surface cools due to reduced convection as soon as the aerosol is added to the atmosphere. If the specific humidity data are readily available, it would be nice to see if this logic holds (i.e. the direct response to SRM is a reduction in relative humidity).

Page 12. Line 5. Confusing intro sentence – I think it meant to say no qualitive differences in Ewv. I suggest the following: "WV changes lead to a robust ensemble average increase in surface SW under SRM. This increase in surface SW is due to both the rapid decrease in WV in direct response to SRM (+0.3 W $m^{-2}$) and the WV decrease due to surface cooling (-0.91 W $m^{-2}$ $K^{-1}$ – note the negative sign corresponds to an increase in surface shortwave with cooling)."

NOTE: I think there was a sign error in the manuscript, since the two should have opposite signs if one is cited as a feedback.

Page 13. Line 28-29. Is the robust cloud reduction in the Western Pacific part of the feedback or rapid response to SRM?   I don't know the literature well enough to evaluate if this spatial pattern is consistent with the cloud response one would expect from the cooling, the rapid adjustment, or is all together different.

Section 4.2. This whole section confused me. Shouldn't the impact of stratospheric changes in LW emissivity be evaluated from radiative changes at the tropopause after stratospheric temperature adjustment? The argument presented is instead written in terms of TOA radiative response and I'm not sure the sign of the impact is even the same. Additionally, the second paragraph talks about evaluating the impact of stratospheric LW adjustment from the rapid response where I would have though the LW rapid adjustment would have been dominated by the substantial changes in clouds in the troposphere.

Page 17. Fist line. I would make this statement stronger – one should expect that the rapid adjustment in response to SRM is very different from that due to CO2 because the vertical distribution of the direct forcing is very different (partitioning of radiative anomalies between the surface and atmospheric column.

---

## Referee Report (RR2)

General Comments:

The paper estimates the shortwave (SW) radiative forcing at the surface from the injection of sulfur in to the stratosphere. Their method is applied to the GeoMIP G4 models where 5 Mt SO2 is injected annually in a transient RCP4.5 scenario. A single layer model for SW radiative transfer is used to calculate the forcing of the aerosol layer, rapid adjustments and reactions from changes in water vapor, cloud amount, and surface albedo. The simple model is a simplification but it allows to differentiate between different effects and provide useful information relevant to human activities and interests and the assumptions made are clearly stated. The paper will be a valuable addition to current literature on the subject after some minor revisions.

A number of scatter plots are included in the analysis where means over 3 decades are used. These single values should be split into decadal means such that you have three values to base the analysis on instead of one. This will not only provide more data points for the regression, but also indicate more clearly the steadiness of the differences in climates over this period of the simulations.

> Throughout the manuscript there is content that belongs only in the figure captions and not in the main text. E.g. "(shown by red symbols)" and similar is unnecessary in the main text; more interesting to read about the meaning of the results in the figures than color coding etc.

You should make some concluding remark on what your findings imply for human activities at the surface, as this is your initial motivation for the paper, and put your findings into a wider context.

Specific comments:

- The brief 'review' of solar constant experiments p. 2 lines 6-14 is not very informative nor important for this paper. I suggest removing it.
- P2, l19: 'arctic' should be 'Arctic'.
- P2 l17-26: you mention a number of sulfate aerosol papers, but you do not say how or why they or their findings are relevant or important. Either just list them and say they cannot be compared or find something relevant. Otherwise, it is merely "stuffing".
- P3 l18-32: you discuss factors explaining the spread in the climate response in G4. It would be interesting if you could also note down the spread in the RCP4,5 models (for the same models as in the G4 study referred to) for comparison. Is the spread larger for G4?
- P3 l34: 'behaivour' -> 'behaviour'.
- P3 l34 and P4 l4, l8: you refer to several, recent and many studies; please include some citations.
- P4 l11: radiation at the surface is also important for oceanic processes like biogeochemistry and ocean carbon cycle. This should be mentioned too. Human activities, like fisheries, might also be affected at sea.
- P5 l4: "In addition, the microphysics of the tropospheric sulphate aerosols is not calculated in MIROC-ESM-CHEM-5 AMP to avoid drift in the simulated climate." Why would this cause a drift? Please note a brief explanation in the text.
- P5 l6: The experiment has also been done with NorESM – and maybe IPSL and MPI-ESM(?). CSIRO-Mk3L has done 'G4S', which is not G4, hence no need to mention.
- P6, l12: 'cloud effects': do you mean feedbacks due to clouds?
- P7: on what time-scales are these assumptions valid?

- P7: $E_c$ –cloud amount: this can also be output as a variable from the models. Do any of the models account for injected sulfate interactions with clouds? e.g. Kuebbeler et al. ?
- Figure 2: Why was year 2020 of RCP4.5 used as 'baseline'? Why not center the baseline on a 5 or 10 year period around 2020, to account for variability of the climate considering 2020 might be a particularly warm or cold year. It would be a cleaner comparison, particularly for BNU-ESM who has a lot of year-to-year variability.
- P8 l11-12: "For all models, T in G4 decreases or remains at the 2020 level for a few decades and begins increasing from around 2040 or earlier": This is a bit inaccurate. For models a-c, yes, but not really for the MIROC models.
- Figure 2 discussion; you're applying a fixed magnitude forcing every year, whilst the anthropogenic forcing in RCP4.5 keeps increasing. Hence the limitation to the cooling evolution you describe on page 8. Include some comment on this in the discussion of the results.
- Why is there hardly any cooling in MIROC-ESM-CHEM?
- P8 l 17: " … and then 25 returns to the RCP4.5 level in each model": HadGEM2-ES has not returned to RCP4.5 levels at the end of the run as this model has a larger temperature response to the forcing. You may include the comment that the stronger the temperature response to the forcing is, the longer it takes to return to the otherwise temperature path and the rate of change of the climate system to sudden termination would become more drastic.
- Figure 3: You find a correlation coefficient of 0.88. Please include the regression line in the figure. Also you select data from three decades as the temperature differences between G4 and RCP4.5 are steady over this period. I suggest you therefore use one value from each decade in every run in the figure. Then you have more data points and information.
- Figure 4: Fsrm varies by ~1 Wm-2 throughout G4 in HadGEM2-ES. Why is this so much more variable than the other model that accounts for formation and transportation of the aerosols (MIROC-ESM-CHEM-AMP)?
- Section 3.3: Considering the focus on the last three decades you should break the 3 decade mean into 1 mean for each decade, as mentioned before. The scatter plots have to be updated accordingly.
- "HadGEM2-ES does not calculate the sulphate aerosols in the tropospheric and stratosphere: separately" is mentioned several times. Please explain. Not clear what is meant by this at all.
- P10, l23-24: please say "… considered having performed the G4 simulation" – or similar at end of sentence. I.e. point to G4.
- P11, end of section 3.3: can you explain more clearly why the cloud amount is strongly dependent on the initial conditions? To my belief the clouds are even more so dependent on the cloud parameters. (perturbed initial conditions ensembles versus perturbed physical parameter ensembles.)
- Figure 7: I am not sure how much sense it makes to draw regression lines for Ec. There is clearly little correlation.
  Also; which models years are included in the figure? 2020 – 2069? Please make a note in caption.
- Figure 7 cont.: Have you tried to plot annual means for years 1-10 of the simulations and then decadal means for the remaining 4 decades? This might be more representative for gauging fast vs slow response.
- Section 3.5 Figure 8: good if you could remind readers at this stage if each model has been weighted by number of ensemble member in these figures.

- Figure 8 clearly indicates that something is going on with the South Pacific Convergence Zone. Detailed study of this is beyond the topic of your study; however, if this is discussed elsewhere in the literature, it would be great to point to it in the text.
- P13, l25: "However, the situation at TOA is also of interest." Please say why.
- Section 4.2: doesn't the LW heating from the aerosols also increase water vapor in the stratosphere, contributing to ozone losses? This aspect might be mentioned here briefly, as stratospheric O3 amounts do indeed impact human activities at the surface.

---

## Author Response (AR2)

Dear Editor Prof. Ulrike Lohmann

Please find below our responses to the reviewers' comments and a marked-up manuscript including supplemental figures and an added table. We have addressed all the comments raised by both reviewers, and we believe our manuscript has been improved.

Thank you very much for your consideration.

Sincerely, Hiroki Kashimura et al.

**"Shortwave radiative forcing, rapid response, and feedback to the surface by sulphate geoengineering: Analysis of the Geoengineering Model Inter-comparison Project G4 scenario" by Hiroki Kashimura et al.**

Response to Dr. Aaron Donohoe

Dear Dr. Donohoe

We thank Dr. Donohoe for a careful review again and constructive comments. Please find below the authors' response. In this reply we denote referee's comments and questions using blue; our responses are in black and relevant text in the manuscript in Times font with changes shown in red.

I want to thank the Author's for the careful and thoughtful revisions and response to reviews. The revised manuscript is technically sound, well written and has addressed my previous concerns. I believe the conclusions are sound, novel and will make a strong impact on the existing literature on geoengineering. I continue to disagree with the emphasis on the surface energy budget (as opposed to TOA) and think the distinction between shortwave feedbacks and rapid response to SRM could be brought to the forefront of the manuscript in order to reach a broader climate dynamics audience. However, that emphasis is simply my opinion on the importance of the work and, as written, the focus on the changes in net shortwave at the surface is justified within the text. I elaborate on my suggestion below and otherwise have only minor suggestions of clarification. I think this manuscript is publishable nearly as is. Thanks again to the Author's for their efforts.

**Primary emphasis of the text.**

The main conclusion I draw from the results presented in this manuscript are: if one accounts for inter-model spread in the adjusted radiative forcing under SRM management, inter-model spread in surface temperature response is primarily due to differences in the shortwave forcing (as opposed to feedbacks). The spread in the adjusted forcing is a consequence of both the inter-model differences in the direct aerosol forcing and the rapid cloud response and the Authors have developed a novel technique for separating the direct forcing and rapid cloud response. The implication of this work is: the climate response to

greenhouse gases and SRM can be understood from simply calculating the adjusted forcing and using the same climate sensitivity. While there is work to be done to understand the rapid cloud response – and where the observed system may lie within the substantial inter-model spread – I think this conclusion is a powerful result that I have not seen stated elsewhere. If robust, it would suggest that running long SRM model simulations is less important than understanding the short term cloud response to aerosol. I think this result is significant to the larger climate dynamics community and should be the central focus of the writing in the manuscript.

Again, this is just my opinion. As written, I think the manuscript addresses the more specific issue of how surface shortwave will change under SRM and this may be of more interest to the geo-engineering community that is reading this special issue. I personally think the results and implications are impactful to a larger community. But I leave it to the Authors to make this determination since they know their audience better than I do.

=> Thank you very much for the suggestion. We think it would be very fruitful to write a companion piece talking about understanding the climate response from this sort of 'dual forcing' experiment. So as not to distract from the main point of the present paper, which we would prefer to keep focused on SRM, we reserve such discussions for future work.

**Specific points:**

Section 3.4:Comparison of feedbacks with those estimated from greenhouse forcing; The Authors find an ensemble average surface albedo feedback of +0.38 W $m^{-2}K^{-1}$ and a shortwave water vapor absorption feedback of -0.91 W $m^{-2}K^{-1}$ at the surface. It would be helpful to compare these feedbacks to those (ensemble average over the larger CMIP5 ensemble) reported in the literature under the response to greenhouse forcing. Numbers for the TOA are more prevalent in the literature and I suggest reporting TOA numbers here too (since the technique used in the manuscript gets at both). Specifically, these numbers seem consistent with the ensemble mean surface albedo feedback at TOA of +0.3W $m^{-2} K^{-1}$ given by Bony et al. feedback review paper (since the surface number should be slightly bigger in magnitude as discussed in the current manuscript) and the shortwave water vapor absorption of +1.0 W $m^{-2}K^{-1}$ in the

atmospheric column and 0.3 W m$^{-2}$K$^{-1}$ at the TOA (implying -0.7 at the surface) given by Donohoe et al. 2014 (Shortwave and longwave contributions to global warming under increasing carbon dioxide, PNAS). I think the point that the SW feedbacks in response to SRM are consistent with those under greenhouse forcing is a powerful statement.

=> Following this suggestion, we have calculated feedback parameters at TOA and compared them with Soden and Held (2006) (which Bony et al referred) and Donohoe et al. (2014). We found that the SW feedback of surface albedo is consistent with those under greenhouse forcing, but that of WV is about a half. This discussion is added at the end of Section 4.1 as follow:

P.14 L.27–35 "To fairly compare feedback parameters in G4 with those under greenhouse gas forcing, we decompose the total reactions at TOA into rapid adjustment and feedback in the same manner that we performed in Section 3.4. The rapid adjustment and feedback parameters calculated at TOA are listed in Table S1. The multi-model-averaged feedback parameter of surface albedo in G4 is 0.27 W m$^{-2}$ K$^{-1}$. This value is close to the surface albedo feedback parameter of 0.26 W m$^{-2}$ K$^{-1}$ in A1B scenario Soden and Held (2006) and that of 0.30 W m$^{-2}$ K$^{-1}$ in the quadrupled $CO_2$ experiment Donohoe et al. (2014). On the other hand, the multi-model-averaged feedback parameter of water vapour in G4 is 0.15 W m$^{-2}$ K$^{-1}$ and that (for SW at TOA) in quadrupled $CO_2$ experiment is 0.30 W m$^{-2}$ K$^{-1}$. These comparisons suggest that the SW feedback of surface albedo under sulphate geoengineering is consistent with that under greenhouse gas forcing, whereas that of water vapour is about a half of that under greenhouse gas forcing."

Page 14. Line 18. Worth pointing out that the 35% difference between surface albedo feedback as measured at the TOA compared to that at the surface is consistent with the (single pass) basic state atmospheric opacity (1-A-R).

=> Thanks to this comments, we realized that the ratio $E_{SA}{}^{TOA}/E_{SA}$ can be mathematically written as (1-$A$-$R$)/(1-$R$). And of course, this ratio is consistent with the 35% difference. We added a word and a sentence as follows:
P.14 L.3–5 "This is mainly because the upward shortwave radiation that was reflected at the surface must pass the atmosphere being decreased by absorption and reflection before reaching the TOA. The ratio $E_{SA}{}^{TOA}/E_{SA}$, of course, agrees with (1-$R^{as}{}_{RCP}$-$A^{as}{}_{RCP}$)/(1-$R^{as}{}_{RCP}$), which can be obtained through algebraic manipulation."

Page 14. Line 20. The impact of shortwave atmospheric absorption on reflected SW at the TOA has 2 contributions in response to decreased water vapor under SRM: 1. Less absorption of shortwave above cloud top results in less shortwave reflected off the top of clouds reaching the TOA and 2. Less absorption above bright surfaces reduces the surface contribution to reflected SW at the TOA. The Author's are correct in pointing out that 2 dominates in the isotropic SW model used in this study by construction since the atmospheric absorption and reflection occur at the same vertical level (hence prohibiting the absorption above cloud top. This is a limitation of the model (I'm criticizing myself not the Authors here – Lol). But, in the real world it's possible that the SW water vapor absorption above cloud top is the dominant affect over the surface contribution. I don't know of and can't think of a way to back out the relative contributions, but it's worth discussing this point in the text.

If we understand the referee's point correctly, we should discuss the relative contribution of absorption above the cloud and that above the surface. As the referee mentioned, our single layer model assumes reflection and absorption at the same level and at the same time, and this model cannot consider the order of reflection and absorption. The SW flux that reflected back to TOA by clouds and the atmosphere without reaching the surface is independent on the change of atmospheric absorption rate in this model.

Thanks to this referee's comment, we realized that $E_{\text{wv}}^{\text{TOA}}$ could be underestimated, and we added a note mentioning about this as follows:

P.14 L.17–20 "Note that, in our single layer model, SW absorption above the clouds is not included, so that upwelling SW at TOA reflected by the clouds without reaching the surface is independent of the absorption rate. Therefore, $E_{\text{wv}}^{\text{TOA}}$ could be underestimated, and the change in water vapour may not be negligible for the energy budget at TOA."

Page 10. Line 32. Suggest rewording to "... implying that decreases in water vapor and cloud amount under SRM lead to more downwelling SW at the surface, counteracting the enhanced aerosol reflection by SRM".

=> We modified the sentence as suggested.

P.9 L19–20 "Both $E_{\text{wv}}$ and $E_{\text{C}}$ are positive, implying that the decreases in water vapour

and cloud amounts under SRM lead to more downwelling SW at the surface, counteracting the enhanced aerosol reflection by SRM.”

Page 10. Lines 33-34. The text suggest that decreases in water vapor are due to surface cooling only, where as I would have thought that the robust positive rapid response in Ewv (positive) implies that the atmosphere dries out before the surface cools due to reduced convection as soon as the aerosol is added to the atmosphere. If the specific humidity data are readily available, it would be nice to see if this logic holds (i.e. the direct response to SRM is a reduction in relative humidity).

=> As you pointed out, because the rapid response of $E_{WV}$ is positive, the temperature drop is not the only reason for the water vapour decrease. We modified the sentence as follows:
P.9 L20–21 “One reason for the decrease of water vapour is the temperature reduction, which results in less evaporation (Kravitz et al., 2013).”
Then, we revised sentences in Section 3.4. Please see the next response.
At this time, the specific humidity data is not available for all models, so we reserve such analysis for future work.

Page 12. Line 5. Confusing intro sentence – I think it meant to say no qualitive differences in Ewv. I suggest the following: “WV changes lead to a robust ensemble average increase in surface SW under SRM. This increase in surface SW is due to both the rapid decrease in WV in direct response to SRM (+0.3 W m$^{-2}$) and the WV decrease due to surface cooling (-0.91 W m$^{-2}$ K$^{-1}$ – note the negative sign corresponds to an increase in surface shortwave with cooling).”

=> To avoid confusion, we revised the related sentences as follows:
P.11 L.24–28 “$E_{WV}$ shows high negative correlation with $\Delta T$ in all models, and the rapid adjustment (+0.30 W m$^{-2}$ in multi-model mean) and the feedback (-0.91 W m$^{-2}$ K$^{-1}$) are clearly separated. That is, the surface SW increase due to less water vapour is caused by both the rapid direct response to SRM and the surface cooling; note that the negative sign corresponds to an increase in surface SW with cooling. The rapid decrease of the water vapour would result from reduced convection due to change in vertical temperature profile caused by the injected stratospheric sulphate aerosols.”

NOTE: I think there was a sign error in the manuscript, since the two should have opposite signs if one is cited as a feedback.

=> You are correct. –0.30 should be +0.30.

 Is the robust cloud reduction in the Western Pacific part of the feedback or rapid response to SRM? I don't know the literature well enough to evaluate if this spatial pattern is consistent with the cloud response one would expect from the cooling, the rapid adjustment, or is all together different.

=> Because $E_C$ is calculated from the difference between G4 and RCP4.5, all features in $E_C$ are caused by SRM directly or indirectly. The Gregory-plot analysis showed that the rapid response is dominant in $E_C$ (though it varies a lot). Detailed analysis is needed to detect what physical effect is dominant in the Western Pacific, but exploration for a specific region is out of scope of this study.

Section 4.2. This whole section confused me. Shouldn't the impact of stratospheric changes in LW emissivity be evaluated from radiative changes at the tropopause after stratospheric temperature adjustment? The argument presented is instead written in terms of TOA radiative response and I'm not sure the sign of the impact is even the same. Additionally, the second paragraph talks about evaluating the impact of stratospheric LW adjustment from the rapid response where I would have though the LW rapid adjustment would have been dominated by the substantial changes in clouds in the troposphere.

=> We carefully reconsider the role and importance of this section in our manuscript. Then we realized that Section 4.2 provides a little information on LW rapid adjustment of WV but we cannot evaluate its importance compared with SW total reactions and LW feedbacks (which cannot be evaluated easily). We decided to remove this section. We believe keeping focus on SW analysis will benefit readers.

 I would make this statement stronger – one should expect that the rapid adjustment in response to SRM is very different from that due to CO2 because the vertical distribution of the direct forcing is very different (partitioning of radiative anomalies between the surface and atmospheric column.

=> Thank you for your suggestion. We changed the sentence as follows:
P.16 L7–9 "One should expect that the rapid adjustment in response to SRM is different from that due to $CO_2$, because the vertical distribution of the direct forcing is different

and the cloud rapid adjustment can be caused by various processes (e.g., changes in atmospheric stability)."

We also corrected the following typos in the manuscript.

- 2070 => 2069 (The SRM term had to be written as year 2020 to 2069 not 2070)
- $E^{TOA}_{SA}$ => $E^{TOA}_{WV}$, $E_{SA}$ => $E_{WV}$ (P.14 L.17, typo)

**"Shortwave radiative forcing, rapid response, and feedback to the surface by sulphate geoengineering: Analysis of the Geoengineering Model Intercomparison Project G4 scenario" by Hiroki Kashimura et al.**

Response to the Referee #3

Dear Referee

We thank the referee for a careful review and constructive comments. Please find below the authors' response. In this reply we denote referee's comments and questions using blue; our responses are in black and relevant text in the manuscript in $\mathrm{Times}$ $\mathrm{font}$ with changes shown in red.

General Comments:

The paper estimates the shortwave (SW) radiative forcing at the surface from the injection of sulfur in to the stratosphere. Their method is applied to the GeoMIP G4 models where 5 Mt SO2 is injected annually in a transient RCP4.5 scenario. A single layer model for SW radiative transfer is used to calculate the forcing of the aerosol layer, rapid adjustments and reactions from changes in water vapor, cloud amount, and surface albedo. The simple model is a simplification but it allows to differentiate between different effects and provide useful information relevant to human activities and interests and the assumptions made are clearly stated. The paper will be a valuable addition to current literature on the subject after some minor revisions.

A number of scatter plots are included in the analysis where means over 3 decades are used. These single values should be split into decadal means such that you have three values to base the analysis on instead of one. This will not only provide more data points for the regression, but also indicate more clearly the steadiness of the differences in climates over this period of the simulations.

=> We carefully considered your suggestion. After splitting the 30 year means into three 10-year means, we obtained the following figure (ensemble runs are

omitted):

[Figure]

We find that this figure does not add additional information beyond what we obtained from the 30-year mean, and it has the drawback of being substantially busier. Note that, in addition to the above figure, 45 data points for each variable are needed to plot ensemble members (15 runs x 3 decades), which are required in the text. Moreover, the baseline experiment (RCP4.5) is transient, so treating the three different decadal means requires some care. We find the 30-year mean to be simpler. Note that time-dependence/steadiness is shown in Fig. 4.

Throughout the manuscript there is content that belongs only in the figure captions and not in the main text. E.g. "(shown by red symbols)" and similar is unnecessary in the main text; more interesting to read about the meaning of the results in the figures than color coding etc.

=> Following your suggestion, we removed them form the main text. Note that words pointing which Figure to see, E.g., "(Fig. 9b)", are remained, because they benefit readers to follow the descriptions and discussions.

You should make some concluding remark on what your findings imply for human activities at the surface, as this is your initial motivation for the paper, and put your findings into a wider context.

=> Our biggest finding is that the large range (uncertainty) of SRM forcing in the simulated sulphate geoengineering. So that, we consider it is not the stage to make a direct conclusion for influence on human activities, even it was our initial motivation. Most important thing is to take note this large uncertainty in considering an environmental assessment of the sulphate geoengineering. To mention this, we added the following sentence:

P.15 L.26–27 "From a point of view of an environmental assessment of sulphate geoengineering, we note that there is such large uncertainty in the simulated SRM forcing."

Specific comments:

● The brief 'review' of solar constant experiments p. 2 lines 6-14 is not very informative nor important for this paper. I suggest removing it.
  => We removed almost all of these lines and just remained one sentence with references.

● P2, l19: 'arctic' should be 'Arctic'.
  => We corrected it as suggested.

● P2 l17-26: you mention a number of sulfate aerosol papers, but you do not say how or why they or their findings are relevant or important. Either just list them and say they cannot be compared or find something relevant. Otherwise, it is merely "stuffing".
  => We agree that the sentences were too long and less informative. Here, what we intended to say is "There are many studies but comparison is difficult", so that listing is necessary enough. We revised to list them in a new Table 1 to avoid long sentences.

● P3 l18-32: you discuss factors explaining the spread in the climate response in G4. It would be interesting if you could also note down the spread in the RCP4,5 models (for the same models as in the G4 study referred to) for comparison. Is the spread larger for G4?
  => This comparison was done in the refereed paper (Yu et al. 2015). The spread for RCP4.5 is ±0.21 and G4's spread is larger. We added a sentence

as follows:

P.3 L.15–16 "This spread is larger than that of ± 0.21 K of temperature increase in RCP4.5 scenario for the same models."

- P3 l34: 'behaivour' -> 'behaviour'.
  => We corrected it as suggested.

- P3 l34 and P4 l4, l8: you refer to several, recent and many studies; please include some citations.
  => We added Andrews (2014) and Zhang et al. (2016) for (previous) P3 L34, Trenberth et al (2014) and Wild et al. (2014) for P4 L4, and Campillo et al. (2012) for P4 L8.

- P4 l11: radiation at the surface is also important for oceanic processes like biogeochemistry and ocean carbon cycle. This should be mentioned too. Human activities, like fisheries, might also be affected at sea.
  => Thank you for the suggestion. We added a sentence as follows:
  P.3 L.32–33 "Surface SW is also important for ocean carbon cycle and fisheries through changes in amounts of phytoplankton (Miller et al., 2006)."

- P5 l4: "In addition, the microphysics of the tropospheric sulphate aerosols is not calculated in MIROC-ESM-CHEM-AMP to avoid drift in the simulated climate." Why would this cause a drift? Please note a brief explanation in the text.
  => We modified the sentence and added a brief explanation as follows:
  P.4 L.28–31 "Because the newly developed microphysics module for sulphate aerosols in MIROC-ESM-CHEM-AMP was not well-tested or tuned for the troposphere by a long-term climate simulation yet, it may cause unexpected drift in the simulated climate due to changes in concentration and/or distribution of the tropospheric sulphate aerosols. To avoid such situation, the sulphate aerosol microphysics was calculated only in the stratosphere in G4 and RCP4.5."

- P5 l6: The experiment has also been done with NorESM – and maybe IPSL and MPI-ESM(?). CSIRO-Mk3L has done 'G4S', which is not G4, hence no need to mention.
  => IPSL-CM5A-LR and NorESM1-M performed the experiment but they have some issues in calculation of LW effects of sulphate aerosols as written in Ferraro and Griffiths (2016, ERL). To the best of our knowledge, MPI-ESM

did not perform G4. We added the following sentence and removed the sentence on CSIRO-Mk3L.

P.5 L.2–3 "IPSL-CM5A-LR (Dufresne et al., 2013) and NorESM1-M (Bentsen et al., 2013; Iversen et al., 2013) have some issues in calculation of the LW effects of the sulphate aerosols (Ferraro and Griffiths, 2016)".

- P6, l12: 'cloud effects': do you mean feedbacks due to clouds?
  => Here, the word "cloud effects" means effects of cloud on radiative transfer, which is simply defined by all-sky value minus clear-sky value of model outputs (radiative fluxes) and calculated R, A and $a$. To avoid confusion, we revised the sentence as follows:
  P.6 L.8 "Defining the cloud effects on radiative transfer for a variable $X$ by"

- P7: on what time-scales are these assumptions valid?
  => We consider these assumptions are independent of time-scale, at least up to the period of G4 experiment.

- P7: EC –cloud amount: this can also be output as a variable from the models. Do any of the models account for injected sulfate interactions with clouds? e.g. Kuebbeler et al. ?
  => Some do, and some don't. The strength of the indirect effects in models (at least for models that include such effects) varies considerably, and we didn't want to distract from the main point of the paper by getting into such discussions. We reserve that for future work.

- Figure 2: Why was year 2020 of RCP4.5 used as 'baseline'? Why not center the baseline on a 5 or 10 year period around 2020, to account for variability of the climate considering 2020 might be a particularly warm or cold year. It would be a cleaner comparison, particularly for BNU-ESM who has a lot of year-to-year variability.
  => As suggested, we updated Fig. 2 to use baseline for mean of 5 years (2018-2022).

- P8 l11-12: "For all models, T in G4 decreases or remains at the 2020 level for a few decades and begins increasing from around 2040 or earlier": This is a bit inaccurate. For models a-c, yes, but not really for the MIROC models.
  => The word "2020 level" was used for roughly same ( ~ ±0.3 K), but as the referee mentioned the range of "level" depends on readers. We revised the

sentence as follows:

P.8 L9–10 "For all models, $T$ in G4 decreases or remains within +0.3 K from the baseline for a few decades and begins increasing from around 2040 or earlier, whereas $T$ in RCP4.5 steadily increases."

● Figure 2 discussion; you're applying a fixed magnitude forcing every year, whilst the anthropogenic forcing in RCP4.5 keeps increasing. Hence the limitation to the cooling evolution you describe on page 8. Include some comment on this in the discussion of the results.
=> We included a sentence as follows:
P.8 L.14–15 "This is simply because the anthropogenic forcing in RCP4.5 keeps increasing but the amount of $SO_2$ injection per year is fixed in G4."
Here, we avoid to say "the magnitude of SRM forcing is fixed" at this point (though it is naturally expected), because we are showing that the evolution of SRM forcing in Fig. 4, which is almost constant during the SRM period.

● Why is there hardly any cooling in MIROC-ESM-CHEM?
=> This is what we explored in this study, and the main reason is the weakness of $F_{SRM}$ in MIROC-ESM-CHEM as shown in Fig. 5.

● P8 l 17: " … and then returns to the RCP4.5 level in each model": HadG-EM2-ES has not returned to RCP4.5 levels at the end of the run as this model has a larger temperature response to the forcing. You may include the comment that the stronger the temperature response to the forcing is, the longer it takes to return to the otherwise temperature path and the rate of change of the climate system to sudden termination would become more drastic.
=> We had used the word "RCP4.5 level" very roughly and it was inaccurate. We revised this sentence as follow:
P.8 L.15–16 "In addition, after halting SRM at 2070, $T$ increases rapidly and then returns to or approaches the RCP4.5 level in each model."
As you commented, it might be true that the stronger temperature cooling by SRM needs longer time to return to RCP4.5 path. However, Fig. 2 is an inter-model comparison, so that model properties (inertia or sensitivity) may also affect the speed of temperature rising. And we cannot distinguish them from our results (at least without many additional analyses). In addition, SRM

termination is not focused in this study. From above reasons, we avoid to state something more on temperature rising at SRM termination.

- Figure 3: You find a correlation coefficient of 0.88. Please include the regression line in the figure. Also you select data from three decades as the temperature differences between G4 and RCP4.5 are steady over this period. I suggest you therefore use one value from each decade in every run in the figure. Then you have more data points and information.
  => Regression line was included as suggested.
  As we explained in the response to the referee's "general comments" we decided not to split 30-year mean to three 10-year mean.

- Figure 4: Fsrm varies by ~1 Wm-2 throughout G4 in HadGEM2-ES. Why is this so much more variable than the other model that accounts for formation and transportation of the aerosols (MIROC-ESM-CHEM-AMP)?
  => Both HadGEM2-ES and MIROC-ESM-CHEM-AMP calculate formation and transportation of the sulphate aerosols, but details are different. We explained this in the text as follows:
  P. 4 L.23–31 "In HadGEM2-ES, the tropospheric aerosol scheme and the associated microphysical properties (Bellouin et al., 2011) is simply extended into the stratosphere. Modifications to the stratospheric aerosol size distribution have been applied in subsequent HadGEM2-ES studies (Jones et al., 2016ab), but have not been applied here. In MIROC-ESM-CHEM-AMP, the microphysics module for stratospheric sulphate aerosols treats them in three modes as shown in Table 2 in Sekiya et al. (2016); however, to calculate radiative processes on the aerosols, a particle size of 0.243 um is assumed for simplification. Because the newly developed microphysics module for sulphate aerosols in MIROC-ESM-CHEM-AMP was not well-tested or tuned for the troposphere by a long-term climate simulation yet, it may cause unexpected drift in the simulated climate due to changes in concentration and/or distribution of the tropospheric sulphate aerosols. To avoid such situation, the sulphate aerosol microphysics was calculated only in the stratosphere in G4 and RCP4.5."
  We imagine that such difference in the implementation of sulphate aerosol microphysics may cause the difference in the magnitude of $F_{SRM}$'s fluctuation. However, at this time, we do not have any evidences or logics that support such imagination. Hence, in the manuscript, we cannot state why this happens.

- Section 3.3: Considering the focus on the last three decades you should break the 3 decade mean into 1 mean for each decade, as mentioned before. The scatter plots have to be updated accordingly.

  => Again, as we explained in the response to the referee's "general comments" we decided not to split 30-year mean to three 10-year mean.

- "HadGEM2-ES does not calculate the sulphate aerosols in the tropospheric and stratosphere: separately" is mentioned several times. Please explain. Not clear what is meant by this at all.

  => In HadGEM2-ES, the sulphate aerosol microphysics module (and its physical parameters) is not tuned for stratosphere. They use the same routine with the same parameters for both troposphere and stratosphere. Accordingly, the output variables related to the sulphate aerosols such as AOD of an air column is the sum of all vertical levels. So that, the available sulphate AOD for HadGEM2-ES is not the AOD for **stratospheric** sulphate aerosols. To make clear this point, we modified the sentence as follows: P.10 L11–14 "Note that the above value for HadGEM2-ES is the difference (G4 - RCP4.5) in the sulphate AOD for both troposphere and stratosphere. This is because HadGEM2-ES used the same microphysics calculation of the sulphate aerosols with the same aerosol size distribution in both the troposphere and the stratosphere; sulphate AOD solely for the stratosphere is not available for HadGEM2-ES."

- P10, l23-24: please say "… considered having performed the G4 simulation" – or similar at end of sentence. I.e. point to G4.

  => We modified the sentence as follows:
  P.10 L.21–23 "Pitari et al. (2014) have shown that, in the G4 simulation, SW radiative forcing at the tropopause calculated off-line by a radiative transfer code (Chou and Suarez, 1999; Chou et al., 2001) varies from around 2.1 to 1.0 W m$^2$ between the models."

- P11, end of section 3.3: can you explain more clearly why the cloud amount is strongly dependent on the initial conditions? To my belief the clouds are even more so dependent on the cloud parameters. (perturbed initial conditions ensembles versus perturbed physical parameter ensembles.)

  => The modeled cloud can be more dependent on the cloud parameters. However, the ensemble runs of MIROC-ESM-CHEM were performed by changing initial condition only, and cloud parameters were same among the

ensemble members. Therefore, we can say that the variety among the ensemble members is due to variations in the initial conditions. We modified the sentence as follows:

P.11 L.11–14 "The variability among the ensemble members implies that the cloud amount is considerably affected by the chaotic properties and high sensitivity to the initial state of the Earth system or ESM, because any model settings other than the initial state are the same among the ensemble members."

- Figure 7: I am not sure how much sense it makes to draw regression lines for Ec. There is clearly little correlation. Also; which models years are included in the figure? 2020 – 2069? Please make a note in caption.
  => I agree that correlation of $E_C$ is very little. However, we consider we should keep the regression line for $E_C$ in the figure for consistency in the information included in the figure.
  The model years of 2021–2070 had been included in the previous manuscript; however, it was inappropriate because the last year of the SRM is 2069. We updated the figure with using 2021-2069. Accordingly, Table 3 was slightly updated. We added the year info in the caption as follows:
  Caption of Fig. 7 "Globally and annually averaged relationship between $\Delta T$ and $E_{WV}$ (orange $\diamondsuit$), E_C (blue +), and $E_{SA}$ (green ×) for each year from 2021 to 2069."

- Figure 7 cont.: Have you tried to plot annual means for years 1-10 of the simulations and then decadal means for the remaining 4 decades? This might be more representative for gauging fast vs slow response.
  =>We tried as suggested. However, clarity did not improve, and almost the same results were obtained. Since the clarity did not improve, we keep the figure as before. We consider plotting annual mean is more simple and straightforward.

- Section 3.5 Figure 8: good if you could remind readers at this stage if each model has been weighted by number of ensemble member in these figures.
  => The following note was added to the caption of Fig. 8.
  Caption of Fig. 8 "Note that the multi-model mean is calculated by averaging ensemble means (or single run for models that has no ensembles) of the six models. That is, in the multi-model mean, each run of a model is weighted by the reciprocal of the ensemble number of the model."

- Figure 8 clearly indicates that something is going on with the South Pacific Convergence Zone. Detailed study of this is beyond the topic of your study; however, if this is discussed elsewhere in the literature, it would be great to point to it in the text.

  => We need more detailed analysis to say something meaningful. However, as you mentioned, exploration for a specific region is out of scope of this study. We left it as our future work.

- P13, l25: "However, the situation at TOA is also of interest." Please say why.

  => The reason is simple: the energy budget of the Earth system is closed at TOA. To mention this, we modified the sentence as follows:

  P.13 L.24–25 "However, the situation at TOA is also of interest, because the energy budget of the Earth system is closed at TOA."

- Section 4.2: doesn't the LW heating from the aerosols also increase water vapor in the stratosphere, contributing to ozone losses? This aspect might be mentioned here briefly, as stratospheric O3 amounts do indeed impact human activities at the surface.

  =>The other referee also commented on this section to say this section is confusing. We carefully reconsidered the role and importance of this section in our manuscript. Then we realized that Section 4.2 provides a little information on LW rapid adjustment of WV but we cannot evaluate its importance compared with SW total reactions and LW feedbacks (which is mentioned in your comment and cannot be evaluated easily). We decided to remove this section. We believe keeping focus on SW analysis will benefit readers. Mentioning ozone losses would benefit the reader, so that we added a sentence at the end of text as follows:

  P.17 L1–3 "On the other hand, in the stratosphere, the LW absorption by the injected sulphate aerosols will heat the air and increase water vapour, which contributes to ozone losses (Tilemes et al., 2008; National Research Council, 2015)."

We also corrected the following typos in the manuscript.

- 2070 => 2069 (The SRM term had to be written as year 2020 to 2069 not 2070)

- $E^{TOA}_{SA}$ => $E^{TOA}_{WV}$, $E_{SA}$ => $E_{WV}$ (P.14 L.17, typo)

[revised manuscript text omitted]
_{\mathrm{SA}}^{\mathrm{TOA}}/E_{\mathrm{SA}}$, of course, agrees with $(1 - R_{\mathrm{RCP}}^{\mathrm{as}} - A_{\mathrm{RCP}}^{\mathrm{as}})/(1 - R_{\mathrm{RCP}}^{\mathrm{as}})$, which can be obtained through algebraic manipulation. The difference of $E_{\mathrm{C}}^{\mathrm{TOA}} - E_{\mathrm{C}}$ is $0.12\ \mathrm{Wm}^{-2}$ (16.5 %) for the multi-model mean. Remember that the effect of the cloud amount change includes both changes in reflection rate ($R^{\mathrm{cl}}$) and absorption rate ($A^{\mathrm{cl}}$). The effect of a change in $R^{\mathrm{cl}}$ should appear almost equally at the surface and TOA, as the case for the SRM forcing, because both $R^{\mathrm{cl}}$ and $R^{\mathrm{cs}}$ appear in the Eqs. (7) and (18) in the same way. Therefore, most of $E_{\mathrm{C}}^{\mathrm{TOA}} - E_{\mathrm{C}}$ should be caused by the difference in how the change of the absorption rate affects the net SW at the surface and that at TOA. This is discussed below.

The total reaction at TOA due to the change in water vapour amount shows a negative sign, which is opposite to that at the surface. This disagreement is attributed as follows: Surface cooling reduces the amount of water vapour in the atmosphere and the SW absorption rate decreases. Then, more incoming solar radiation reaches the surface, so that the decrease in water vapour

amount increases SW flux at the surface. On the other hand, when the SW absorption rate decreases, the more upwelling SW that was reflected at the surface pass through the atmosphere and reaches TOA. This leads to a cooling effect. Because the effect of decrease in the SW absorption rate is carried to TOA by the upwelling SW that was reflected at the surface by the rate of $\alpha$, $|E_{\mathrm{SA}}^{\mathrm{TOA}}|$  $|E_{\mathrm{WV}}^{\mathrm{TOA}}|$ is much less than  $|E_{\mathrm{WV}}|$. Note that, in our single layer model, SW absorption above the clouds is not included, so that upwelling SW at TOA reflected by the clouds without reaching the surface is independent of the absorption rate. Therefore, $E_{\mathrm{WV}}^{\mathrm{TOA}}$ could be underestimated, and the change in water vapour  may not be negligible for the energy budget at TOA. Furthermore, we have not explored LW in this study. An analysis on LW rapid adjustment  and feedbacks due to changes in water vapour and clouds is left as our future work.

From the above discussion, we have found that the effect of changes in atmospheric SW absorption rate appears differently between at the surface and at TOA (in its sign and amount), but that in reflection rate appears almost equally. The effect of change in the surface albedo is weaker at TOA than at the surface. We will bear these properties in our mind, when we discuss the influence of SRM on the energy budget of the climate system, which is usually considered at TOA, and human activities, which are mainly performed at the surface.

To fairly compare feedback parameters in G4 with those under greenhouse gas forcing, we decompose the total reactions at TOA into rapid adjustment and feedback in the same manner that we performed in Section 3.4. The rapid adjustment and feedback parameters calculated at TOA are listed in Table S1. The multi-model-averaged feedback parameter of surface albedo in G4 is 0.27 $\mathrm{Wm}^{-2}\mathrm{K}^{-1}$. This value is close to the surface albedo feedback parameter of 0.26 $\mathrm{Wm}^{-2}\mathrm{K}^{-1}$ in A1B scenario (Soden and Held, 2006) and that of 0.30 $\mathrm{Wm}^{-2}\mathrm{K}^{-1}$ in the quadrupled $CO_2$ experiment (Donohoe et al., 2014). On the other hand, the multi-model-averaged feedback parameter of water vapour in G4 is 0.15 $\mathrm{Wm}^{-2}\mathrm{K}^{-1}$ and that (for SW at TOA) in quadrupled $CO_2$ experiment is 0.30 $\mathrm{Wm}^{-2}\mathrm{K}^{-1}$. These comparisons suggest that the SW feedback of surface albedo under sulphate geoengineering is consistent with that under greenhouse gas forcing, whereas that of water vapour is about a half of that under greenhouse gas forcing.

**4.2**

~~This study has concentrated on SW for the reasons described in Section 1; however, it may be valuable for some readers to mention the role of LW. A well-known effect of LW in sulphate aerosol geoengineering is heating of the stratosphere. The sulphate aerosols induced by the injection absorb LW and heat air in the lower the stratosphere (e.g., Heckendorn et al., 2009; Pitari et al. For the energy budget at TOA , increase of the LW absorption results in decrease of the outgoing LW, which manifests as a heating of the climate system. Needless to say, there are many interactions among LW, temperature, and various other components of the climate system through the emission and absorption of LW. Because of such complexity, unlike the SW changes that we have explored in this study, it is difficult to distinguish and estimate the effect of each factor on LW changes.~~

~~One possible and useful analysis for LW is to estimate the rapid adjustment (or response), which is independent of $\Delta T$, by the same method used in Section 3.4. Gregory-like plots are made for the difference of net LW for clear-sky at the surface ($\Delta \mathrm{LW}_{\mathrm{SURF}}^{\mathrm{cs}}$) and at TOA ($\Delta \mathrm{LW}_{\mathrm{TOA}}^{\mathrm{cs}}$) 
[revised manuscript text omitted]

[Figure]

**Figure S1.** Annual cycle of stratospheric sulphate AOD averaged zonally and temporally over 2040–2069 for (a–c) each run of HadGEM2-ES and (d) MIROC-ESM-CHEM-AMP, (e) the prescribed AOD with same color shading, and (f) latitudinal distribution of the temporal means, where #1, #2, and #3 of HadGEM2-ES are shown by solid, dashed, and dotted purple lines, respectively, MIROC-ESM-CHEM-AMP by red line, and the prescribed AOD by black line. Note that HadGEM2-ES's AOD is approximately obtained by subtraction of sulphate aerosol AOD for both stratosphere and troposphere in G4 from that in RCP4.5.

**Table S1.** Same as Table 3 but for values at TOA.

| Models | $Q_{\mathrm{WV}}^{\mathrm{TOA}}$ | $-P_{\mathrm{WV}}^{\mathrm{TOA}}$ | $\mathrm{R}_{\mathrm{WV}}^{\mathrm{TOA}}$ | $Q_{\mathrm{C}}^{\mathrm{TOA}}$ | $-P_{\mathrm{C}}^{\mathrm{TOA}}$ | $\mathrm{R}_{\mathrm{C}}^{\mathrm{TOA}}$ | $Q_{\mathrm{SA}}^{\mathrm{TOA}}$ | $-P_{\mathrm{SA}}^{\mathrm{TOA}}$ | $\mathrm{R}_{\mathrm{SA}}^{\mathrm{TOA}}$ |
|---|---|---|---|---|---|---|---|---|---|
| BNU-ESM | $1.4 \times 10^{-2}$ | 0.15 | 0.87 | 0.80 | $-0.01$ | $-0.01$ | $-3.1 \times 10^{-2}$ | 0.21 | 0.51 |
| CanESM2 | $3.9 \times 10^{-2}$ | 0.12 | 0.63 | 0.56 | $-0.32$ | $-0.18$ | $-6.2 \times 10^{-3}$ | 0.22 | 0.63 |
| HadGEM2-ES | $-2.5 \times 10^{-2}$ | 0.14 | 0.90 | 1.28 | 0.36 | 0.32 | $1.4 \times 10^{-3}$ | 0.17 | 0.70 |
| MIROC-ESM | $1.2 \times 10^{-2}$ | 0.17 | 0.75 | 1.05 | 1.17 | 0.31 | $4.0 \times 10^{-4}$ | 0.37 | 0.52 |
| MIROC-ESM-CHEM | $-2.2 \times 10^{-2}$ | 0.14 | 0.85 | 0.73 | $-0.03$ | $-0.02$ | $6.3 \times 10^{-4}$ | 0.31 | 0.74 |
| MIROC-ESM-CHEM-AMP | $-3.7 \times 10^{-2}$ | 0.15 | 0.81 | 1.98 | 0.58 | 0.29 | $2.8 \times 10^{-2}$ | 0.32 | 0.67 |
| Multi-model mean | $-3.2 \times 10^{-3}$ | 0.15 | 0.80 | 1.07 | 0.29 | 0.12 | $-1.1 \times 10^{-3}$ | 0.27 | 0.63 |

(using run #1 only)

[Figure]

**Figure S2.** Same as Fig. 9 but using one run for each model.

[Figure]

**Figure S3.** Same as Fig. 9 but the three MIROC-based models are weighted by 1/3 for the multi-model means and red lines indicate the means of the three MIROC-based models.

---

## Author Response (AR3)

Dear Editor Prof. Ulrike Lohmann

Please find below our response to your comment and a marked-up manuscript.

Thank you very much for your consideration.

Sincerely, Hiroki Kashimura et al.

[Editor's comment]
Thank you very much for the revised paper. I only have one additional question that I would like to get an answer on. You write: 'the SW feedback of surface albedo under sulphate geoengineering is consistent with that under greenhouse gas forcing, whereas that of water vapour is about a half of that under greenhouse gas forcing". Please add an explanation for this or a hypothesis, why this is the case.

=> We consider that the difference in the water vapour feedback would be due to differences in the vertical temperature profile and/or the atmospheric circulation under sulphate geoengineering and those under greenhouse gas forcing (e.g., McCusker et al., 2015). In contrast, the surface albedo feedback would not depend on such atmospheric features, and mostly depends on surface temperature via changes in sea ice and snow.

To answer quantitative difference (why half?) in the water vapour feedback, more detailed studies are required but we reserve it for future work.

We added the following sentences to the end of Section 4.1:

[revised manuscript text omitted]
^{\uparrow}_{\text{TOA}} - F^{\downarrow}_{\text{SURF}}F^{\uparrow}_{\text{SURF}}}{S^2 - F^{\uparrow 2}_{\text{SURF}}}, \tag{5}$$

for calculating the value of $R$. Then, $A$ is calculated using values of $R$ and $\alpha$ by the following form of Eq. (2):

$$A = (1 - R) - \frac{F^{\downarrow}_{\text{SURF}}}{S}(1 - \alpha R). \tag{6}$$

Note that, $R$, $A$, and $\alpha$ cannot be obtained when $S = 0$ such as during the polar night.

Based on the DB11's single-layer model described above, the strength of the SRM forcing and the total reactions due
5   to changes in the water vapour amount, cloud amount, and surface albedo are estimated using the method described in the remainder of this section. Since GeoMIP participating models provide all-sky and clear-sky values for $F^{\uparrow}_{\text{TOA}}$, $F^{\downarrow}_{\text{SURF}}$, and $F^{\uparrow}_{\text{SURF}}$, values of $R$, $A$, and $\alpha$ can be calculated for both all-sky and clear-sky; superscript "as" is used for all-sky and "cs" for clear-sky. Defining the cloud effects on radiative transfer for a variable $X$ by $X^{\text{cl}} \equiv X^{\text{as}} - X^{\text{cs}}$, the all-sky value is the sum of the clear-sky value and the cloud effect: $X^{\text{as}} = X^{\text{cs}} + X^{\text{cl}}$, where superscript "cl" is for the cloud effect. For further simplicity,
10   the cloud effect on the surface albedo is assumed to be negligible (i.e., $\alpha^{\text{as}} \approx \alpha^{\text{cs}}$), and $\alpha^{\text{as}}$ is used in the following analyses and the superscript omitted. Now, the monthly mean of $R^{\text{cs}}$, $R^{\text{cl}}$, $A^{\text{cs}}$, $A^{\text{cl}}$, and $\alpha$ is calculated on each grid-point for RCP4.5 and G4 experiments.

Net SW at the surface is a key variable in this study and can be written as follows:

$$F^{\text{net}}_{\text{SURF}} \equiv F^{\downarrow \text{as}}_{\text{SURF}} - F^{\uparrow \text{as}}_{\text{SURF}} = (1 - \alpha)S\left[\frac{1 - (R^{\text{cs}} + R^{\text{cl}}) - (A^{\text{cs}} + A^{\text{cl}})}{1 - \alpha(R^{\text{cs}} + R^{\text{cl}})}\right]. \tag{7}$$

Here, $F^{\text{net}}_{\text{SURF}}$ is regarded as a function of $S$, $R^{\text{cs}}$, $R^{\text{cl}}$, $A^{\text{cs}}$, $A^{\text{cl}}$, and $\alpha$. The difference of $F^{\text{net}}_{\text{SURF}}$ between RCP4.5 and G4
15   experiments is defined as

$$\Delta F^{\text{net}}_{\text{SURF}} \equiv F^{\text{net}}_{\text{SURF}}(S, R^{\text{cs}}_{\text{G4}}, R^{\text{cl}}_{\text{G4}}, A^{\text{cs}}_{\text{G4}}, A^{\text{cl}}_{\text{G4}}, \alpha_{\text{G4}}) - F^{\text{net}}_{\text{SURF}}(S, R^{\text{cs}}_{\text{RCP}}, R^{\text{cl}}_{\text{RCP}}, A^{\text{cs}}_{\text{RCP}}, A^{\text{cl}}_{\text{RCP}}, \alpha_{\text{RCP}}), \tag{8}$$

where the experiment names are indicated by subscripts "RCP" and "G4". ($S$, the TOA downwelling solar radiation, is same for RCP4.5 and G4.) Hereafter, $F^{\text{net}}_{\text{SURF}}(\text{RCP}) \equiv F^{\text{net}}_{\text{SURF}}(S, R^{\text{cs}}_{\text{RCP}}, R^{\text{cl}}_{\text{RCP}}, A^{\text{cs}}_{\text{RCP}}, A^{\text{cl}}_{\text{RCP}}, \alpha_{\text{RCP}})$ is written for convenience.

[revised manuscript text omitted]

**Figure S1.** Annual cycle of stratospheric sulphate AOD averaged zonally and temporally over 2040–2069 for (a–c) each run of HadGEM2-ES and (d) MIROC-ESM-CHEM-AMP, (e) the prescribed AOD with same color shading, and (f) latitudinal distribution of the temporal means, where #1, #2, and #3 of HadGEM2-ES are shown by solid, dashed, and dotted purple lines, respectively, MIROC-ESM-CHEM-AMP by red line, and the prescribed AOD by black line. Note that HadGEM2-ES's AOD is approximately obtained by subtraction of sulphate aerosol AOD for both stratosphere and troposphere in G4 from that in RCP4.5.

[Figure]

**Figure S2.** Same as Fig. 9 but using one run for each model.

[Figure]

**Figure S3.** Same as Fig. 9 but the three MIROC-based models are weighted by 1/3 for the multi-model means and red lines indicate the means of the three MIROC-based models.

**Table S1.** Same as Table 3 but for values at TOA.

| Models | $Q_{\mathrm{WV}}^{\mathrm{TOA}}$ | $-P_{\mathrm{WV}}^{\mathrm{TOA}}$ | $R_{\mathrm{WV}}^{\mathrm{TOA}}$ | $Q_{\mathrm{C}}^{\mathrm{TOA}}$ | $-P_{\mathrm{C}}^{\mathrm{TOA}}$ | $R_{\mathrm{C}}^{\mathrm{TOA}}$ | $Q_{\mathrm{SA}}^{\mathrm{TOA}}$ | $-P_{\mathrm{SA}}^{\mathrm{TOA}}$ | $R_{\mathrm{SA}}^{\mathrm{TOA}}$ |
|---|---|---|---|---|---|---|---|---|---|
| BNU-ESM | $1.4 \times 10^{-2}$ | 0.15 | 0.87 | 0.80 | $-0.01$ | $-0.01$ | $-3.1 \times 10^{-2}$ | 0.21 | 0.51 |
| CanESM2 | $3.9 \times 10^{-2}$ | 0.12 | 0.63 | 0.56 | $-0.32$ | $-0.18$ | $-6.2 \times 10^{-3}$ | 0.22 | 0.63 |
| HadGEM2-ES | $-2.5 \times 10^{-2}$ | 0.14 | 0.90 | 1.28 | 0.36 | 0.32 | $1.4 \times 10^{-3}$ | 0.17 | 0.70 |
| MIROC-ESM | $1.2 \times 10^{-2}$ | 0.17 | 0.75 | 1.05 | 1.17 | 0.31 | $4.0 \times 10^{-4}$ | 0.37 | 0.52 |
| MIROC-ESM-CHEM | $-2.2 \times 10^{-2}$ | 0.14 | 0.85 | 0.73 | $-0.03$ | $-0.02$ | $6.3 \times 10^{-4}$ | 0.31 | 0.74 |
| MIROC-ESM-CHEM-AMP | $-3.7 \times 10^{-2}$ | 0.15 | 0.81 | 1.98 | 0.58 | 0.29 | $2.8 \times 10^{-2}$ | 0.32 | 0.67 |
| Multi-model mean | $-3.2 \times 10^{-3}$ | 0.15 | 0.80 | 1.07 | 0.29 | 0.12 | $-1.1 \times 10^{-3}$ | 0.27 | 0.63 |